# `TimeWak`: Temporal Chained-Hashing Watermark for Time Series Data

**Zhi Wen Soi**[1,*]    **Chaoyi Zhu**[2,*]    **Fouad Abiad**[2]    **Aditya Shankar**[2]

**Jeroen M. Galjaard**[2]    **Huijuan Wang**[2]    **Lydia Y. Chen**[1,2,†]

[1]University of Neuchâtel    [2]Delft University of Technology

`{zhi.soi, yiyu.chen}@unine.ch`
`{c.zhu-2, f.abiad, a.shankar, j.m.galjaard, h.wang}@tudelft.nl`

## Abstract

Synthetic time series generated by diffusion models enable sharing privacy-sensitive datasets, such as patients' functional MRI records. Key criteria for synthetic data include high data utility and traceability to verify the data source. Recent watermarking methods embed in homogeneous latent spaces, but state-of-the-art time series generators operate in data space, making latent-based watermarking incompatible. This creates the challenge of watermarking directly in data space while handling feature heterogeneity and temporal dependencies. We propose `TimeWak`, the first watermarking algorithm for multivariate time series diffusion models. To handle temporal dependence and spatial heterogeneity, `TimeWak` embeds a temporal chained-hashing watermark directly within the temporal-feature data space. The other unique feature is the $\epsilon$-exact inversion, which addresses the non-uniform reconstruction error distribution across features from inverting the diffusion process to detect watermarks. We derive the error bound of inverting multivariate time series while preserving robust watermark detectability. We extensively evaluate `TimeWak` on its impact on synthetic data quality, watermark detectability, and robustness under various post-editing attacks, against five datasets and baselines of different temporal lengths. Our results show that `TimeWak` achieves improvements of 61.96% in context-FID score, and 8.44% in correlational scores against the strongest state-of-the-art baseline, while remaining consistently detectable. Our code is available at `https://github.com/soizhiwen/TimeWak`.

## 1 Introduction

Multivariate time series data drive key applications in healthcare [19], finance [13], and science [25]. However, access to real-world datasets is often restricted by privacy regulations, limited availability, and high acquisition costs. To address these issues, synthetic time series generated by models are increasingly adopted as practical alternatives [9, 25]. Among generative techniques, *diffusion* models [10] have gained prominence for producing high-quality samples, often outperforming the mainstream Generative Adversarial Networks and Variational Autoencoders [6, 12].

Beyond generation quality, *traceability* is equally critical, as it ensures verifiability and safeguards against misuse [18, 41]. In this context, *watermarking* has become the de-facto approach for tracking and auditing synthetic data [32, 18]. The challenge lies in striking a delicate balance: embedding imperceptible signals that preserve the quality of generated content while remaining detectable, even

---

*Equal contribution.

†Corresponding author.

39th Conference on Neural Information Processing Systems (NeurIPS 2025).

under post-processing [40]. Recent works embed watermarks *during* generation by adding them into the *latent* space, offering advantages such as generality and lightweight computation [32, 41]. However, latent-space watermarks are not always viable, especially since many state-of-the-art (SOTA) time series generators operate within the data space [26, 1, 34]. Additionally, latent generators introduce a trade-off in detectability, as diffusion inversion and the encode-decode cycle are inherently lossy and can degrade the watermarks [21].

While generation-time watermarks have proven effective for images [32] and tables [41], their applicability to time series data remains unexplored. Multivariate time series data possess temporal dependencies and heterogeneous features, like gender versus income. The ensuing challenges are twofold: (i) embedding watermarks *directly* in the data space while preserving inter- *and* intra-variate temporal dependencies of the generated time series, and (ii) ensuring accurate watermark detection despite the lossy nature of diffusion inversion, whilst handling mixed feature types.

We propose `TimeWak`, the first *generation-time* watermark for multivariate time series diffusion models, featuring **temporal chained-hashing** with $\epsilon$**-exact inversion**. `TimeWak` first embeds cyclic watermark patterns, i.e., the positional seeds of Gaussian noises, along the temporal direction. First, we *chained-hash* the seeds along the temporal axis, then shuffle the seeds across features to maintain temporal correlations while preserving the unique characteristics of each feature. To ensure reliable watermark detection, we introduce an $\epsilon$-exact inversion strategy that makes a practical concession in the otherwise exactly invertible diffusion process: the *Bi-Directional Integration Approximation* (BDIA) [36]. We further provide theoretical guarantees on the resulting inversion error in Appendix C.

We evaluate `TimeWak` against five SOTA watermarking methods on five datasets under varying temporal lengths. `TimeWak` achieves the best detectability under six post-editing attack configurations with the minimum data quality degradation. Additionally, results show that `TimeWak` preserves temporal and cross-variate dependencies with high quality with near-exact watermark bit reconstruction. To summarize, we list our contributions as follows:

- We propose `TimeWak`, the first generation-time watermarking scheme for multivariate time series diffusion models, preserving realistic spatio-temporal dependencies while remaining detectable.

- To preserve temporal characteristics and boost robustness against post-processing operations, we design a *temporal chained-hashing* scheme that embeds watermark seeds along the temporal direction, followed by a shuffle across features.

- For robust detectability, we propose $\epsilon$-*exact* inversion by extending BDIA sampling into a data space diffusion generator, and provide a theoretical error bound analysis.

- Our extensive evaluation shows that `TimeWak` achieves up to **61.96%** better context-FID scores and **8.44%** better correlation scores compared to the strongest SOTA watermarking method.

## 2   Related work

We summarize related works on watermarking diffusion models according to their generating method (post-processing or generation-time generation), and data modality. To the best of our knowledge, our work is the first generation-time watermarking scheme for time series diffusion models.

**Watermarking diffusion models.** Watermarking has become a critical solution for tracing and authenticating machine-generated content. *Post-generation* techniques embed watermarks after synthesis, often degrading the generated data's quality due to direct modifications [5, 38]. Alternatively, recent advancements embed watermarks within the training process. Studies such as *Stable Signature* [7] and *FixedWM* [18] *fine-tune* diffusion models to embed and extract watermarks. However, these methods modify model parameters, risking overfitting which hurts generalizability.

**Watermarking images.** *Tree-Ring* (TR) [29] embeds the watermark during sampling by modifying the initial noise latent vector in the Fourier space. However, it disrupts the Gaussian noise distribution, reducing the diversity and quality of generated samples. *Gaussian Shading* (GS) [32] improves robustness by embedding watermarks directly in the latent space using invertible transformations. However, GS is tailored towards images and requires reversing synthetic samples into the latent space for detection, which is a noisy and error-prone process that limits the detection accuracy.

**Watermarking tables.** *TabWak* [41] watermarks tabular data in latent space using seeded *self-cloning*, *shuffling* with a secret key, and a *valid-bit* mechanism. However, it relies on latent models and does

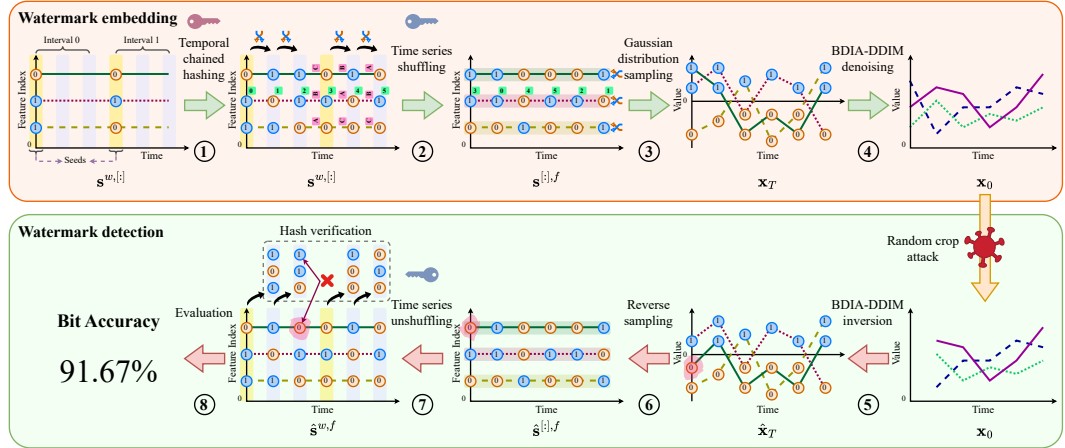

Figure 1: Overview of `TimeWak`. First, we assign random seeds at the beginning of each interval. ① Temporally chained-hashing. A, B, and C ( pink ) show seeds being copied from the previous step and the feature order shuffled. ② Shuffling the seeds for each series. Positional indices are highlighted in green . ③ Constructing an initial Gaussian noise. ④ Generating multivariate time series. ⑤ Reversing the diffusion process. ⑥ Recovering the watermark seed. ⑦ Unshuffling the seeds in the opposite way they were shuffled. ⑧ Bit accuracy between the hash and recovered seed.

not account for temporal dependencies in a time series. Furthermore, its detectability is limited by the invertibility of the diffusion process and the lossy conversion to and from latent representations.

## 3 `TimeWak`

We first highlight the unique challenges of watermarking multivariate time series, motivating the design of `TimeWak`, shown in Figure 1. Then we introduce `TimeWak`'s key novelties: (i) temporal chained-hashing the watermark seeds cyclically along the temporal axis; (ii) shuffling the seeds across features, accounting for feature heterogeneity in the time series; and (iii) an adapted BDIA-DDIM sampling method with a theoretically bounded $\epsilon$-exact inversion, enhancing robust detectability. Key notations are summarized in Appendix A.

### 3.1 Time series diffusion and observations

**Time series diffusion.** We define a time series sample of $F$ features (variates) and $W$ timesteps as $\mathbf{x}_0 \in \mathbb{R}^{W \times F}$, with $\mathbf{x}_0^{w,f}$ denoting the value of feature $f$ at timestep $w$. Unlike image and tabular diffusion, SOTA time series diffusion models operate on the data space, which have heterogeneous features, e.g., income vs. gender, and temporal dependence [26, 1, 34]. These time series generators use *Denoising Diffusion Implicit Models* (DDIM) [23] to synthesize time series starting from Gaussian noise, $\mathbf{x}_T$, by iteratively denoising over $T$ steps, i.e., $\mathbf{x}_T, \mathbf{x}_{T-1}, \ldots, \mathbf{x}_0$. Specifically, DDIM sets the state $\mathbf{x}_{t-1}$ at diffusion step $t-1$ as follows:

$$\mathbf{x}_{t-1} = \alpha_{t-1} \left( \frac{\mathbf{x}_t - \sigma_t \hat{\boldsymbol{\epsilon}}_{\boldsymbol{\theta}}(\mathbf{x}_t, t)}{\alpha_t} \right) + \sigma_{t-1} \hat{\boldsymbol{\epsilon}}_{\boldsymbol{\theta}}(\mathbf{x}_t, t), \tag{1}$$

where $\alpha_t$ and $\sigma_t$ are time-dependent diffusion coefficients, and $\hat{\boldsymbol{\epsilon}}_{\boldsymbol{\theta}}$ represents the model's noise estimate. DDIM approximates $\mathbf{x}_t$ as follows:

$$\mathbf{x}_t = \alpha_t \left( \frac{\mathbf{x}_{t-1} - \sigma_{t-1} \hat{\boldsymbol{\epsilon}}_{\boldsymbol{\theta}}(\mathbf{x}_t, t)}{\alpha_{t-1}} \right) + \sigma_t \hat{\boldsymbol{\epsilon}}_{\boldsymbol{\theta}}(\mathbf{x}_t, t) \approx \alpha_t \left( \frac{\mathbf{x}_{t-1} - \sigma_{t-1} \hat{\boldsymbol{\epsilon}}_{\boldsymbol{\theta}}(\mathbf{x}_{t-1}, t)}{\alpha_{t-1}} \right) + \sigma_t \hat{\boldsymbol{\epsilon}}_{\boldsymbol{\theta}}(\mathbf{x}_{t-1}, t). \tag{2}$$

However, this approximation introduces errors, producing inconsistencies between the forward and backward processes. It also introduces the following time series-specific watermarking challenges:

**Spatial heterogeneity.** Watermarks must be embedded directly within the temporal and feature spaces of the data. Features can be very diverse, e.g., gender vs. income distribution, which increases

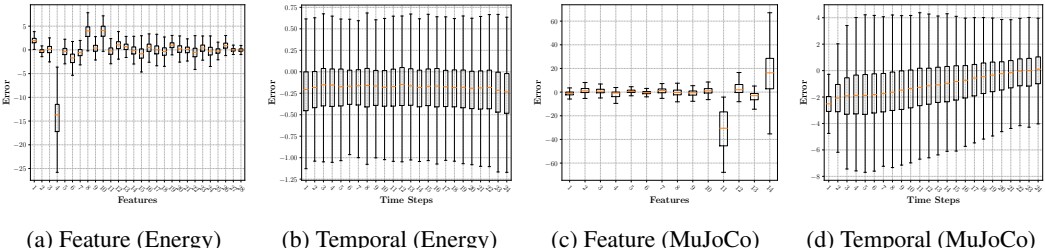

| (a) Feature (Energy) | (b) Temporal (Energy) | (c) Feature (MuJoCo) | (d) Temporal (MuJoCo) |

Figure 2: Average reconstruction error distribution across feature indices and timesteps on Diffusion-TS with DDIM and DDIM inversion. Reconstruction error is the signed absolute difference between reconstructed and original values.

the difficulty of detecting watermarks. Specifically, the key detection step inverts the time series back to Gaussian noise. Unfortunately, this inversion process is inexact, yielding reconstruction errors during noise-estimation. Figure 2 shows the impact of heterogeneity on the reconstruction errors for the Energy and MuJoCo datasets. Due to spatial heterogeneity, the reconstruction errors across the features vary significantly more than they do along the temporal axis. Existing tabular watermarks [41] implicitly assume a uniform distribution across features and compare watermark seeds across features, which prevents reliable watermark verification in multivariate time series.

**Temporal dependence.** Time series consist of values that are inherently correlated across timesteps. It is critical to preserve such temporal consistencies when generating time series. Consequently, reconstruction errors are not fully independent across timesteps within each sample, as errors at neighbouring timesteps often exhibit stronger correlations than more distant ones. To ensure robustness, solutions must embed the watermark in a way that respects these temporal dependencies while remaining detectable. This requires designing watermarking strategies that align with the sequential nature of time series diffusion models, invalidating the applicability of existing watermark approaches that neglect the temporal dependence of time series data.

### 3.2 `TimeWak` **algorithm**

To address the challenges of spatial heterogeneity and temporal dependence, we propose `TimeWak`, a method that enables per-sample watermark detection while mitigating non-uniform reconstruction errors across features and preserving temporal structure. Through a structured propagation mechanism, `TimeWak` enhances the watermark's robustness, even in the presence of inversion errors.

**Overview.** We begin with a high-level overview of `TimeWak`'s watermarking pipeline, which consists of four main stages: watermark embedding, time series generation, inversion, and detection. Each stage involves multiple steps that we describe briefly here and explain in detail in the subsequent sections. Following Figure 1, the complete process works as follows:

1. **Embedding I:** generating watermark seeds ($\mathbf{s}$). We first split a multivariate time series into intervals along the time axis, then we randomly sample seeds $\mathbf{s}^{w,[:]}$ (with values in $\{0, 1\}$) at the start of each interval. ① We temporally chain-hash the seeds in a cyclic manner across timesteps until the end of the current interval, by applying a unique permutation key at each timestep. Then, ② we independently shuffle the seeds for each feature using distinct permutation keys.

2. **Embedding II:** generating time series from the watermarked seeds ($\mathbf{s} + \mathbf{x}_T \rightarrow \mathbf{x}_0$). ③ Sampling from a feature-wise pseudo-random Gaussian distribution based on all the aforementioned seeds, where the seeds determine the sign of the sampled values ($\mathbf{s}^{w,f} = 1$ becomes positive, $\mathbf{s}^{w,f} = 0$ becomes negative). These noise signals are used as input to a BDIA variant of a DDIM diffusion model, which is then ④ used to generate a multivariate time series. An attack, such as a random crop attack, may occur at this stage.

3. **Detection I**: inversion of the time series ($\mathbf{x}_0 \rightarrow \hat{\mathbf{x}}_T$). ⑤ The inverse BDIA-DDIM process is applied to the time series, inverting it to Gaussian noise $\hat{\mathbf{x}}_T$. Then ⑥ we reverse-sample each time series to get the seeds (positive values become 1, negative values become 0); we now have the shuffled seed features.

4. **Detection II:** watermark detection ($\hat{\mathbf{x}}_T \rightarrow \hat{\mathbf{s}}$). Given shuffled seeded features, we ⑦ unshuffle the seeds in the opposite way they were shuffled (using the inverse of the permutation keys of

step ②), to obtain the retrieved seed features $\hat{\mathbf{s}}$. We then ⑧ verify the hash of these retrieved seed features by comparing them with the original temporal chained-hash seeds $\mathbf{s}$, for which we compute the bit accuracy between the hashed and recovered versions of each seed.

### 3.2.1 Chained-hashing watermark seeds ($\mathbf{s} + \mathbf{x}_T \to \mathbf{x}_0$)

Existing watermarking methods assign a watermark seed to each feature dimension with $L$ bits, forming a seed matrix of dimensions $W \times F$, denoted as $\mathbf{s} \in \mathbb{R}^{W \times F}$, where $W$ and $F$ are the respective total timesteps and total features of the time series [32, 41]. However, such approaches lack the ability to leverage temporal dependencies and may introduce inconsistencies across timesteps. To improve the watermark's temporal coherence, we partition the time series data into $n = \lfloor W/H \rfloor$ non-overlapping intervals, each of length $H$. At the start of each interval, the watermark seed across all features, $\mathbf{s}^{kH+1,[:]} \in \mathbb{R}^F$, is sampled from a discrete uniform distribution $\mathcal{U}\left(\{0, L-1\}\right)^F$, with $[:]$ denoting all indices along a dimension.

Within each interval, the watermark seed evolves over timesteps using a **temporal chained-hashing** mechanism. Specifically, for all features, the seed at timestep $w$ is recursively derived from the seed at the previous timestep $w-1$, ensuring temporal consistency. Formally, for $k = 0, \ldots, n-1$, we initialize the watermark seed as:

$$\mathbf{s}^{w,[:]} = \begin{cases} \mathcal{U}\left(\{0, L-1\}\right)^F & \textbf{if } w = kH + 1, \\ \mathcal{H}\left(\kappa, w, \mathbf{s}^{w-1,[:]}\right) & \textbf{otherwise}, \end{cases} \tag{3}$$

where $\kappa$ is a cryptographic key controlling the hashing process, $\mathcal{H}$ is a deterministic permutation hash function ensuring temporal consistency, and $\mathcal{U}\left(\{0, L-1\}\right)^F$ is a vector of $F$ i.i.d discrete uniform samples over $\{0, 1, \ldots, L-1\}$. The parameter $n = \lfloor W/H \rfloor$ denotes the total integer count of intervals, while $k$ indexes the intervals, ranging from 0 to $n-1$. While temporal chaining preserves coherence across timesteps by linking each seed to its past, it may lead to repetitive patterns across intervals. To increase diversity, we further permute the seeds along the temporal axis for each feature:

$$\mathbf{s}^{[:],f} \leftarrow \pi_\kappa(\mathbf{s}^{[:],f}), \tag{4}$$

where $\pi_\kappa$ is a permutation function parameterized by the cryptographic key $\kappa$. This step preserves inter-feature seed correlations while adding generation diversity.

After obtaining the watermark seed, we construct an initial Gaussian noise sample as follows. First, we draw a variable from the continuous uniform distribution $u \sim \mathcal{U}(0, 1)$ and use it to generate the noise variable $\mathbf{x}_T^{w,f}$ at diffusion step $T$ as:

$$\mathbf{x}_T^{w,f} = \Phi^{-1}\left(\frac{u + \mathbf{s}^{w,f}}{L}\right), \tag{5}$$

where $\Phi^{-1}(\cdot)$ is the percent point function (PPF) of the standard Gaussian distribution $\Phi(\cdot)$, and $\mathbf{s}^{w,f}$ is the watermark seed for feature $f$ at timestep $w$. Finally, the final time series sample $\mathbf{x}_0$ is obtained by denoising the initial noise $\mathbf{x}_T$ with the learned diffusion model.

### 3.2.2 $\epsilon$-Exact inversion ($\mathbf{x}_0 \to \hat{\mathbf{x}}_T$)

Here, we propose a near-lossless inversion procedure by adopting the Bi-directional Integration Approximation (BDIA) technique [36], a novel approach to address inconsistencies in DDIM inversion [24]. We introduce a practical approximation in BDIA by removing the assumption of known $\mathbf{x}_1$, and derive the bound of the inversion error. BDIA improves upon DDIM by jointly leveraging the forward and backward diffusion updates. Specifically, obtaining each $\mathbf{x}_{t-1}$ as a linear combination of $(\mathbf{x}_{t+1}, \mathbf{x}_t, \hat{\epsilon}_{\boldsymbol{\theta}}(\mathbf{x}_t, t))$, where $\hat{\epsilon}_{\boldsymbol{\theta}}$ represents the noise estimator of the diffusion model:

$$\mathbf{x}_{t-1} = \gamma\left(\mathbf{x}_{t+1} - \mathbf{x}_t\right) - \gamma\left(\frac{\mathbf{x}_t}{a_{t+1}} - \frac{b_{t+1}}{a_{t+1}}\hat{\epsilon}_{\boldsymbol{\theta}}(\mathbf{x}_t, t) - \mathbf{x}_t\right) + \left(a_t\mathbf{x}_t + b_t\hat{\epsilon}_{\boldsymbol{\theta}}(\mathbf{x}_t, t)\right), \tag{6}$$

where $\gamma \in [0, 1]$, $a_t$ and $b_t$ are differentiable functions of $t$ with bounded derivatives. Consequently, the inversion process can be directly calculated without approximation as follows:

$$\mathbf{x}_{t+1} = \frac{\mathbf{x}_{t-1}}{\gamma} - \frac{1}{\gamma}\left(a_t\mathbf{x}_t + b_t\hat{\epsilon}_{\boldsymbol{\theta}}(\mathbf{x}_t, t)\right) + \left(\frac{\mathbf{x}_t}{a_{t+1}} - \frac{b_{t+1}}{a_{t+1}}\hat{\epsilon}_{\boldsymbol{\theta}}(\mathbf{x}_t, t)\right). \tag{7}$$

By design, the introduced symmetry ensures time-reversible updates, meaning that if $\mathbf{x}_t$ and $\mathbf{x}_{t-1}$ are known, $\mathbf{x}_{t+1}$ can be computed without error. However, obtaining an exact inversion requires knowing both $\mathbf{x}_1$ and $\mathbf{x}_0$, the latter of which is available only as the model's denoised sample in practice.

To address this limitation, we introduce an adaptation to BDIA: we directly approximate $\mathbf{x}_1$ by equating it to $\mathbf{x}_0$. This seemingly simple estimation effectively enables practical application of BDIA while maintaining reasonable accuracy. To quantify the estimation error, we establish Theorem 3.1, which demonstrates that the final error remains bounded in terms of the initial estimation $\epsilon = \mathbf{x}_1 - \mathbf{x}_2$. We refer to this property as '$\epsilon$-exact' inversion. We defer our proof of Theorem 3.1 to Appendix C.

**Theorem 3.1.** *Let $\{\mathbf{x}_t\}_{t=0}^{T}$ be the sequence of diffusion states governed by the BDIA-DDIM recurrence for a given dataset, following Equation (7). Given the noise estimator $\hat{\epsilon}_{\theta}$ follows Assumption 1. Suppose that instead of the exact terminal state $\mathbf{x}_1$, an approximation of $\mathbf{x}_1$, termed as $\mathbf{x}_1^{approx}$, is used with a small perturbation $\epsilon$, given by:*

$$\mathbf{x}_1^{approx} = \mathbf{x}_2 = \mathbf{x}_1^{orig} + \epsilon. \tag{8}$$

*Let the propagated error at time $t$ be defined as,*

$$\delta_t = \|\mathbf{x}_t^{approx} - \mathbf{x}_t^{orig}\|. \tag{9}$$

*Then, for $t \geq 1$, the error is bounded by,*

$$\|\delta_T\| \leq |\epsilon| \prod_{t=1}^{T-1} \left( \left| \frac{1}{\gamma} - \frac{a_t}{\gamma} + \frac{1}{a_{t+1}} \right| + \frac{b_t}{\gamma} \Delta_t + \frac{b_{t+1}}{a_{t+1}} \Delta_t \right), \tag{10}$$

*where $\Delta_t$ quantifies the sensitivity of the noise estimator at timestep $t$.*

**Assumption 1** (Lipschitz continuity of the noise estimator). *There exists a time-dependent constant $\Delta_t > 0$ and $\delta_t > 0$ such that, for any diffusion state $\mathbf{x}_t^{orig}$ encountered during sampling from a given dataset and any $\mathbf{x}_t^{approx}$ satisfying $\|\mathbf{x}_t^{approx} - \mathbf{x}_t^{orig}\| \leq \delta_t$, the noise estimator $\hat{\epsilon}_{\theta}$ satisfies the Lipschitz condition:*

$$\|\hat{\epsilon}_{\theta}(\mathbf{x}_t^{approx}, t) - \hat{\epsilon}_{\theta}(\mathbf{x}_t^{orig}, t)\| \leq \Delta_t \left\| \mathbf{x}_t^{approx} - \mathbf{x}_t^{orig} \right\|. \tag{11}$$

### 3.2.3 Watermark detection ($\mathbf{x}_0 \rightarrow \hat{\mathbf{x}}_T \rightarrow \hat{\mathbf{s}}$)

To verify the presence of `TimeWak`'s watermark in a generated time series, we first recover the initial noise used in the generative process via **diffusion inversion**. Given a time series instance $\mathbf{x}_0$, we estimate the initial noise $\hat{\mathbf{x}}_T$ through inversion. The watermark seed at each timestep and feature is then recovered by inverting the Gaussian mapping as follows:

$$\hat{\mathbf{s}}^{w,f} = \lfloor L \cdot \Phi(\hat{\mathbf{x}}_T^{w,f}) \rfloor, \tag{12}$$

where $\Phi(\cdot)$ is the cumulative distribution function (CDF) of the standard Gaussian distribution. This operation reconstructs the discrete watermark values embedded during sample generation.

Since the watermarking mechanism applies a feature-wise permutation controlled by a cryptographic key, the extracted watermark values are initially shuffled across feature dimensions. We restore the original seed assignments along the timesteps using the inverse permutation:

$$\hat{\mathbf{s}}^{[:],f} \leftarrow \pi_{\kappa}^{-1}(\hat{\mathbf{s}}^{[:],f}). \tag{13}$$

Beyond feature-space consistency, the extracted watermark sequence should exhibit structured temporal dependencies. Specifically, the watermark seed at each step must follow a predefined hash function, given by:

$$\forall w \neq kH + 1, \quad \hat{\mathbf{s}}^{w,[:]} = \mathcal{H}(\kappa, w, \hat{\mathbf{s}}^{w-1,[:]}). \tag{14}$$

To quantify detection confidence, we compute the **bit accuracy** of the extracted watermark sequence, measuring the proportion of correctly recovered bits:

$$\text{Acc} = \frac{1}{|\mathcal{W}^*|F} \sum_{w \in \mathcal{W}^*} \sum_{f=1}^{F} \mathbb{I}\left[\hat{\mathbf{s}}^{w,f} = \mathbf{s}^{w,f}\right], \tag{15}$$

where $\mathbb{I}[\,\cdot\,]$ is an indicator function that evaluates to 1 if the extracted bit matches the ground-truth watermark, and $\mathcal{W}^* = \{w \mid w \neq kH + 1\}$ represents the valid timesteps for comparison. By combining diffusion inversion, feature unshuffling, and temporal consistency verification, this detection framework ensures robust identification of watermarked time series samples.

To assess the statistical significance, we compute the Z-score to measure how strongly the observed bit accuracy deviates from the expected accuracy under a null hypothesis, detailed in Appendix D.2.

## 4 Evaluation

### 4.1 Experiments setup

**Datasets.** We use five time series datasets to evaluate `TimeWak`'s impact on generation quality, watermark detection accuracy, and robustness towards post-editing operations. These are: *Stocks* [33], *ETTh* [39], *MuJoCo* [27], *Energy* [3], and *fMRI* [22]. Additional dataset details are in Appendix D.1.

**Metrics.** *Synthetic data quality:* Context-FID score [11] measures the closeness between the real and synthetic time series distributions using the Fréchet distance [8]. Correlational score [15] measures the cross-correlation error between the real and synthetic multivariates. Discriminative score [33] trains a classifier to distinguish between synthetic and real data, with low scores implying they are indistinguishable. Predictive score [33] measures the downstream task performance by training a sequence model on the synthetic data and evaluating on real data. *Watermark detectability:* Z-score quantifies the difference in mean values between synthetic data with and without the watermark, with larger positive values indicating better detectability. TPR@X%FPR measures the True Positive Rate (TPR) at a fixed False Positive Rate (FPR) of X% in detecting watermarked time series. We provide additional details on these metrics in Appendix D.2.

**Baselines.** We compare against three sampling-based diffusion watermarks: TR [29], GS [32], and TabWak [41]. We also compare with TabWak$^\top$, an adaptation of TabWak that transposes the representation and watermarks along the temporal axis instead of feature-wise. We also compare against a post-generation watermarking method for time series, *Heads Tails Watermark* (HTW), which embeds a watermark by slightly adjusting the time series values by assigning a 'heads' or 'tails' based on a predefined ratio on the proportion of 'heads' values [28]. Detailed implementations of these methods are provided in Appendix D.3. Hardware specifications are detailed in Appendix D.4.

### 4.2 Synthetic data quality and watermark detectability

When evaluating synthetic time series quality, Table 1 shows that `TimeWak` consistently delivers top-tier performance across all metrics, outperforming or comparable to other baselines like HTW and TabWak$^\top$. While HTW sometimes surpasses un-watermarked data, possibly due to subtle perturbations introduced during watermarking that unintentionally bring synthetic samples closer to the ground truth, it fails to offer strong detectability, as reflected in its low Z-scores. In contrast, `TimeWak` and TabWak$^\top$ offer a far more favorable trade-off between quality and detectability. When benchmarked against TabWak$^\top$, `TimeWak` shows substantial gains, achieving up to 61.96% better Context-FID score on MuJoCo and 8.44% better correlation score on fMRI. Moreover, low discriminative and predictive scores further emphasize that `TimeWak`'s watermarking remains imperceptible and does not degrade downstream utility. This is made possible by its temporal chained-hashing mechanism, which precisely embeds the watermark while preserving both temporal structure and inter-variate relationships. Meanwhile, traditional image-based watermarking methods such as TR and GS perform poorly across all quality metrics. These methods struggle with time series data because they are optimized for spatial domains. Time series, however, are governed by temporal continuity and feature heterogeneity, thus requiring fundamentally different treatment.

`TimeWak` achieves significant improvements in detection performance. Using $\epsilon$-exact inversion via BDIA-DDIM, it reconstructs high-fidelity noise estimates $\hat{\mathbf{x}}_T$ that closely resemble the ground truth $\mathbf{x}_T$. This results in consistently higher Z-scores across all datasets, outperforming all baselines, except on the fMRI dataset, where TabWak$^\top$ slightly edges out. Unlike GS, which maintains moderate detection at the cost of quality, or TR, which fails on both fronts, `TimeWak` delivers strong detectability without compromising fidelity.

Table 1: Results of synthetic time series quality and watermark detectability. No watermarking ('W/O') is included. Quality metrics are for 24-length sequences. Best results are in **bold**, and second-best are underlined.

| Dataset | Method | Quality Metric ↓ | | | | Z-score ↑ | | |
| --- | --- | --- | --- | --- | --- | --- | --- | --- |
| | | Context-FID | Correlational | Discriminative | Predictive | 24-length | 64-length | 128-length |
| Stocks | W/O | 0.258±0.047 | 0.027±0.015 | 0.120±0.049 | 0.038±0.000 | - | - | - |
| | TR | 1.069±0.231 | 0.091±0.007 | 0.209±0.056 | 0.039±0.000 | 0.43±0.04 | 0.40±0.08 | 0.08±0.10 |
| | GS | 8.802±2.415 | 0.052±0.026 | 0.403±0.031 | 0.041±0.003 | 86.07±0.74 | 148.92±1.08 | 172.23±1.08 |
| | HTW | 0.279±0.052 | 0.017±0.003 | 0.122±0.034 | **0.037±0.000** | 4.45±0.62 | 7.34±0.94 | 10.39±1.33 |
| | TabWak | 0.292±0.064 | 0.017±0.012 | 0.124±0.024 | 0.038±0.000 | -67.22±1.17 | 16.14±0.89 | -10.49±1.28 |
| | TabWak$^\top$ | 0.314±0.071 | **0.016±0.017** | 0.132±0.022 | **0.037±0.000** | 55.39±0.86 | 88.81±0.82 | 129.39±0.90 |
| | TimeWak | **0.277±0.019** | 0.020±0.018 | **0.120±0.039** | 0.038±0.000 | **182.10±0.73** | **395.34±1.24** | **550.05±1.18** |
| ETTh | W/O | 0.232±0.018 | 0.086±0.023 | 0.093±0.016 | 0.120±0.009 | - | - | - |
| | TR | 1.570±0.102 | 0.187±0.017 | 0.283±0.020 | 0.134±0.004 | 7.84±0.12 | 7.73±0.13 | 6.18±0.16 |
| | GS | 4.530±0.393 | 0.433±0.017 | 0.390±0.015 | 0.169±0.007 | 101.07±1.17 | 197.41±2.10 | 327.47±4.44 |
| | HTW | 0.243±0.024 | **0.077±0.025** | 0.103±0.003 | 0.123±0.002 | 3.43±0.83 | 5.08±1.48 | 6.84±2.22 |
| | TabWak | 0.251±0.027 | 0.335±0.029 | **0.085±0.013** | 0.125±0.002 | -14.95±1.08 | -6.16±1.18 | -20.57±0.96 |
| | TabWak$^\top$ | 0.450±0.057 | 0.116±0.020 | 0.096±0.014 | 0.120±0.005 | 109.35±0.91 | 162.44±1.11 | 235.03±1.48 |
| | TimeWak | **0.237±0.017** | 0.212±0.043 | 0.102±0.014 | 0.122±0.002 | **134.83±0.95** | **236.08±1.63** | **340.36±2.06** |
| MuJoCo | W/O | 0.065±0.011 | 0.419±0.084 | 0.032±0.026 | 0.008±0.001 | - | - | - |
| | TR | 1.512±0.179 | 1.153±0.065 | 0.261±0.069 | 0.015±0.003 | 1.38±0.03 | 1.49±0.04 | 1.31±0.04 |
| | GS | 6.548±1.267 | 1.327±0.061 | 0.474±0.006 | 0.014±0.002 | 21.13±0.77 | 10.13±0.62 | 39.63±0.71 |
| | HTW | 0.261±0.067 | 0.493±0.056 | 0.413±0.024 | 0.010±0.002 | 2.89±0.54 | 3.41±0.96 | 4.20±1.37 |
| | TabWak | 0.545±0.122 | 0.975±0.061 | 0.207±0.046 | 0.009±0.001 | 31.30±1.07 | 1.26±0.96 | -3.00±1.04 |
| | TabWak$^\top$ | 0.234±0.032 | **0.463±0.059** | 0.123±0.011 | **0.007±0.001** | -4.85±0.87 | -4.51±0.85 | 3.91±0.88 |
| | TimeWak | **0.089±0.017** | 0.532±0.137 | **0.044±0.021** | 0.008±0.001 | **85.69±1.08** | **56.45±1.26** | **123.36±1.43** |
| Energy | W/O | 0.118±0.021 | 1.245±0.236 | 0.137±0.014 | 0.253±0.000 | - | - | - |
| | TR | 0.649±0.128 | 3.870±0.537 | 0.455±0.017 | 0.337±0.007 | 9.51±0.09 | 17.29±0.11 | 22.58±0.19 |
| | GS | 1.480±0.273 | 3.831±0.272 | 0.494±0.004 | 0.330±0.004 | 51.22±0.88 | 68.67±1.20 | 45.42±1.05 |
| | HTW | **0.099±0.009** | **1.312±0.280** | 0.138±0.019 | **0.253±0.000** | 3.06±0.36 | 4.30±0.68 | 5.42±1.00 |
| | TabWak | 0.179±0.027 | 2.724±0.203 | 0.162±0.011 | 0.255±0.001 | 3.26±0.89 | 3.86±1.02 | 0.57±0.87 |
| | TabWak$^\top$ | 0.213±0.024 | 1.740±0.290 | **0.129±0.013** | 0.265±0.004 | 40.82±0.81 | 46.68±0.86 | 26.00±1.12 |
| | TimeWak | 0.121±0.016 | 1.977±0.750 | 0.142±0.008 | 0.254±0.000 | **231.28±1.45** | **267.53±2.60** | **245.37±2.88** |
| fMRI | W/O | 0.190±0.006 | 1.952±0.087 | 0.132±0.027 | 0.100±0.000 | - | - | - |
| | TR | 2.474±0.341 | 13.312±0.254 | 0.496±0.003 | 0.146±0.004 | 6.49±0.05 | 8.25±0.05 | 9.94±0.04 |
| | GS | 0.714±0.051 | 14.628±0.052 | 0.499±0.001 | 0.108±0.001 | 420.02±1.44 | 321.52±0.59 | 701.90±0.72 |
| | HTW | **0.180±0.011** | **1.900±0.047** | 0.140±0.019 | **0.100±0.000** | 4.32±0.22 | 6.80±0.41 | 9.43±0.61 |
| | TabWak | 0.326±0.042 | 6.825±0.395 | 0.452±0.092 | 0.112±0.000 | 84.02±1.04 | 204.16±0.82 | 47.29±0.83 |
| | TabWak$^\top$ | 0.350±0.014 | 2.191±0.095 | 0.208±0.049 | 0.101±0.000 | **464.67±0.50** | **743.33±0.55** | **1031.96±0.79** |
| | TimeWak | 0.199±0.010 | 2.006±0.053 | **0.122±0.033** | **0.100±0.000** | 379.51±0.82 | 595.68±1.03 | 526.81±13.12 |

This strength is further shown in Figure 3, which plots TPR@0.1%FPR on 64-length sequences as a function of the number of samples. Across all settings, TimeWak consistently outperforms GS and TabWak$^\top$, achieving significantly higher TPR values. Notably, TimeWak reaches a perfect TPR of 1.0 in all cases, requiring only one sample in three settings and two in the remaining one. This high sensitivity makes TimeWak well-suited for real-world use, where only limited samples may be available. Additional results are provided in Appendix E.8.

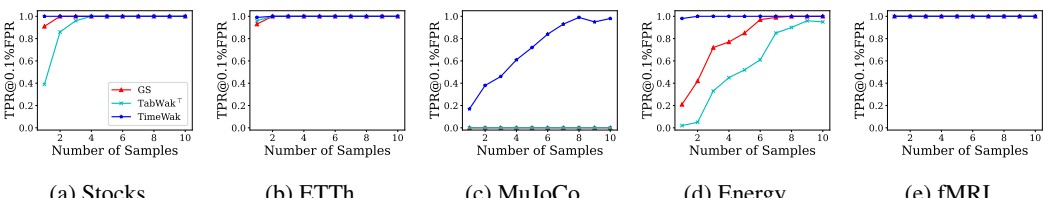

| (a) Stocks | (b) ETTh | (c) MuJoCo | (d) Energy | (e) fMRI |

Figure 3: TPR@0.1%FPR against number of samples across five datasets under 64-length sequences.

## 4.3 Robustness against post-editing attacks

To evaluate the robustness, we first design a set of post-editing attacks. *Offsetting* perturbs the time series by adding a constant offset to each feature based on 5% or 30% of its magnitude, and applied uniformly across all timesteps. *Random cropping* masks out a subregion of the time series by a fixed proportion (5% or 30%), along the rows and columns, similar to the image domain [32]. The *min-max*

Table 2: Results of robustness against post-editing attacks. Average Z-score on 64-length sequences, including un-attacked scores from Table 1. Best results are in **bold**, and second-best are underlined.

| Dataset | Method | Without Attack | Offset ↑ | | Random Crop ↑ | | Min-Max Insertion ↑ | |
|---|---|---|---|---|---|---|---|---|
| | | | 5% | 30% | 5% | 30% | 5% | 30% |
| Stocks | TR | $0.40_{\pm0.08}$ | $0.35_{\pm0.07}$ | $0.17_{\pm0.09}$ | $\underline{57.09}_{\pm1.09}$ | $\mathbf{76.87}_{\pm\mathbf{1.29}}$ | $0.99_{\pm0.10}$ | $5.43_{\pm0.18}$ |
| | GS | $\underline{148.92}_{\pm1.08}$ | $\underline{152.81}_{\pm1.33}$ | $\underline{164.23}_{\pm1.54}$ | $42.53_{\pm1.11}$ | $32.80_{\pm0.78}$ | $\underline{136.73}_{\pm0.90}$ | $\underline{77.20}_{\pm0.86}$ |
| | HTW | $7.34_{\pm0.94}$ | $7.34_{\pm0.94}$ | $7.34_{\pm0.94}$ | $-0.81_{\pm0.42}$ | $-0.92_{\pm0.32}$ | $6.41_{\pm1.16}$ | $2.72_{\pm1.15}$ |
| | TabWak | $16.14_{\pm0.89}$ | $25.74_{\pm1.04}$ | $45.02_{\pm1.40}$ | $40.59_{\pm1.16}$ | $19.71_{\pm1.06}$ | $19.11_{\pm0.75}$ | $30.87_{\pm0.45}$ |
| | TabWak$^\top$ | $88.81_{\pm0.82}$ | $88.84_{\pm0.85}$ | $85.87_{\pm0.86}$ | $30.55_{\pm0.92}$ | $1.71_{\pm0.87}$ | $65.25_{\pm0.79}$ | $15.10_{\pm0.56}$ |
| | TimeWak | $\mathbf{395.34}_{\pm\mathbf{1.24}}$ | $\mathbf{375.96}_{\pm\mathbf{0.96}}$ | $\mathbf{371.04}_{\pm\mathbf{1.02}}$ | $\mathbf{78.20}_{\pm\mathbf{2.23}}$ | $10.40_{\pm1.05}$ | $\mathbf{296.60}_{\pm\mathbf{1.62}}$ | $\mathbf{83.74}_{\pm\mathbf{1.55}}$ |
| ETTh | TR | $7.73_{\pm0.13}$ | $7.96_{\pm0.11}$ | $8.83_{\pm0.12}$ | $27.22_{\pm0.39}$ | $\underline{38.53}_{\pm0.32}$ | $21.86_{\pm0.26}$ | $\underline{49.85}_{\pm0.64}$ |
| | GS | $\underline{197.41}_{\pm2.10}$ | $\underline{186.62}_{\pm2.21}$ | $\underline{159.04}_{\pm2.20}$ | $\mathbf{105.11}_{\pm\mathbf{2.09}}$ | $-39.87_{\pm1.24}$ | $\mathbf{182.03}_{\pm\mathbf{2.09}}$ | $\mathbf{105.23}_{\pm\mathbf{1.21}}$ |
| | HTW | $5.08_{\pm1.48}$ | $5.08_{\pm1.48}$ | $5.08_{\pm1.48}$ | $0.54_{\pm1.33}$ | $-1.99_{\pm0.80}$ | $4.47_{\pm1.45}$ | $2.03_{\pm1.08}$ |
| | TabWak | $-6.16_{\pm1.18}$ | $-2.75_{\pm1.29}$ | $4.92_{\pm1.52}$ | $\underline{84.37}_{\pm2.67}$ | $\mathbf{88.07}_{\pm\mathbf{2.59}}$ | $4.29_{\pm0.94}$ | $36.83_{\pm0.71}$ |
| | TabWak$^\top$ | $162.44_{\pm1.11}$ | $157.75_{\pm1.09}$ | $135.66_{\pm1.23}$ | $70.38_{\pm1.16}$ | $10.79_{\pm1.14}$ | $\underline{99.23}_{\pm0.96}$ | $9.01_{\pm0.80}$ |
| | TimeWak | $\mathbf{236.08}_{\pm\mathbf{1.63}}$ | $\mathbf{243.86}_{\pm\mathbf{1.61}}$ | $\mathbf{207.90}_{\pm\mathbf{1.70}}$ | $75.35_{\pm1.18}$ | $2.47_{\pm1.08}$ | $\underline{171.51}_{\pm1.46}$ | $27.78_{\pm1.22}$ |
| MuJoCo | TR | $1.49_{\pm0.04}$ | $1.46_{\pm0.03}$ | $1.54_{\pm0.03}$ | $7.14_{\pm0.08}$ | $\underline{7.17}_{\pm0.07}$ | $6.09_{\pm0.11}$ | $\mathbf{14.83}_{\pm\mathbf{0.22}}$ |
| | GS | $\underline{10.13}_{\pm0.62}$ | $\underline{10.35}_{\pm0.69}$ | $\underline{8.76}_{\pm0.75}$ | $\underline{28.99}_{\pm0.80}$ | $\mathbf{10.09}_{\pm\mathbf{0.91}}$ | $\underline{9.35}_{\pm0.76}$ | $\underline{10.40}_{\pm1.07}$ |
| | HTW | $3.41_{\pm0.96}$ | $3.41_{\pm0.96}$ | $3.41_{\pm0.96}$ | $1.13_{\pm0.78}$ | $-1.32_{\pm0.67}$ | $3.09_{\pm0.91}$ | $1.67_{\pm0.65}$ |
| | TabWak | $1.26_{\pm0.96}$ | $0.48_{\pm0.89}$ | $5.48_{\pm1.22}$ | $-1.75_{\pm1.21}$ | $-6.65_{\pm0.71}$ | $-0.57_{\pm0.91}$ | $-0.31_{\pm0.59}$ |
| | TabWak$^\top$ | $-4.51_{\pm0.85}$ | $-3.46_{\pm0.99}$ | $-0.24_{\pm0.94}$ | $-32.20_{\pm1.04}$ | $-42.74_{\pm1.25}$ | $-34.16_{\pm0.90}$ | $-80.63_{\pm1.05}$ |
| | TimeWak | $\mathbf{56.45}_{\pm\mathbf{1.26}}$ | $\mathbf{58.90}_{\pm\mathbf{1.36}}$ | $\mathbf{51.53}_{\pm\mathbf{1.31}}$ | $\mathbf{49.30}_{\pm\mathbf{1.20}}$ | $1.85_{\pm1.12}$ | $\mathbf{36.59}_{\pm\mathbf{1.19}}$ | $10.35_{\pm1.18}$ |
| Energy | TR | $17.29_{\pm0.11}$ | $16.74_{\pm0.11}$ | $15.09_{\pm0.09}$ | $\mathbf{128.26}_{\pm\mathbf{0.39}}$ | $\mathbf{148.31}_{\pm\mathbf{0.29}}$ | $37.72_{\pm0.20}$ | $\mathbf{75.09}_{\pm\mathbf{0.29}}$ |
| | GS | $\underline{68.67}_{\pm1.20}$ | $\underline{63.93}_{\pm1.09}$ | $\underline{54.84}_{\pm1.09}$ | $66.30_{\pm0.99}$ | $\underline{96.29}_{\pm0.80}$ | $\underline{66.40}_{\pm1.27}$ | $\underline{53.06}_{\pm1.65}$ |
| | HTW | $4.30_{\pm0.68}$ | $4.30_{\pm0.68}$ | $4.30_{\pm0.68}$ | $0.40_{\pm0.48}$ | $-0.60_{\pm0.34}$ | $3.87_{\pm0.66}$ | $2.03_{\pm0.48}$ |
| | TabWak | $3.86_{\pm1.02}$ | $-5.32_{\pm1.01}$ | $-8.99_{\pm1.03}$ | $-8.31_{\pm0.65}$ | $9.94_{\pm0.60}$ | $4.21_{\pm0.78}$ | $2.67_{\pm0.36}$ |
| | TabWak$^\top$ | $46.68_{\pm0.86}$ | $43.10_{\pm0.84}$ | $42.38_{\pm0.89}$ | $12.26_{\pm1.01}$ | $-49.34_{\pm1.74}$ | $11.87_{\pm0.85}$ | $-43.15_{\pm0.71}$ |
| | TimeWak | $\mathbf{267.53}_{\pm\mathbf{2.60}}$ | $\mathbf{296.74}_{\pm\mathbf{2.49}}$ | $\mathbf{191.63}_{\pm\mathbf{2.13}}$ | $4.91_{\pm0.96}$ | $15.73_{\pm0.96}$ | $\mathbf{195.37}_{\pm\mathbf{1.97}}$ | $34.26_{\pm0.99}$ |
| fMRI | TR | $8.25_{\pm0.05}$ | $8.22_{\pm0.05}$ | $8.10_{\pm0.05}$ | $8.24_{\pm0.05}$ | $7.84_{\pm0.05}$ | $11.45_{\pm0.06}$ | $23.36_{\pm0.10}$ |
| | GS | $321.52_{\pm0.59}$ | $319.93_{\pm0.60}$ | $312.57_{\pm0.62}$ | $\underline{286.05}_{\pm1.13}$ | $\underline{116.24}_{\pm0.87}$ | $\underline{320.52}_{\pm1.66}$ | $\underline{275.34}_{\pm2.20}$ |
| | HTW | $6.80_{\pm0.41}$ | $6.80_{\pm0.41}$ | $6.80_{\pm0.41}$ | $4.82_{\pm0.46}$ | $-0.70_{\pm0.46}$ | $5.95_{\pm0.45}$ | $2.56_{\pm0.41}$ |
| | TabWak | $204.16_{\pm0.82}$ | $205.01_{\pm0.91}$ | $215.64_{\pm1.07}$ | $154.27_{\pm2.15}$ | $\mathbf{452.61}_{\pm\mathbf{3.14}}$ | $248.19_{\pm3.95}$ | $\mathbf{297.10}_{\pm\mathbf{8.67}}$ |
| | TabWak$^\top$ | $\mathbf{743.33}_{\pm\mathbf{0.55}}$ | $\mathbf{743.16}_{\pm\mathbf{0.59}}$ | $\mathbf{742.43}_{\pm\mathbf{0.53}}$ | $\mathbf{636.28}_{\pm\mathbf{0.67}}$ | $\underline{317.24}_{\pm1.18}$ | $\mathbf{614.25}_{\pm\mathbf{0.77}}$ | $224.27_{\pm0.85}$ |
| | TimeWak | $\underline{595.68}_{\pm1.03}$ | $\underline{601.68}_{\pm0.81}$ | $\underline{601.53}_{\pm0.96}$ | $\underline{459.66}_{\pm1.00}$ | $112.68_{\pm0.85}$ | $\underline{498.39}_{\pm0.84}$ | $\underline{189.43}_{\pm1.00}$ |

*insertion* attack perturbs the series by randomly replacing a proportion of points (5% or 30%) in each feature with random values drawn uniformly between the feature's minimum and maximum values.

Table 2 presents the Z-scores of 64-length watermarked synthetic time series data under these attacks, and averaged over 100 trials. Random cropping at 30% proves especially challenging, with several methods showing negative Z-scores. Nevertheless, `TimeWak` demonstrates the best overall robustness, consistently outperforming all baselines across most attack scenarios while maintaining high generation quality and accurate watermark detection. In contrast, although HTW has better quality, it does poorly under attacks, indicating a struggle in balancing trade-offs between quality and robustness.

Interestingly, TR's detection scores further improve under certain post-processing attacks. In particular, significant gains are observed under random cropping and min-max insertion, likely due to the inherent robustness of watermarking in the Fourier domain. However, its overall performance lags behind `TimeWak`. Both TabWak and TabWak$^\top$ show significant degradation under attacks, particularly on MuJoCo and Energy datasets, where detection frequently fails. GS overall demonstrates strong robustness, maintaining detectability under all attacks except for 30% cropping on the ETTh dataset. However, it produces low-quality synthetic samples, highlighting the need for a time series specific watermark that can navigate the trade-offs between generation quality and watermark robustness.

## 5  Conclusion

Motivated by the need to ensure the traceability of synthetic time series, we propose `TimeWak`, the first watermarking algorithm for multivariate time series diffusion models. `TimeWak` embeds seeds through a temporally chained-hash and feature-wise shuffling in data space, preserving the temporal and feature dependencies and enhancing the watermark detectability. To address non-uniform error distribution in the time series diffusion process, we optimize `TimeWak` for $\epsilon$-exact inversion and provide the bounded error analysis. Compared to multiple SOTA watermarking algorithms, `TimeWak`

balances synthetic data quality, watermark detectability, and robustness against post-editing attacks. Extensive evaluations on five datasets show that `TimeWak` improves context-FID score by 61.96% and correlational scores by 8.44%, against the strongest SOTA baseline, while maintaining strong detectability.

**Limitations.** While `TimeWak` does not natively support streaming data, it can be applied to streaming scenarios by processing data in small, fixed-length windows and watermarking each window independently. However, finer-grained cases, such as watermarking at the per-timestep level, remain unexplored. This represents a key limitation of `TimeWak` and an important direction for future work.

## Acknowledgments

This research is partly funded by Priv-GSyn project, 200021E_229204 of Swiss National Science Foundation, the DEPMAT project, P20-22 / N21022, of the research programme Perspectief of the Dutch Research Council, and ASM International NV.

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

# A Nomenclature

| | |
|---|---|
| $\alpha_t$ | Noise schedule parameter |
| $\beta_t$ | Variance schedule for noise addition |
| $\Delta_t$ | Lipschitz constant for noise estimator sensitivity at time step $t$ |
| $\epsilon$ | Perturbation error in inversion |
| $\gamma$ | Scaling factor in BDIA inversion |
| $\hat{\boldsymbol{\epsilon}}_{\boldsymbol{\theta}}$ | Noise estimator function |
| $\kappa$ | Cryptographic key for watermarking |
| $\mathbb{I}[\cdot]$ | Indicator function |
| $\mathbf{s}^{[:],f}$ | Watermark seed across all timesteps at feature $f$ |
| $\mathbf{s}^{w,[:]}$ | Watermark seed across all features at timestep $w$ |
| $\mathbf{s}^{w,f}$ | Watermark seed at timestep $w$ and feature $f$ |
| $\mathbf{x}_0$ | Generated sequence of length $W$ and $F$ features |
| $\mathbf{x}_T$ | Noised sequence at timestep $T$ |
| $\mathbf{x}_t^{approx}$ | Approximate diffusion state at step $t$ |
| $\mathbf{x}_t$ | Diffusion state at step $t$ |
| $\mathcal{H}(\kappa, w, \mathbf{s}^{w-1,[:]})$ | Temporal chained hashing function |
| $\mathcal{N}(0, I)$ | Standard normal distribution |
| $\mathcal{U}(0, 1)$ | Continuous uniform distribution between 0 and 1 |
| $\mathcal{U}(\{0, L-1\})^F$ | Vector in $\mathbb{R}^F$ with i.i.d. discrete uniform components over $\{0, \ldots, L-1\}$ |
| $\mathcal{W}^*$ | Set of valid timesteps for comparison |
| $\Phi$ | CDF of standard normal distribution |
| $\Phi^{-1}$ | Inverse CDF of standard normal distribution |
| $\pi_\kappa$ | Feature permutation function |
| $\sigma_t$ | Standard deviation at timestep $t$ |
| $a_t, b_t$ | BDIA parameters for inversion process |
| $F$ | Number of features |
| $H$ | Interval length for watermark partitioning |
| $L$ | Bit length |
| $n$ | Number of intervals in time window |
| $T$ | Total diffusion steps |
| $W$ | Time window length |

# B Diffusion and diffusion inversion

## B.1 Time series diffusion model

This section covers the necessary background knowledge for time series diffusion models.

Denoising Diffusion Probabilistic Models (DDPMs) are a powerful generative models, especially for time series synthesis [10, 1, 12, 34]. As shown in Figure 4, they work by iteratively forward noising data and then learning to invert this process through during the backward step [10]. For a time series window, one step of forward noise is given by:

$$\mathbf{x}_t = \sqrt{1 - \beta_t}\mathbf{x}_{t-1} + \sqrt{\beta_t}\epsilon_t. \tag{16}$$

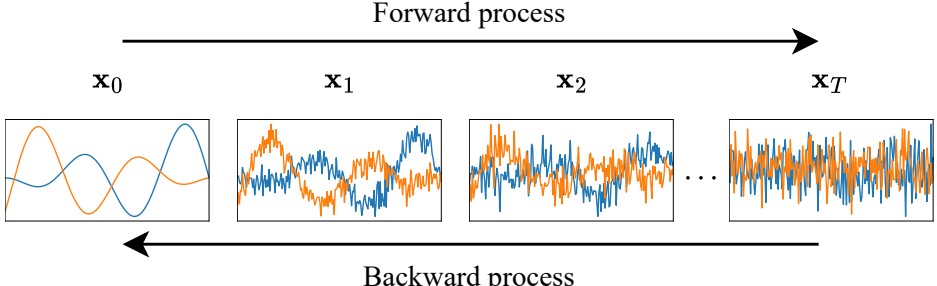

Figure 4: Forward and backward diffusion process. $\mathbf{x}_0$ denotes the initial signal window and $\mathbf{x}_T$ corresponds to the fully diffused version of the signal obtained after $T$ forward diffusion steps.

In this formulation, $\mathbf{x}_t$ represents a multivariate time series signal after undergoing $t$ steps of noise addition. The term $\beta_t$ denotes the noise variance at step $t$. The noise component $\epsilon_t$ is sampled from a normal distribution, $\mathcal{N}(0, I)$. Using this notation, $\mathbf{x}_0$ and $\mathbf{x}_T$ define a noise-free sequence and a fully noised sequence, respectively, for an arbitrary window slice, where $T$ is the total number of noise steps. Without iteratively applying Equation (16), an intermediate noising step can be efficiently computed using a *reparameterization trick* [10]:

$$\mathbf{x}_t = \sqrt{\bar{\alpha}_t}\mathbf{x}_0 + \sqrt{1 - \bar{\alpha}_t}\epsilon, \tag{17}$$

where $\bar{\alpha}_t$ is the cumulative product of noise reduction factors up to step $t$, expressed as $\bar{\alpha}_t = \prod_{s=1}^{t} \alpha_s = \prod_{s=1}^{t}(1 - \beta_s)$, with $\beta_s$ being the noise variance at timestep $s$ and $\epsilon \sim \mathcal{N}(0, I)$.

The denoising process reverses the noising steps to reconstruct $\mathbf{x}_0$ from $\mathbf{x}_t$, ideally restoring the original signal. This is achieved by training a neural network, $\epsilon_{\boldsymbol{\theta}}$, to estimate the noise component at each step. A single reverse step is given by:

$$\tilde{\mathbf{x}}_{t-1} = \frac{1}{\sqrt{\alpha_t}} \left( \tilde{\mathbf{x}}_t - \frac{\beta_t}{\sqrt{1 - \bar{\alpha}_t}} \epsilon_{\boldsymbol{\theta}}(\tilde{\mathbf{x}}_t, t) \right) + \sigma_t z. \tag{18}$$

Here, $\tilde{\mathbf{x}}_t$ and $\tilde{\mathbf{x}}_{t-1}$ denote the signal estimates at time steps $t$ and $t - 1$, respectively, where $\tilde{\mathbf{x}}_j = \mathbf{x}_j$ at the final noising step. The term $\sigma_t$ is typically a function of $\beta$ to introduce stochasticity in the sampling process, and $z \sim \mathcal{N}(0, I)$ [10]. The objective is to iteratively refine the denoised sample so that $\tilde{x}_0 \approx x_0$.

## B.2 Diffusion models for time series

Diffusion models show great promise in time series tasks, excelling in both forecasting [20, 1, 14] and generation [16, 37]. Among these, Denoising Diffusion Probabilistic Models (DDPMs) [10] are a leading framework, which progressively denoise samples to reconstruct data from noise [17] and rely on stochastic noise addition during sampling [10]. On the other hand, Denoising Diffusion Implicit Models (DDIMs) use a deterministic sampling process that removes noise, enabling faster sampling and fewer steps to generate high-quality samples with greater predictability [23]. However, this reduces the model's ability to explore a wide range of outputs, leading to lower diversity and reduced robustness (i.e., consistency in generating diverse and reliable samples) [10, 23].

Diffusion-TS, which serves as the backbone model of `TimeWak` presented in this paper, employs a DDPM combined with seasonal-trend decomposition to better capture underlying structures and dependencies of multivariate time series [34]. It also introduces a Fourier-based loss to optimize reconstruction, improving accuracy by better matching the frequency components [34]. Its innovation lies in the integration of seasonal-trend decomposition with DDPMs, and the use of the Fourier-based loss to enhance the model's ability to capture complex temporal patterns.

Some diffusion models that synthesize time series data include ScoreGrad [31], SSSD$^{S4}$ [1], TSD-iff [12], but some also incorporate transformer-based elements like TimeGrad [20], CSDI [26], and TDSTF [4]. Diffusion-TS [34] effectively addresses weaknesses in these models. Unlike ScoreGrad and TimeGrad which use autoregressive models [31, 20], it avoids error accumulation and slow inference over long horizons by using DDPM, a stable diffusion-based framework. It outperforms SSSD$^{S4}$ and TSDiff [1, 12] by replacing resource-intensive S4 layers with an efficient latent layer

that simplifies handling multivariate data. Diffusion-TS handles incomplete datasets better than CSDI [26] by avoiding the need for explicit pairing of observed and missing data during training, while being more adaptable and efficient on diverse datasets compared to TDSTF [4], which struggles with real-time forecasting.

### B.3 DDIM and DDIM inversion

The Denoising Diffusion Implicit Model (DDIM) [23] offers deterministic diffusion and sampling, extending the traditional Markovian diffusion process to a broader class of non-Markovian processes. Given a initial noise vector $\mathbf{x}_T$ and a neural network $\boldsymbol{\epsilon_\theta}$ that predicts the noise $\boldsymbol{\epsilon_\theta}(\mathbf{x}_t, t)$ at each timestep $t$, the DDIM sampling step to generate sample $\mathbf{x}_{t-1}$ from $\mathbf{x}_t$, is defined as:

$$\mathbf{x}_{t-1} = \sqrt{\alpha_{t-1}} \left( \frac{\mathbf{x}_t - \sqrt{1 - \alpha_t} \boldsymbol{\epsilon_\theta}(\mathbf{x}_t, t)}{\sqrt{\alpha_t}} \right) + \sqrt{1 - \alpha_{t-1} - \sigma_t^2} \cdot \boldsymbol{\epsilon_\theta}(\mathbf{x}_t, t) + \sigma_t \epsilon_t, \quad (19)$$

where $\alpha_1, \ldots, \alpha_T$ are computed from a predefined variance schedule, $\epsilon_t \sim \mathcal{N}(0, I)$ denotes standard Gaussian noise independent of $\mathbf{x}_t$, and the $\sigma_t$ values can be varied to yield different generative processes. Setting $\sigma_t$ to 0 for all $t$ makes the sampling process deterministic:

$$\mathbf{x}_{t-1} = \sqrt{\frac{\alpha_{t-1}}{\alpha_t}} \mathbf{x}_t + \left( \sqrt{1 - \alpha_{t-1}} - \sqrt{\frac{\alpha_{t-1}}{\alpha_t} - \alpha_{t-1}} \right) \boldsymbol{\epsilon_\theta}(\mathbf{x}_t, t). \quad (20)$$

This sampling process ensures the same latent matrix $\mathbf{x}_0$ is consistently generated by a given noise matrix $\mathbf{x}_T$.

Having large $T$ values (being limited with small steps) allows to cross the timesteps in the backward direction toward increasing noise levels, which gives out a deterministic diffusion process from $\mathbf{x}_0$ to $\mathbf{x}_T$; this is also known as DDIM inversion:

$$\mathbf{x}_{t+1} \approx \sqrt{\frac{\alpha_{t+1}}{\alpha_t}} \mathbf{x}_t + \left( \sqrt{1 - \alpha_{t+1}} - \sqrt{\frac{\alpha_{t+1}}{\alpha_t} - \alpha_{t+1}} \right) \boldsymbol{\epsilon_\theta}(\mathbf{x}_t, t). \quad (21)$$

## C Proof

**Theorem 3.1.** *Let $\{\mathbf{x}_t\}_{t=0}^T$ be the sequence of diffusion states governed by the BDIA-DDIM recurrence for a given dataset, following Equation (7). Given the noise estimator $\hat{\epsilon}_\theta$ follows Assumption 1. Suppose that instead of the exact terminal state $\mathbf{x}_1$, an approximation of $\mathbf{x}_1$, termed as $\mathbf{x}_1^{approx}$, is used with a small perturbation $\epsilon$, given by:*

$$\mathbf{x}_1^{approx} = \mathbf{x}_2 = \mathbf{x}_1^{orig} + \epsilon. \quad (8)$$

*Let the propagated error at time $t$ be defined as,*

$$\delta_t = \|\mathbf{x}_t^{approx} - \mathbf{x}_t^{orig}\|. \quad (9)$$

*Then, for $t \geq 1$, the error is bounded by,*

$$\|\delta_T\| \leq |\epsilon| \prod_{t=1}^{T-1} \left( \left| \frac{1}{\gamma} - \frac{a_t}{\gamma} + \frac{1}{a_{t+1}} \right| + \frac{b_t}{\gamma} \Delta_t + \frac{b_{t+1}}{a_{t+1}} \Delta_t \right), \quad (10)$$

*where $\Delta_t$ quantifies the sensitivity of the noise estimator at timestep $t$.*

**Proof:**

For $t = 1$, we have:

$$\boldsymbol{\delta}_1 = \mathbf{x}_1^{approx} - \mathbf{x}_1^{orig} = \boldsymbol{\epsilon}. \quad (22)$$

For $t = 2$, using the recurrence,

$$\mathbf{x}_2 = \frac{\mathbf{x}_0}{\gamma} - \frac{a_1 \mathbf{x}_1 + b_1 \hat{\epsilon}_\theta(\mathbf{x}_1, 1)}{\gamma} + \left( \frac{\mathbf{x}_1}{a_2} - \frac{b_2}{a_2} \hat{\epsilon}_\theta(\mathbf{x}_1, 2) \right). \quad (23)$$

Subtracting the approximated and original cases and using Assumption 1, we obtain:

$$\boldsymbol{\delta}_2 = -\frac{a_1 \boldsymbol{\delta}_1}{\gamma} + \frac{\boldsymbol{\delta}_1}{a_2} - \frac{b_1}{\gamma} \Delta_1 \boldsymbol{\delta}_1 - \frac{b_2}{a_2} \Delta_1 \boldsymbol{\delta}_1. \quad (24)$$

Thus, defining

$$C_2 = \left| \frac{1}{a_2} - \frac{a_1}{\gamma} \right| + \frac{b_1}{\gamma} \Delta_1 + \frac{b_2}{a_2} \Delta_1, \tag{25}$$

we bound

$$\|\boldsymbol{\delta}_2\| \leq C_2 \|\boldsymbol{\delta}_1\| = C_2 \|\boldsymbol{\epsilon}\|. \tag{26}$$

Suppose for some $t \geq 2$, there exists a constant $C_t$ such that

$$\|\boldsymbol{\delta}_t\| \leq C_t \|\boldsymbol{\epsilon}\|. \tag{27}$$

For $t + 1$, using the recurrence:

$$\mathbf{x}_{t+1} = \frac{\mathbf{x}_{t-1}}{\gamma} - \frac{a_t \mathbf{x}_t + b_t \hat{\boldsymbol{\epsilon}}_{\boldsymbol{\theta}}(\mathbf{x}_t, t)}{\gamma} + \left( \frac{\mathbf{x}_t}{a_{t+1}} - \frac{b_{t+1}}{a_{t+1}} \hat{\boldsymbol{\epsilon}}_{\boldsymbol{\theta}}(\mathbf{x}_t, t+1) \right). \tag{28}$$

Taking differences, we obtain

$$\boldsymbol{\delta}_{t+1} = \frac{\boldsymbol{\delta}_{t-1}}{\gamma} - \frac{a_t \boldsymbol{\delta}_t}{\gamma} - \frac{b_t}{\gamma} \Delta_t \boldsymbol{\delta}_t + \frac{\boldsymbol{\delta}_t}{a_{t+1}} - \frac{b_{t+1}}{a_{t+1}} \Delta_t \boldsymbol{\delta}_t. \tag{29}$$

Bounding the terms, we define:

$$C_{t+1} = C_t \left( \left| \frac{1}{\gamma} - \frac{a_t}{\gamma} + \frac{1}{a_{t+1}} \right| + \frac{b_t}{\gamma} \Delta_t + \frac{b_{t+1}}{a_{t+1}} \Delta_t \right). \tag{30}$$

Thus, we conclude:

$$\|\boldsymbol{\delta}_{t+1}\| \leq C_{t+1} \|\boldsymbol{\epsilon}\|. \tag{31}$$

By induction, we obtain:

$$C_T = \prod_{t=1}^{T-1} \left( \left| \frac{1}{\gamma} - \frac{a_i}{\gamma} + \frac{1}{a_{t+1}} \right| + \frac{b_t}{\gamma} \Delta_t + \frac{b_{t+1}}{a_{t+1}} \Delta_t \right). \tag{32}$$

Thus, the perturbation remains bounded for all $T$, i.e.

$$\|\delta_T\| \leq |\epsilon| \prod_{t=1}^{T-1} \left( \left| \frac{1}{\gamma} - \frac{a_t}{\gamma} + \frac{1}{a_{t+1}} \right| + \frac{b_t}{\gamma} \Delta_t + \frac{b_{t+1}}{a_{t+1}} \Delta_t \right), \tag{33}$$

completing the proof. $\square$

To further validate Assumption 1, we empirically computed $\Delta_t$ across four different datasets, using 10,000 samples from each. Specifically, $\Delta_t$ is calculated as the maximum ratio for different $t$ using the $L_1$ norm of the data samples, as illustrated in Figure 5.

We also generate 10,000 samples with 64-length sequences across five datasets, computing the final output $\mathbf{x}_0$ and the one step prior $\mathbf{x}_1$. Results in Table 3 show consistently small $L_1$ norm between $\mathbf{x}_1$ and $\mathbf{x}_0$, with an average of $5.1 \times 10^{-3}$ to $7.0 \times 10^{-3}$ and maximum of $< 0.23$, validating our $\epsilon$-exact approximation. These empirically small errors confirm Theorem 3.1's theoretical bounds and demonstrate reliable watermark detection despite the approximation.

Table 3: $L_1$ norm between $\mathbf{x}_1$ and $\mathbf{x}_0$ for 64-length sequences over 10,000 samples.

| Dataset | Avg. $L_1$ $\left( \times 10^{-3} \right)$ | Max. $L_1$ |
|---|---|---|
| Stocks | 7.031 | 0.082 |
| ETTh | 6.776 | 0.104 |
| MuJoCo | 5.146 | 0.081 |
| Energy | 5.687 | 0.230 |
| fMRI | 5.945 | 0.070 |

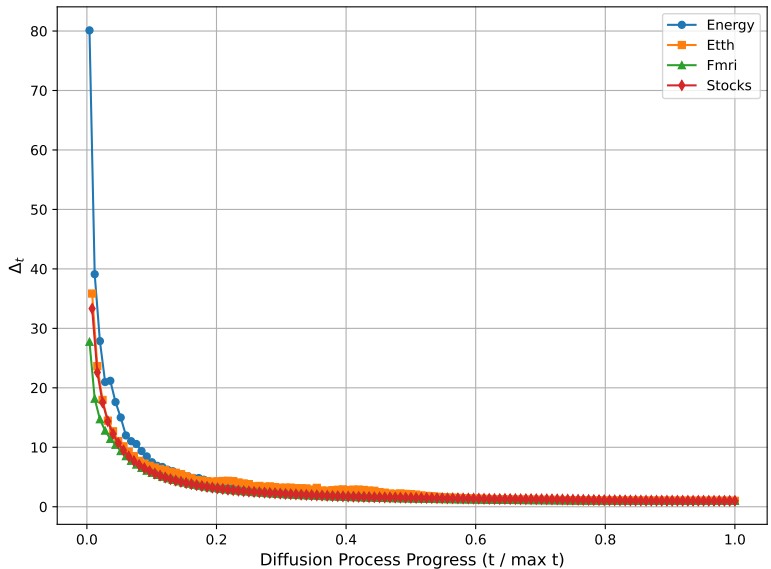

Figure 5: $\Delta_t$ for different datasets.

# D Experiment details

## D.1 Datasets

Table 4 shows the details of all datasets used in the experiments.

Table 4: Details of datasets used in experiments.

| Dataset | Number of Rows | Number of Features | Source |
|---|---|---|---|
| Stocks | 3,773 | 6 | https://finance.yahoo.com/quote/GOOG |
| ETTh | 17,420 | 7 | https://github.com/zhouhaoyi/ETDataset |
| MuJoCo | 10,000 | 14 | https://github.com/deepmind/dm_control |
| Energy | 19,711 | 28 | https://archive.ics.uci.edu/ml/datasets |
| fMRI | 10,000 | 50 | https://www.fmrib.ox.ac.uk/datasets |

## D.2 Evaluation metrics

**Context-FID** Jeha et al. [11] introduced the Context-FID score, which is a refined adaptation of the Fréchet Inception Distance (FID) used for evaluating the similarity between real and synthetic time series distributions. Unlike traditional FID, which relies on the Inception model as a feature extractor for images, Context-FID uses TS2Vec [35], which is a specialized time series embedding model. Yue et al. [35] demonstrated that models with lower Context-FID scores tend to perform well in downstream tasks, revealing a strong correlation between Context-FID and the forecasting performance of generative models. Ultimately, a lower Context-FID score signifies a closer resemblance between real and synthetic distributions.

**Correlational** We calculates the covariance between the $i^{th}$ and $j^{th}$ features of a time series using the following equation [15]:

$$\text{Cov}_{i,j} = \frac{1}{W} \sum_{t=1}^{W} \text{K}_i^t \text{K}_i^t - \left( \frac{1}{W} \sum_{t=1}^{W} \text{K}_i^t \right) \left( \frac{1}{W} \sum_{t=1}^{W} \text{K}_j^t \right). \tag{34}$$

where $W$ is the total number of time steps, $\text{K}_i^t$ and $\text{K}_j^t$ are the values of the $i^{th}$ and $j^{th}$ features at time step $t$, and the summations compute the average product of these values, subtracting the product

of their individual means. To assess the correlation between real and synthetic time series, we use the following metric [34]:

$$\frac{1}{10} \sum_{i,j}^{d} \left| \frac{\text{Cov}_{i,j}^{R}}{\sqrt{\text{Cov}_{i,i}^{R} \text{Cov}_{j,j}^{R}}} - \frac{\text{Cov}_{i,j}^{S}}{\sqrt{\text{Cov}_{i,i}^{S} \text{Cov}_{j,j}^{S}}} \right|, \tag{35}$$

where Cov is the covariance between its subscripts ($\langle i, i \rangle$, $\langle j, j \rangle$, $\langle i, j \rangle$), where $i$ and $j$ are the features in the real (denoted by $R$) and synthetic (denoted by $S$) time series data, and $d$ is the total number of features, with the summation taken over all feature pairs.

**Discriminative**   The discriminative score is computed as $|\text{accuracy} - 0.5|$, quantifying the model's ability to distinguish between real and synthetic time series. A lower score indicates better performance, as it indicates greater difficulty in differentiation, implying a higher degree of similarity between the two distributions. To ensure consistency, we follow the experimental setup of TimeGAN [33] by using a two-layer GRU-based neural network as the classifier.

**Predictive**   The predictive score is evaluated using the mean absolute error (MAE) between the predicted and actual values on the test data. Again, we use the experimental setup of TimeGAN [33] by using a two-layer GRU-based neural network for sequence prediction.

**Z-score**   The Z-score is a statistical measure used to assess watermark detectability by quantifying the deviation between watermarked and non-watermarked samples. It facilitates hypothesis testing, where the null hypothesis $H_0$ states that a given sample is not watermarked by the corresponding watermarking method. A sufficiently high positive Z-score provides evidence against $H_0$, suggesting the presence of a watermark.

For sample-wise bit accuracy in time series, the Z-score is computed as $Z = \frac{\mu_{\text{Acc, W}} - \mu_{\text{Acc, NW}}}{\sigma_{\text{Acc, NW}} / \sqrt{n}}$, where $\mu_{\text{Acc, W}}$ and $\mu_{\text{Acc, NW}}$ are the mean bit accuracy of watermarked and non-watermarked samples, respectively, $\sigma_{\text{Acc, NW}}$ is the standard deviation of bit accuracy in the non-watermarked samples, and $n$ is the number of watermarked samples. Under $H_0$, the expected difference in means is negligible, resulting in a Z-score close to zero. A large positive Z-score provides statistical evidence for the presence of a watermark.

For the Tree-Ring watermarking method, the Z-score is computed in the Fourier domain instead of the sample-wise bit accuracy domain. It measures the deviation in the amplitude spectrum between watermarked and non-watermarked samples, given by $Z = \frac{\mu_{\mathcal{F}_{\text{NW}}} - \mu_{\mathcal{F}_{\text{W}}}}{\sigma_{\mathcal{F}_{\text{NW}}}}$ where $\mathcal{F}_{\text{W}}$ and $\mathcal{F}_{\text{NW}}$ denote the Fourier amplitude spectrum of watermarked and non-watermarked samples, respectively, with $\mu$ and $\sigma$ representing their mean and standard deviation. Since this method does not rely on per-sample bit accuracy, the test statistic is independent of $n$, and the computation utilizes the opposite tail of the distribution.

**TPR@X%FPR**   This metric measures the True Positive Rate (TPR) at a specified False Positive Rate (FPR) of X%, where X% denotes a fixed false positive threshold. It reflects the effectiveness of watermark detection by quantifying how reliably watermarked samples are identified under controlled false positive conditions. A higher TPR@X%FPR indicates stronger detection performance and greater robustness of the watermarking method.

### D.3   Baselines

**Tree-Ring**   We adapt the latent-representation-based Tree-Ring watermark for multivariate time series in the data space. Initially, the Tree-Ring watermark was proposed for images with square dimensions as it places the circular ring watermark pattern centrally. However, multivariate time series often have rectangular dimensions with varying numbers of features and timesteps. To accommodate this structure, we apply a flexible ring pattern with a predefined radius indicating the outermost circle of the watermark. Finally, we embed and detect the watermark in the Fourier domain.

**Gaussian Shading**   Similar to Tree-Ring, we implement Gaussian Shading from the image domain to multivariate time series by embedding the watermark directly in the data space. To maintain

coherence and efficiency, we use a single control seed across all samples to avoid the need for additional indices for each sample.

**Heads Tails Watermark**   The HTW is a post-watermarking technique designed for univariate time series. To extend its applicability, we adapt it for multivariate time series by iterating through each variate in the generated synthetic sample. During evaluation, the watermarked time series is reversed, and processing each series independently. In short, we treat each variate as a single series.

**TabWak**   TabWak was originally designed for the tabular data domain, where it operates effectively in the latent space. We extend its application to multivariate time series in the data space. However, tabular data primarily captures feature dependencies, while time series data inherently relies on both feature and temporal dependencies. Therefore, we implement another version called TabWak$^{\top}$ by transposing the watermarking direction onto the temporal axis. Similarly, we evaluate the watermark detectability on a sample by sample basis.

### D.4   Training and sampling

All code implementations are done in PyTorch (version 2.3.1) using a single NVIDIA GeForce RTX 2080 Graphics Card coupled with an Intel(R) Xeon(R) Platinum 8562Y+ CPU for all experiments. Dataset splits are 80% for training and 20% for testing. Table 5 shows training and sampling time for all datasets across window of sizes 24, 64 and 128. We train the time series diffusion model following the Diffusion-TS settings [34], and generate 10,000 watermarked synthetic samples using `TimeWak` for each sampling run.

Table 5: Details of training and sampling time.

| Dataset | Window Size | Training Time ($\sim$ min.) | Sampling Time ($\sim$ min.) |
|---------|-------------|-----------------------------|-----------------------------|
| Stocks  | 24          | 6.1                         | 0.4                         |
|         | 64          | 6.2                         | 2.1                         |
|         | 128         | 8.0                         | 4.6                         |
| ETTh    | 24          | 11.6                        | 0.4                         |
|         | 64          | 12.7                        | 2.4                         |
|         | 128         | 13.7                        | 5.2                         |
| MuJoCo  | 24          | 9.7                         | 0.7                         |
|         | 64          | 9.9                         | 4.9                         |
|         | 128         | 10.6                        | 12.6                        |
| Energy  | 24          | 23.5                        | 1.4                         |
|         | 64          | 24.2                        | 10.5                        |
|         | 128         | 24.3                        | 23.0                        |
| fMRI    | 24          | 16.8                        | 1.8                         |
|         | 64          | 17.5                        | 13.5                        |
|         | 128         | 18.1                        | 28.5                        |

# E Additional experimental results

## E.1 Quality performance

Table 6 shows the quality of synthetic time series generated in 64 and 128 window sizes. `TimeWak` remains stable across all datasets and even comparable to the quality of non-watermarked samples.

Table 6: Results of synthetic time series quality. No watermarking ('W/O') is included.

| Dataset | Method | 64-length ↓ | | | | 128-length ↓ | | | |
|---|---|---|---|---|---|---|---|---|---|
| | | Context-FID | Correlational | Discriminative | Predictive | Context-FID | Correlational | Discriminative | Predictive |
| Stocks | W/O | $0.444_{\pm0.114}$ | $0.030_{\pm0.023}$ | $0.104_{\pm0.014}$ | $0.037_{\pm0.000}$ | $0.536_{\pm0.135}$ | $0.020_{\pm0.015}$ | $0.148_{\pm0.019}$ | $0.037_{\pm0.000}$ |
| | TR | $1.812_{\pm0.210}$ | $0.102_{\pm0.012}$ | $0.206_{\pm0.038}$ | $0.038_{\pm0.001}$ | $3.555_{\pm0.868}$ | $0.130_{\pm0.025}$ | $0.276_{\pm0.030}$ | $0.041_{\pm0.004}$ |
| | GS | $1.643_{\pm0.126}$ | $0.011_{\pm0.005}$ | $0.252_{\pm0.072}$ | $0.042_{\pm0.001}$ | $2.647_{\pm0.262}$ | $0.025_{\pm0.015}$ | $0.186_{\pm0.071}$ | $0.040_{\pm0.000}$ |
| | HTW | $0.426_{\pm0.073}$ | $0.026_{\pm0.016}$ | $0.105_{\pm0.032}$ | $0.037_{\pm0.000}$ | $0.622_{\pm0.092}$ | $0.017_{\pm0.013}$ | $0.152_{\pm0.086}$ | $0.037_{\pm0.000}$ |
| | TabWak | $0.242_{\pm0.045}$ | $0.011_{\pm0.008}$ | $0.141_{\pm0.021}$ | $0.037_{\pm0.000}$ | $0.394_{\pm0.034}$ | $0.009_{\pm0.009}$ | $0.150_{\pm0.031}$ | $0.037_{\pm0.000}$ |
| | TabWak$^\top$ | $0.284_{\pm0.091}$ | $0.005_{\pm0.005}$ | $0.099_{\pm0.025}$ | $0.037_{\pm0.000}$ | $0.367_{\pm0.035}$ | $0.010_{\pm0.007}$ | $0.129_{\pm0.054}$ | $0.037_{\pm0.000}$ |
| | TimeWak | $0.387_{\pm0.054}$ | $0.017_{\pm0.017}$ | $0.092_{\pm0.041}$ | $0.037_{\pm0.000}$ | $0.316_{\pm0.044}$ | $0.021_{\pm0.024}$ | $0.140_{\pm0.029}$ | $0.037_{\pm0.000}$ |
| ETTh | W/O | $0.384_{\pm0.034}$ | $0.070_{\pm0.011}$ | $0.106_{\pm0.009}$ | $0.115_{\pm0.006}$ | $1.086_{\pm0.070}$ | $0.098_{\pm0.021}$ | $0.166_{\pm0.013}$ | $0.111_{\pm0.008}$ |
| | TR | $2.224_{\pm0.209}$ | $0.217_{\pm0.011}$ | $0.284_{\pm0.032}$ | $0.131_{\pm0.004}$ | $2.450_{\pm0.128}$ | $0.270_{\pm0.017}$ | $0.284_{\pm0.063}$ | $0.138_{\pm0.005}$ |
| | GS | $3.398_{\pm0.384}$ | $0.248_{\pm0.025}$ | $0.356_{\pm0.029}$ | $0.154_{\pm0.001}$ | $4.998_{\pm0.603}$ | $0.233_{\pm0.027}$ | $0.381_{\pm0.042}$ | $0.162_{\pm0.010}$ |
| | HTW | $0.372_{\pm0.027}$ | $0.076_{\pm0.006}$ | $0.118_{\pm0.024}$ | $0.116_{\pm0.006}$ | $1.119_{\pm0.065}$ | $0.091_{\pm0.014}$ | $0.165_{\pm0.010}$ | $0.120_{\pm0.006}$ |
| | TabWak | $0.597_{\pm0.044}$ | $0.346_{\pm0.027}$ | $0.176_{\pm0.020}$ | $0.122_{\pm0.007}$ | $1.931_{\pm0.254}$ | $0.466_{\pm0.058}$ | $0.248_{\pm0.017}$ | $0.128_{\pm0.005}$ |
| | TabWak$^\top$ | $0.503_{\pm0.045}$ | $0.095_{\pm0.043}$ | $0.119_{\pm0.012}$ | $0.120_{\pm0.010}$ | $1.477_{\pm0.075}$ | $0.129_{\pm0.037}$ | $0.143_{\pm0.014}$ | $0.112_{\pm0.011}$ |
| | TimeWak | $0.297_{\pm0.038}$ | $0.133_{\pm0.040}$ | $0.097_{\pm0.015}$ | $0.115_{\pm0.003}$ | $1.090_{\pm0.100}$ | $0.135_{\pm0.057}$ | $0.174_{\pm0.007}$ | $0.110_{\pm0.009}$ |
| MuJoCo | W/O | $0.103_{\pm0.016}$ | $0.341_{\pm0.025}$ | $0.023_{\pm0.015}$ | $0.007_{\pm0.000}$ | $0.179_{\pm0.011}$ | $0.290_{\pm0.025}$ | $0.055_{\pm0.027}$ | $0.005_{\pm0.001}$ |
| | TR | $2.627_{\pm0.230}$ | $0.948_{\pm0.064}$ | $0.270_{\pm0.128}$ | $0.016_{\pm0.005}$ | $2.348_{\pm0.236}$ | $0.865_{\pm0.025}$ | $0.358_{\pm0.008}$ | $0.008_{\pm0.001}$ |
| | GS | $7.162_{\pm1.301}$ | $1.034_{\pm0.087}$ | $0.447_{\pm0.011}$ | $0.011_{\pm0.002}$ | $4.797_{\pm1.290}$ | $1.335_{\pm0.041}$ | $0.454_{\pm0.023}$ | $0.007_{\pm0.001}$ |
| | HTW | $0.316_{\pm0.034}$ | $0.334_{\pm0.044}$ | $0.248_{\pm0.079}$ | $0.011_{\pm0.001}$ | $0.431_{\pm0.051}$ | $0.309_{\pm0.064}$ | $0.255_{\pm0.174}$ | $0.009_{\pm0.001}$ |
| | TabWak | $0.372_{\pm0.064}$ | $0.671_{\pm0.083}$ | $0.137_{\pm0.027}$ | $0.007_{\pm0.001}$ | $0.369_{\pm0.073}$ | $0.589_{\pm0.059}$ | $0.175_{\pm0.042}$ | $0.006_{\pm0.002}$ |
| | TabWak$^\top$ | $0.238_{\pm0.019}$ | $0.339_{\pm0.059}$ | $0.087_{\pm0.025}$ | $0.006_{\pm0.001}$ | $0.275_{\pm0.033}$ | $0.333_{\pm0.066}$ | $0.084_{\pm0.028}$ | $0.006_{\pm0.001}$ |
| | TimeWak | $0.108_{\pm0.014}$ | $0.413_{\pm0.062}$ | $0.038_{\pm0.021}$ | $0.007_{\pm0.001}$ | $0.155_{\pm0.016}$ | $0.316_{\pm0.022}$ | $0.046_{\pm0.030}$ | $0.005_{\pm0.001}$ |
| Energy | W/O | $0.112_{\pm0.016}$ | $1.032_{\pm0.289}$ | $0.124_{\pm0.008}$ | $0.251_{\pm0.001}$ | $0.120_{\pm0.011}$ | $0.798_{\pm0.213}$ | $0.202_{\pm0.073}$ | $0.249_{\pm0.000}$ |
| | TR | $0.902_{\pm0.093}$ | $3.503_{\pm0.589}$ | $0.427_{\pm0.072}$ | $0.307_{\pm0.005}$ | $1.195_{\pm0.111}$ | $2.303_{\pm0.569}$ | $0.498_{\pm0.001}$ | $0.287_{\pm0.003}$ |
| | GS | $2.205_{\pm0.192}$ | $3.277_{\pm0.129}$ | $0.479_{\pm0.010}$ | $0.310_{\pm0.005}$ | $3.680_{\pm0.444}$ | $4.233_{\pm0.287}$ | $0.474_{\pm0.039}$ | $0.280_{\pm0.006}$ |
| | HTW | $0.133_{\pm0.015}$ | $1.045_{\pm0.357}$ | $0.135_{\pm0.013}$ | $0.251_{\pm0.001}$ | $0.133_{\pm0.017}$ | $0.822_{\pm0.310}$ | $0.103_{\pm0.043}$ | $0.249_{\pm0.001}$ |
| | TabWak | $0.168_{\pm0.021}$ | $1.811_{\pm0.530}$ | $0.136_{\pm0.013}$ | $0.251_{\pm0.000}$ | $0.201_{\pm0.020}$ | $2.001_{\pm0.440}$ | $0.156_{\pm0.077}$ | $0.250_{\pm0.001}$ |
| | TabWak$^\top$ | $0.237_{\pm0.029}$ | $1.321_{\pm0.252}$ | $0.138_{\pm0.015}$ | $0.252_{\pm0.000}$ | $0.274_{\pm0.019}$ | $1.211_{\pm0.196}$ | $0.136_{\pm0.023}$ | $0.249_{\pm0.001}$ |
| | TimeWak | $0.143_{\pm0.019}$ | $1.662_{\pm0.298}$ | $0.145_{\pm0.019}$ | $0.251_{\pm0.000}$ | $0.148_{\pm0.027}$ | $1.687_{\pm0.328}$ | $0.140_{\pm0.057}$ | $0.249_{\pm0.000}$ |
| fMRI | W/O | $0.435_{\pm0.033}$ | $1.899_{\pm0.075}$ | $0.268_{\pm0.150}$ | $0.100_{\pm0.000}$ | $0.859_{\pm0.058}$ | $1.823_{\pm0.064}$ | $0.209_{\pm0.263}$ | $0.100_{\pm0.000}$ |
| | TR | $3.358_{\pm0.402}$ | $13.699_{\pm0.132}$ | $0.411_{\pm0.158}$ | $0.141_{\pm0.001}$ | $5.391_{\pm0.919}$ | $13.000_{\pm0.084}$ | $0.434_{\pm0.072}$ | $0.149_{\pm0.002}$ |
| | GS | $0.756_{\pm0.098}$ | $8.378_{\pm0.028}$ | $0.500_{\pm0.001}$ | $0.104_{\pm0.001}$ | $1.046_{\pm0.052}$ | $5.941_{\pm0.048}$ | $0.499_{\pm0.001}$ | $0.106_{\pm0.001}$ |
| | HTW | $0.413_{\pm0.017}$ | $1.822_{\pm0.111}$ | $0.338_{\pm0.032}$ | $0.100_{\pm0.000}$ | $0.811_{\pm0.051}$ | $1.757_{\pm0.053}$ | $0.063_{\pm0.064}$ | $0.100_{\pm0.000}$ |
| | TabWak | $0.655_{\pm0.137}$ | $6.532_{\pm0.158}$ | $0.242_{\pm0.302}$ | $0.108_{\pm0.001}$ | $1.114_{\pm0.135}$ | $5.967_{\pm0.033}$ | $0.153_{\pm0.218}$ | $0.111_{\pm0.001}$ |
| | TabWak$^\top$ | $0.554_{\pm0.049}$ | $1.955_{\pm0.058}$ | $0.331_{\pm0.039}$ | $0.100_{\pm0.000}$ | $0.919_{\pm0.024}$ | $1.807_{\pm0.024}$ | $0.265_{\pm0.201}$ | $0.100_{\pm0.000}$ |
| | TimeWak | $0.441_{\pm0.035}$ | $1.786_{\pm0.043}$ | $0.314_{\pm0.041}$ | $0.100_{\pm0.000}$ | $0.855_{\pm0.072}$ | $1.704_{\pm0.060}$ | $0.298_{\pm0.227}$ | $0.100_{\pm0.000}$ |

## E.2 Post-editing attacks

Table 7–8 show the detectability results of several post-editing attacks on 24 and 128 window sizes, respectively.

Table 7: Results of robustness against post-editing attacks. Average Z-score on 24-length sequences, including un-attacked scores from Table 1. Best results are in **bold**, and second-best are underlined.

| Dataset | Method | Without Attack | Offset ↑ 5% | Offset ↑ 30% | Random Crop ↑ 5% | Random Crop ↑ 30% | Min-Max Insertion ↑ 5% | Min-Max Insertion ↑ 30% |
|---|---|---|---|---|---|---|---|---|
| Stocks | TR | $0.43_{\pm 0.04}$ | $0.41_{\pm 0.04}$ | $0.26_{\pm 0.05}$ | $61.01_{\pm 1.18}$ | $\mathbf{76.69_{\pm 1.17}}$ | $0.18_{\pm 0.05}$ | $0.81_{\pm 0.08}$ |
| | GS | $\underline{86.07_{\pm 0.74}}$ | $\underline{85.74_{\pm 0.71}}$ | $\underline{84.38_{\pm 0.67}}$ | $-34.10_{\pm 0.69}$ | $3.08_{\pm 0.58}$ | $\underline{82.31_{\pm 0.75}}$ | $\mathbf{57.07_{\pm 0.72}}$ |
| | HTW | $4.45_{\pm 0.62}$ | $4.45_{\pm 0.62}$ | $4.45_{\pm 0.62}$ | $-0.30_{\pm 0.39}$ | $-0.40_{\pm 0.30}$ | $3.96_{\pm 0.72}$ | $1.80_{\pm 0.71}$ |
| | TabWak | $-67.22_{\pm 1.17}$ | $-66.62_{\pm 1.23}$ | $-75.89_{\pm 1.10}$ | $-137.99_{\pm 1.23}$ | $-72.71_{\pm 1.76}$ | $-70.11_{\pm 1.14}$ | $-89.22_{\pm 0.64}$ |
| | TabWak$^\top$ | $55.39_{\pm 0.86}$ | $55.59_{\pm 0.74}$ | $54.76_{\pm 0.78}$ | $9.32_{\pm 0.94}$ | $7.43_{\pm 0.73}$ | $45.69_{\pm 0.73}$ | $21.19_{\pm 0.56}$ |
| | TimeWak | $\mathbf{182.10_{\pm 0.73}}$ | $\mathbf{181.98_{\pm 0.84}}$ | $\mathbf{180.68_{\pm 0.77}}$ | $\mathbf{62.29_{\pm 1.01}}$ | $\underline{9.33_{\pm 1.04}}$ | $\mathbf{155.72_{\pm 0.91}}$ | $\underline{56.86_{\pm 1.06}}$ |
| ETTh | TR | $7.84_{\pm 0.12}$ | $7.94_{\pm 0.12}$ | $8.19_{\pm 0.13}$ | $26.95_{\pm 0.32}$ | $\mathbf{34.36_{\pm 0.26}}$ | $14.11_{\pm 0.28}$ | $\underline{28.72_{\pm 0.50}}$ |
| | GS | $101.07_{\pm 1.17}$ | $99.45_{\pm 1.09}$ | $\underline{105.70_{\pm 1.08}}$ | $\mathbf{98.99_{\pm 1.21}}$ | $2.10_{\pm 1.37}$ | $\underline{98.78_{\pm 1.13}}$ | $\mathbf{73.67_{\pm 1.30}}$ |
| | HTW | $3.43_{\pm 0.83}$ | $3.43_{\pm 0.83}$ | $3.43_{\pm 0.83}$ | $0.38_{\pm 0.77}$ | $-1.08_{\pm 0.47}$ | $3.09_{\pm 0.82}$ | $1.54_{\pm 0.66}$ |
| | TabWak | $-14.95_{\pm 1.08}$ | $-7.92_{\pm 0.97}$ | $36.49_{\pm 1.08}$ | $-11.76_{\pm 1.31}$ | $\underline{8.65_{\pm 1.38}}$ | $-7.95_{\pm 0.93}$ | $21.70_{\pm 0.82}$ |
| | TabWak$^\top$ | $\underline{109.35_{\pm 0.91}}$ | $\underline{110.22_{\pm 0.85}}$ | $103.34_{\pm 0.82}$ | $\underline{50.64_{\pm 0.96}}$ | $1.06_{\pm 1.02}$ | $80.98_{\pm 0.87}$ | $21.83_{\pm 0.95}$ |
| | TimeWak | $\mathbf{134.83_{\pm 0.95}}$ | $\mathbf{130.50_{\pm 0.99}}$ | $\mathbf{118.65_{\pm 1.03}}$ | $35.30_{\pm 1.14}$ | $5.99_{\pm 1.04}$ | $\mathbf{101.53_{\pm 0.96}}$ | $20.77_{\pm 0.95}$ |
| MuJoCo | TR | $1.38_{\pm 0.03}$ | $1.34_{\pm 0.03}$ | $1.22_{\pm 0.03}$ | $5.31_{\pm 0.05}$ | $5.65_{\pm 0.06}$ | $2.42_{\pm 0.05}$ | $5.53_{\pm 0.09}$ |
| | GS | $21.13_{\pm 0.77}$ | $23.95_{\pm 0.71}$ | $27.15_{\pm 0.76}$ | $23.06_{\pm 0.86}$ | $-10.38_{\pm 0.72}$ | $22.89_{\pm 0.83}$ | $22.89_{\pm 1.06}$ |
| | HTW | $2.89_{\pm 0.54}$ | $2.89_{\pm 0.54}$ | $2.89_{\pm 0.54}$ | $0.85_{\pm 0.48}$ | $-0.47_{\pm 0.40}$ | $2.65_{\pm 0.53}$ | $1.51_{\pm 0.43}$ |
| | TabWak | $\underline{31.30_{\pm 1.07}}$ | $\underline{31.59_{\pm 0.96}}$ | $\underline{39.05_{\pm 1.05}}$ | $\underline{38.80_{\pm 1.11}}$ | $\mathbf{25.04_{\pm 1.13}}$ | $\underline{31.60_{\pm 0.92}}$ | $\underline{28.70_{\pm 0.59}}$ |
| | TabWak$^\top$ | $-4.85_{\pm 0.87}$ | $-6.27_{\pm 0.90}$ | $0.74_{\pm 0.78}$ | $-15.92_{\pm 0.88}$ | $-56.15_{\pm 1.50}$ | $-10.95_{\pm 1.01}$ | $-43.08_{\pm 1.17}$ |
| | TimeWak | $\mathbf{85.69_{\pm 1.08}}$ | $\mathbf{77.78_{\pm 1.26}}$ | $\mathbf{49.48_{\pm 1.12}}$ | $\mathbf{48.03_{\pm 1.17}}$ | $\underline{11.46_{\pm 1.18}}$ | $\mathbf{74.94_{\pm 1.10}}$ | $\mathbf{38.87_{\pm 1.14}}$ |
| Energy | TR | $9.51_{\pm 0.09}$ | $9.30_{\pm 0.10}$ | $8.77_{\pm 0.08}$ | $\mathbf{110.47_{\pm 0.23}}$ | $\mathbf{129.09_{\pm 0.19}}$ | $17.65_{\pm 0.15}$ | $35.69_{\pm 0.18}$ |
| | GS | $\underline{51.22_{\pm 0.88}}$ | $\underline{50.27_{\pm 0.96}}$ | $\underline{47.31_{\pm 0.97}}$ | $11.65_{\pm 0.72}$ | $\underline{38.70_{\pm 1.34}}$ | $\underline{48.80_{\pm 0.97}}$ | $\underline{39.80_{\pm 0.94}}$ |
| | HTW | $3.06_{\pm 0.36}$ | $3.06_{\pm 0.36}$ | $3.06_{\pm 0.36}$ | $0.56_{\pm 0.35}$ | $-0.13_{\pm 0.23}$ | $2.81_{\pm 0.36}$ | $1.61_{\pm 0.30}$ |
| | TabWak | $3.26_{\pm 0.89}$ | $0.52_{\pm 1.03}$ | $-2.91_{\pm 1.08}$ | $3.27_{\pm 0.63}$ | $3.54_{\pm 0.53}$ | $2.20_{\pm 0.87}$ | $-2.08_{\pm 0.49}$ |
| | TabWak$^\top$ | $40.82_{\pm 0.81}$ | $38.47_{\pm 0.72}$ | $40.20_{\pm 0.82}$ | $\underline{33.83_{\pm 1.08}}$ | $-7.47_{\pm 1.50}$ | $31.89_{\pm 0.89}$ | $5.37_{\pm 0.93}$ |
| | TimeWak | $\mathbf{231.28_{\pm 1.45}}$ | $\mathbf{228.22_{\pm 1.71}}$ | $\mathbf{185.48_{\pm 1.52}}$ | $-9.20_{\pm 0.96}$ | $-1.54_{\pm 1.02}$ | $\mathbf{189.15_{\pm 1.44}}$ | $\mathbf{56.39_{\pm 1.16}}$ |
| fMRI | TR | $6.49_{\pm 0.05}$ | $6.43_{\pm 0.05}$ | $6.12_{\pm 0.05}$ | $5.42_{\pm 0.05}$ | $4.40_{\pm 0.05}$ | $5.54_{\pm 0.05}$ | $1.37_{\pm 0.07}$ |
| | GS | $\underline{420.02_{\pm 1.44}}$ | $\underline{416.83_{\pm 1.43}}$ | $\underline{401.86_{\pm 1.72}}$ | $\underline{360.27_{\pm 1.40}}$ | $\underline{171.38_{\pm 1.16}}$ | $\underline{386.10_{\pm 1.54}}$ | $\mathbf{245.44_{\pm 1.28}}$ |
| | HTW | $4.32_{\pm 0.22}$ | $4.32_{\pm 0.22}$ | $4.32_{\pm 0.22}$ | $2.92_{\pm 0.27}$ | $-0.39_{\pm 0.28}$ | $3.86_{\pm 0.26}$ | $1.78_{\pm 0.25}$ |
| | TabWak | $84.02_{\pm 1.04}$ | $83.55_{\pm 1.08}$ | $78.94_{\pm 1.16}$ | $78.63_{\pm 1.24}$ | $27.05_{\pm 1.30}$ | $76.49_{\pm 1.17}$ | $50.55_{\pm 1.33}$ |
| | TabWak$^\top$ | $\mathbf{464.67_{\pm 0.50}}$ | $\mathbf{464.04_{\pm 0.48}}$ | $\mathbf{458.81_{\pm 0.57}}$ | $\mathbf{400.47_{\pm 0.72}}$ | $\mathbf{202.57_{\pm 1.04}}$ | $\mathbf{403.49_{\pm 0.66}}$ | $\underline{168.64_{\pm 0.75}}$ |
| | TimeWak | $379.51_{\pm 0.82}$ | $378.95_{\pm 0.85}$ | $374.99_{\pm 0.78}$ | $277.64_{\pm 0.78}$ | $77.74_{\pm 1.05}$ | $327.13_{\pm 0.86}$ | $133.68_{\pm 0.85}$ |

Table 8: Results of robustness against post-editing attacks. Average Z-score on 128-length sequences, including un-attacked scores from Table 1. Best results are in **bold**, and second-best are underlined.

| Dataset | Method | Without Attack | Offset ↑ 5% | Offset ↑ 30% | Random Crop ↑ 5% | Random Crop ↑ 30% | Min-Max Insertion ↑ 5% | Min-Max Insertion ↑ 30% |
|---|---|---|---|---|---|---|---|---|
| Stocks | TR | $0.08_{\pm0.10}$ | $0.05_{\pm0.10}$ | $0.02_{\pm0.12}$ | $\mathbf{93.62_{\pm0.99}}$ | $\mathbf{119.45_{\pm1.16}}$ | $6.87_{\pm0.22}$ | $23.98_{\pm0.55}$ |
| | GS | $172.23_{\pm1.08}$ | $167.39_{\pm1.10}$ | $145.89_{\pm1.83}$ | $13.22_{\pm1.43}$ | $-2.23_{\pm1.03}$ | $144.05_{\pm0.98}$ | $64.79_{\pm1.01}$ |
| | HTW | $10.39_{\pm1.33}$ | $10.39_{\pm1.33}$ | $10.39_{\pm1.33}$ | $-1.30_{\pm0.44}$ | $-1.42_{\pm0.33}$ | $9.05_{\pm1.67}$ | $3.76_{\pm1.63}$ |
| | TabWak | $-10.49_{\pm1.28}$ | $49.98_{\pm3.31}$ | $36.78_{\pm3.15}$ | $69.44_{\pm3.32}$ | $70.32_{\pm2.93}$ | $9.26_{\pm1.41}$ | $33.81_{\pm0.81}$ |
| | TabWak$^\top$ | $129.39_{\pm0.90}$ | $131.70_{\pm0.97}$ | $129.83_{\pm1.08}$ | $7.70_{\pm1.04}$ | $-4.86_{\pm1.20}$ | $80.70_{\pm1.03}$ | $15.70_{\pm0.57}$ |
| | TimeWak | $\mathbf{550.05_{\pm1.18}}$ | $\mathbf{535.16_{\pm0.96}}$ | $\mathbf{525.09_{\pm1.20}}$ | $83.81_{\pm2.15}$ | $15.42_{\pm1.02}$ | $\mathbf{385.68_{\pm2.48}}$ | $\mathbf{85.38_{\pm1.37}}$ |
| ETTh | TR | $6.18_{\pm0.16}$ | $6.14_{\pm0.16}$ | $6.08_{\pm0.14}$ | $24.37_{\pm0.28}$ | $36.22_{\pm0.21}$ | $23.02_{\pm0.25}$ | $55.97_{\pm0.40}$ |
| | GS | $327.47_{\pm4.44}$ | $322.35_{\pm5.15}$ | $\mathbf{296.89_{\pm5.10}}$ | $\mathbf{189.84_{\pm2.98}}$ | $8.49_{\pm1.16}$ | $\mathbf{296.61_{\pm4.10}}$ | $\mathbf{172.81_{\pm2.05}}$ |
| | HTW | $6.84_{\pm2.22}$ | $6.84_{\pm2.22}$ | $6.84_{\pm2.22}$ | $0.79_{\pm1.90}$ | $-2.94_{\pm1.15}$ | $6.00_{\pm2.16}$ | $2.59_{\pm1.54}$ |
| | TabWak | $-20.57_{\pm0.96}$ | $-18.36_{\pm1.21}$ | $-3.56_{\pm1.91}$ | $37.32_{\pm2.48}$ | $\mathbf{78.54_{\pm2.07}}$ | $-7.75_{\pm0.79}$ | $22.65_{\pm0.45}$ |
| | TabWak$^\top$ | $235.03_{\pm1.48}$ | $227.57_{\pm1.33}$ | $194.84_{\pm1.65}$ | $101.64_{\pm1.73}$ | $23.82_{\pm1.45}$ | $155.78_{\pm1.24}$ | $23.70_{\pm0.91}$ |
| | TimeWak | $\mathbf{340.36_{\pm2.06}}$ | $\mathbf{333.66_{\pm1.93}}$ | $276.99_{\pm2.04}$ | $107.52_{\pm1.90}$ | $7.13_{\pm1.14}$ | $244.68_{\pm1.72}$ | $44.75_{\pm1.21}$ |
| MuJoCo | TR | $1.31_{\pm0.04}$ | $1.33_{\pm0.04}$ | $1.50_{\pm0.04}$ | $6.93_{\pm0.07}$ | $6.08_{\pm0.05}$ | $10.52_{\pm0.12}$ | $25.61_{\pm0.20}$ |
| | GS | $39.63_{\pm0.71}$ | $38.07_{\pm0.67}$ | $32.10_{\pm0.81}$ | $37.22_{\pm0.89}$ | $10.61_{\pm0.85}$ | $38.46_{\pm0.89}$ | $31.58_{\pm1.16}$ |
| | HTW | $4.20_{\pm1.37}$ | $4.20_{\pm1.37}$ | $4.20_{\pm1.37}$ | $1.73_{\pm1.18}$ | $-2.13_{\pm0.99}$ | $3.78_{\pm1.29}$ | $2.02_{\pm0.90}$ |
| | TabWak | $-3.00_{\pm1.04}$ | $-5.68_{\pm0.92}$ | $-3.91_{\pm1.16}$ | $-8.02_{\pm1.07}$ | $16.29_{\pm0.61}$ | $-0.31_{\pm0.87}$ | $2.90_{\pm0.55}$ |
| | TabWak$^\top$ | $3.91_{\pm0.88}$ | $5.10_{\pm1.00}$ | $9.07_{\pm0.97}$ | $3.36_{\pm0.90}$ | $\mathbf{24.65_{\pm1.15}}$ | $25.56_{\pm1.00}$ | $\mathbf{41.50_{\pm1.00}}$ |
| | TimeWak | $\mathbf{123.36_{\pm1.43}}$ | $\mathbf{139.07_{\pm1.49}}$ | $\mathbf{111.27_{\pm1.15}}$ | $\mathbf{93.19_{\pm1.46}}$ | $12.44_{\pm1.01}$ | $\mathbf{71.22_{\pm1.21}}$ | $7.26_{\pm0.95}$ |
| Energy | TR | $22.58_{\pm0.19}$ | $21.92_{\pm0.18}$ | $20.13_{\pm0.17}$ | $\mathbf{167.48_{\pm0.64}}$ | $\mathbf{207.75_{\pm0.39}}$ | $68.40_{\pm0.29}$ | $\mathbf{148.79_{\pm0.50}}$ |
| | GS | $45.42_{\pm1.05}$ | $43.41_{\pm0.96}$ | $30.53_{\pm0.93}$ | $71.60_{\pm0.94}$ | $67.39_{\pm1.11}$ | $44.20_{\pm1.16}$ | $44.13_{\pm1.18}$ |
| | HTW | $5.42_{\pm1.00}$ | $5.42_{\pm1.00}$ | $5.42_{\pm1.00}$ | $0.24_{\pm0.62}$ | $-1.11_{\pm0.43}$ | $4.86_{\pm0.95}$ | $2.47_{\pm0.67}$ |
| | TabWak | $0.57_{\pm0.87}$ | $-21.06_{\pm0.96}$ | $-28.01_{\pm1.32}$ | $-25.69_{\pm0.82}$ | $-10.44_{\pm0.76}$ | $-0.07_{\pm0.63}$ | $-7.15_{\pm0.36}$ |
| | TabWak$^\top$ | $26.00_{\pm1.12}$ | $33.08_{\pm0.98}$ | $19.25_{\pm0.95}$ | $9.30_{\pm1.14}$ | $10.84_{\pm1.92}$ | $24.21_{\pm0.95}$ | $33.38_{\pm0.69}$ |
| | TimeWak | $\mathbf{245.37_{\pm2.88}}$ | $\mathbf{307.31_{\pm1.98}}$ | $\mathbf{183.59_{\pm1.74}}$ | $9.57_{\pm0.97}$ | $2.48_{\pm0.84}$ | $\mathbf{171.81_{\pm1.77}}$ | $21.44_{\pm1.10}$ |
| fMRI | TR | $9.94_{\pm0.04}$ | $9.93_{\pm0.04}$ | $9.88_{\pm0.04}$ | $10.47_{\pm0.05}$ | $8.96_{\pm0.04}$ | $16.62_{\pm0.06}$ | $39.21_{\pm0.13}$ |
| | GS | $701.90_{\pm0.72}$ | $701.47_{\pm0.65}$ | $699.01_{\pm0.63}$ | $621.08_{\pm1.08}$ | $256.13_{\pm1.63}$ | $667.14_{\pm2.43}$ | $\mathbf{528.66_{\pm2.82}}$ |
| | HTW | $9.43_{\pm0.61}$ | $9.43_{\pm0.61}$ | $9.43_{\pm0.61}$ | $6.75_{\pm0.67}$ | $-1.09_{\pm0.65}$ | $8.24_{\pm0.66}$ | $3.45_{\pm0.57}$ |
| | TabWak | $47.29_{\pm0.83}$ | $43.52_{\pm0.79}$ | $24.95_{\pm0.86}$ | $-57.42_{\pm2.04}$ | $71.03_{\pm3.25}$ | $78.43_{\pm4.67}$ | $0.17_{\pm8.00}$ |
| | TabWak$^\top$ | $\mathbf{1031.96_{\pm0.79}}$ | $\mathbf{1031.53_{\pm0.78}}$ | $\mathbf{1030.14_{\pm0.72}}$ | $\mathbf{889.62_{\pm0.84}}$ | $\mathbf{383.15_{\pm1.37}}$ | $\mathbf{801.16_{\pm0.83}}$ | $229.37_{\pm1.15}$ |
| | TimeWak | $526.81_{\pm13.12}$ | $834.78_{\pm1.10}$ | $834.62_{\pm1.24}$ | $632.77_{\pm1.07}$ | $160.62_{\pm1.02}$ | $651.80_{\pm1.13}$ | $216.50_{\pm0.94}$ |

## E.3 Reconstruction attack

We implement the reconstruction attack using the original diffusion model. Specifically, we first applied the $q$-sampling process up to half of the total diffusion steps (i.e., midpoint timestep), and then performed reverse sampling starting from this midpoint. The results obtained from this approach are presented in Table 9. We observe that although the Z-score decreases, our watermark remains detectable.

Table 9: Results of synthetic time series quality and watermark detectability. Comparing `TimeWak` and `TimeWak`$_{recon}$ under reconstruction attack. Quality metrics and Z-score are for 24-length sequences.

| Dataset | Method | Context-FID ↓ | Correlational ↓ | Discriminative ↓ | Predictive ↓ | Z-score ↑ |
|---|---|---|---|---|---|---|
| Stocks | TimeWak | $0.277_{\pm0.019}$ | $0.020_{\pm0.018}$ | $0.120_{\pm0.039}$ | $0.038_{\pm0.000}$ | $182.10_{\pm0.73}$ |
| | TimeWak$_{recon}$ | $4.570_{\pm0.502}$ | $0.031_{\pm0.028}$ | $0.393_{\pm0.073}$ | $0.148_{\pm0.017}$ | $179.41_{\pm0.81}$ |
| ETTh | TimeWak | $0.237_{\pm0.017}$ | $0.212_{\pm0.043}$ | $0.102_{\pm0.014}$ | $0.122_{\pm0.002}$ | $134.83_{\pm0.95}$ |
| | TimeWak$_{recon}$ | $1.743_{\pm0.225}$ | $0.138_{\pm0.012}$ | $0.290_{\pm0.009}$ | $0.158_{\pm0.002}$ | $82.11_{\pm2.58}$ |
| MuJoCo | TimeWak | $0.089_{\pm0.017}$ | $0.532_{\pm0.137}$ | $0.044_{\pm0.021}$ | $0.008_{\pm0.001}$ | $85.69_{\pm1.08}$ |
| | TimeWak$_{recon}$ | $0.925_{\pm0.047}$ | $0.622_{\pm0.063}$ | $0.261_{\pm0.010}$ | $0.008_{\pm0.002}$ | $81.97_{\pm1.33}$ |
| Energy | TimeWak | $0.121_{\pm0.016}$ | $1.977_{\pm0.750}$ | $0.142_{\pm0.008}$ | $0.254_{\pm0.000}$ | $231.28_{\pm1.45}$ |
| | TimeWak$_{recon}$ | $5.158_{\pm0.568}$ | $6.678_{\pm0.087}$ | $0.444_{\pm0.006}$ | $0.263_{\pm0.002}$ | $39.99_{\pm2.03}$ |
| fMRI | TimeWak | $0.199_{\pm0.010}$ | $2.006_{\pm0.053}$ | $0.122_{\pm0.033}$ | $0.100_{\pm0.000}$ | $379.51_{\pm0.82}$ |
| | TimeWak$_{recon}$ | $0.595_{\pm0.036}$ | $2.374_{\pm0.105}$ | $0.431_{\pm0.017}$ | $0.102_{\pm0.000}$ | $457.90_{\pm0.81}$ |

## E.4 BDIA-DDIM on other baselines

The baselines in our main experiments were not evaluated with BDIA-DDIM. This is because BDIA-DDIM tends to degrade data quality compared to standard DDIM, representing a trade-off for achieving lower inversion error. For many baselines, such as Tree-Ring and Gaussian Shading, the generated quality is already poor. Applying BDIA-DDIM in these cases might improve detectability but would further deteriorate quality, making the results less meaningful. And for TabWak, we include results with BDIA-DDIM applied to both TabWak and TabWak$^\top$ in Table 10. While BDIA-DDIM improves detectability for both variants, it comes at the cost of further quality degradation compared to the original results in Table 1. In contrast, `TimeWak` maintains stable performance across both quality metrics and Z-score, highlighting its robustness.

Table 10: Results of synthetic time series quality and watermark detectability. All results are applied with BDIA-DDIM. Quality metrics are for 24-length sequences.

| Dataset | Method | Quality Metric ↓ | | | | Z-score ↑ | | |
|---|---|---|---|---|---|---|---|---|
| | | Context-FID | Correlational | Discriminative | Predictive | 24-length | 64-length | 128-length |
| Stocks | TabWak | $0.267_{\pm0.042}$ | $0.017_{\pm0.014}$ | $0.122_{\pm0.029}$ | $0.039_{\pm0.000}$ | $43.10_{\pm0.75}$ | $92.56_{\pm0.37}$ | $89.18_{\pm0.39}$ |
| | TabWak$^\top$ | $0.273_{\pm0.103}$ | $0.011_{\pm0.008}$ | $0.115_{\pm0.042}$ | $0.037_{\pm0.000}$ | $117.12_{\pm0.15}$ | $170.41_{\pm0.14}$ | $267.27_{\pm0.18}$ |
| | TimeWak | $0.277_{\pm0.019}$ | $0.020_{\pm0.018}$ | $0.120_{\pm0.039}$ | $0.038_{\pm0.000}$ | $182.10_{\pm0.73}$ | $395.34_{\pm1.24}$ | $550.05_{\pm1.18}$ |
| ETTh | TabWak | $0.431_{\pm0.033}$ | $0.436_{\pm0.020}$ | $0.133_{\pm0.029}$ | $0.134_{\pm0.002}$ | $1.73_{\pm0.76}$ | $16.26_{\pm0.98}$ | $31.59_{\pm0.87}$ |
| | TabWak$^\top$ | $0.454_{\pm0.049}$ | $0.132_{\pm0.018}$ | $0.104_{\pm0.024}$ | $0.119_{\pm0.006}$ | $149.16_{\pm0.62}$ | $220.72_{\pm0.84}$ | $315.34_{\pm1.16}$ |
| | TimeWak | $0.237_{\pm0.017}$ | $0.212_{\pm0.043}$ | $0.102_{\pm0.014}$ | $0.122_{\pm0.002}$ | $134.83_{\pm0.95}$ | $236.08_{\pm1.63}$ | $340.36_{\pm2.06}$ |
| MuJoCo | TabWak | $0.489_{\pm0.036}$ | $0.958_{\pm0.067}$ | $0.204_{\pm0.056}$ | $0.010_{\pm0.004}$ | $13.97_{\pm1.07}$ | $20.52_{\pm0.85}$ | $4.23_{\pm0.93}$ |
| | TabWak$^\top$ | $0.270_{\pm0.024}$ | $0.378_{\pm0.033}$ | $0.128_{\pm0.015}$ | $0.008_{\pm0.002}$ | $68.32_{\pm1.14}$ | $93.55_{\pm1.06}$ | $228.10_{\pm1.67}$ |
| | TimeWak | $0.089_{\pm0.017}$ | $0.532_{\pm0.137}$ | $0.044_{\pm0.021}$ | $0.008_{\pm0.001}$ | $85.69_{\pm1.08}$ | $56.45_{\pm1.26}$ | $123.36_{\pm1.43}$ |
| Energy | TabWak | $0.189_{\pm0.022}$ | $2.915_{\pm0.410}$ | $0.166_{\pm0.012}$ | $0.255_{\pm0.000}$ | $3.10_{\pm0.63}$ | $-9.67_{\pm0.72}$ | $-18.64_{\pm0.71}$ |
| | TabWak$^\top$ | $0.199_{\pm0.007}$ | $1.648_{\pm0.188}$ | $0.137_{\pm0.019}$ | $0.265_{\pm0.004}$ | $246.86_{\pm1.30}$ | $258.45_{\pm1.48}$ | $302.78_{\pm1.98}$ |
| | TimeWak | $0.121_{\pm0.016}$ | $1.977_{\pm0.750}$ | $0.142_{\pm0.008}$ | $0.254_{\pm0.000}$ | $231.28_{\pm1.45}$ | $267.53_{\pm2.60}$ | $245.37_{\pm2.88}$ |
| fMRI | TabWak | $0.319_{\pm0.019}$ | $6.772_{\pm0.129}$ | $0.484_{\pm0.007}$ | $0.110_{\pm0.001}$ | $57.99_{\pm1.06}$ | $268.67_{\pm0.88}$ | $-94.33_{\pm0.99}$ |
| | TabWak$^\top$ | $0.317_{\pm0.028}$ | $2.185_{\pm0.148}$ | $0.218_{\pm0.031}$ | $0.100_{\pm0.000}$ | $471.86_{\pm0.45}$ | $739.39_{\pm0.60}$ | $1014.93_{\pm0.74}$ |
| | TimeWak | $0.199_{\pm0.010}$ | $2.006_{\pm0.053}$ | $0.122_{\pm0.033}$ | $0.100_{\pm0.000}$ | $379.51_{\pm0.82}$ | $595.68_{\pm1.03}$ | $526.81_{\pm13.12}$ |

### E.5  Watermarked dataset on downstream tasks

We implement time series forecasting and imputation as the downstream tasks, i.e, taking either the real, synthetic, or watermarked synthetic data to build a diffusion model that can predict the future or missing values of time series. We compared the mean squared error (MSE) between the predicted and actual values and summarized the results in Table 11 and Table 12. The results indicate that training on watermarked synthetic data has a minimal impact on forecasting and imputation performance compared to training on non-watermarked synthetic data.

Table 11: Results of 64-length time series forecasting that trains on real and synthetic data (Synth$_{\text{W/O}}$ and Synth$_{\texttt{TimeWak}}$) and tests on real data with a 24 timesteps forecast horizon. MSE values $\times 10^{-3}$.

| Dataset | Training Data | MSE $\downarrow$ |
|---|---|---|
| Stocks | Real | 2.119 |
| | Synth$_{\text{W/O}}$ | 2.022 |
| | Synth$_{\texttt{TimeWak}}$ | 2.014 |
| ETTh | Real | 6.678 |
| | Synth$_{\text{W/O}}$ | 8.541 |
| | Synth$_{\texttt{TimeWak}}$ | 8.655 |
| MuJoCo | Real | 1.312 |
| | Synth$_{\text{W/O}}$ | 1.615 |
| | Synth$_{\texttt{TimeWak}}$ | 1.741 |
| Energy | Real | 12.715 |
| | Synth$_{\text{W/O}}$ | 13.480 |
| | Synth$_{\texttt{TimeWak}}$ | 13.717 |
| fMRI | Real | 36.423 |
| | Synth$_{\text{W/O}}$ | 67.796 |
| | Synth$_{\texttt{TimeWak}}$ | 67.944 |

Table 12: Results of 64-length time series imputation that trains on real and synthetic data (Synth$_{\text{W/O}}$ and Synth$_{\texttt{TimeWak}}$) and tests on real data with 70% missing ratio. MSE values $\times 10^{-3}$.

| Dataset | Training Data | MSE $\downarrow$ |
|---|---|---|
| Stocks | Real | 1.020 |
| | Synth$_{\text{W/O}}$ | 0.855 |
| | Synth$_{\texttt{TimeWak}}$ | 0.858 |
| ETTh | Real | 1.526 |
| | Synth$_{\text{W/O}}$ | 1.842 |
| | Synth$_{\texttt{TimeWak}}$ | 1.963 |
| MuJoCo | Real | 0.101 |
| | Synth$_{\text{W/O}}$ | 0.364 |
| | Synth$_{\texttt{TimeWak}}$ | 0.387 |
| Energy | Real | 7.926 |
| | Synth$_{\text{W/O}}$ | 8.258 |
| | Synth$_{\texttt{TimeWak}}$ | 8.279 |
| fMRI | Real | 27.439 |
| | Synth$_{\text{W/O}}$ | 45.412 |
| | Synth$_{\texttt{TimeWak}}$ | 47.349 |

## E.6 Ablation study

To assess the effectiveness of the `TimeWak`, we compare its full version with three distinct variants outlined in Table 13. Table 14 presents the quality of the watermarked time series data for sequences of length 24, and the detectability of the watermarks across lengths of 24, 64 and 128. `TimeWak` demonstrates a comparable quality performance and high detectability.

Table 13: List of methods, including `TimeWak`, to be compared.

| Method | Watermarking Direction | Sampling |
|---|---|---|
| SpatDDIM | Spatial | DDIM |
| SpatBDIA | Spatial | BDIA-DDIM |
| TempDDIM | Temporal | DDIM |
| TimeWak | Temporal | BDIA-DDIM |

Table 14: Results of synthetic time series quality and watermark detectability. Quality metrics are for 24-length sequences. All methods are originally found in Table 13. Best results are in **bold**, and second-best are underlined.

| Dataset | Method | Quality Metric ↓ | | | | Z-score ↑ | | |
|---|---|---|---|---|---|---|---|---|
| | | Context-FID | Correlational | Discriminative | Predictive | 24-length | 64-length | 128-length |
| Stocks | SpatDDIM | $0.233_{\pm0.025}$ | $\mathbf{0.012}_{\pm\mathbf{0.005}}$ | $0.127_{\pm0.019}$ | $\mathbf{0.037}_{\pm\mathbf{0.000}}$ | $11.06_{\pm1.07}$ | $0.18_{\pm0.77}$ | $-1.03_{\pm0.96}$ |
| | SpatBDIA | $\mathbf{0.199}_{\pm\mathbf{0.024}}$ | $0.024_{\pm0.028}$ | $\underline{0.124}_{\pm0.023}$ | $\mathbf{0.037}_{\pm\mathbf{0.000}}$ | $126.97_{\pm1.43}$ | $\underline{156.84}_{\pm0.80}$ | $\underline{170.13}_{\pm1.59}$ |
| | TempDDIM | $0.277_{\pm0.054}$ | $\underline{0.013}_{\pm0.005}$ | $\underline{0.124}_{\pm0.043}$ | $0.038_{\pm0.000}$ | $25.14_{\pm0.89}$ | $43.24_{\pm0.87}$ | $58.82_{\pm0.93}$ |
| | TimeWak | $0.277_{\pm0.019}$ | $0.020_{\pm0.018}$ | $\mathbf{0.120}_{\pm\mathbf{0.039}}$ | $\underline{0.038}_{\pm0.000}$ | $\mathbf{182.10}_{\pm\mathbf{0.73}}$ | $\mathbf{395.34}_{\pm\mathbf{1.24}}$ | $\mathbf{550.05}_{\pm\mathbf{1.18}}$ |
| ETTh | SpatDDIM | $0.249_{\pm0.020}$ | $\mathbf{0.145}_{\pm\mathbf{0.026}}$ | $\mathbf{0.094}_{\pm\mathbf{0.011}}$ | $0.124_{\pm0.002}$ | $10.80_{\pm1.06}$ | $11.13_{\pm0.81}$ | $7.29_{\pm0.95}$ |
| | SpatBDIA | $\underline{0.246}_{\pm0.015}$ | $0.150_{\pm0.034}$ | $0.098_{\pm0.011}$ | $0.123_{\pm0.007}$ | $28.11_{\pm1.10}$ | $40.67_{\pm1.21}$ | $47.73_{\pm1.04}$ |
| | TempDDIM | $0.249_{\pm0.013}$ | $\underline{0.150}_{\pm0.020}$ | $\underline{0.097}_{\pm0.010}$ | $\mathbf{0.121}_{\pm\mathbf{0.005}}$ | $63.78_{\pm0.97}$ | $102.06_{\pm1.16}$ | $152.63_{\pm1.29}$ |
| | TimeWak | $\mathbf{0.237}_{\pm\mathbf{0.017}}$ | $0.212_{\pm0.043}$ | $0.102_{\pm0.014}$ | $\underline{0.122}_{\pm0.002}$ | $\mathbf{134.83}_{\pm\mathbf{0.95}}$ | $\mathbf{236.08}_{\pm\mathbf{1.63}}$ | $\mathbf{340.36}_{\pm\mathbf{2.06}}$ |
| MuJoCo | SpatDDIM | $0.091_{\pm0.008}$ | $0.476_{\pm0.049}$ | $0.062_{\pm0.027}$ | $0.008_{\pm0.002}$ | $-4.62_{\pm0.91}$ | $15.42_{\pm0.96}$ | $7.12_{\pm1.22}$ |
| | SpatBDIA | $\underline{0.090}_{\pm0.016}$ | $0.491_{\pm0.078}$ | $\underline{0.051}_{\pm0.023}$ | $0.008_{\pm0.002}$ | $11.73_{\pm0.94}$ | $\underline{18.02}_{\pm0.95}$ | $25.42_{\pm0.90}$ |
| | TempDDIM | $0.098_{\pm0.010}$ | $\mathbf{0.450}_{\pm\mathbf{0.047}}$ | $0.059_{\pm0.027}$ | $\mathbf{0.007}_{\pm\mathbf{0.001}}$ | $\underline{21.85}_{\pm0.84}$ | $-0.88_{\pm0.99}$ | $-2.58_{\pm1.03}$ |
| | TimeWak | $\mathbf{0.089}_{\pm0.017}$ | $0.532_{\pm0.137}$ | $\mathbf{0.044}_{\pm\mathbf{0.021}}$ | $\underline{0.008}_{\pm0.001}$ | $\mathbf{85.69}_{\pm\mathbf{1.08}}$ | $\mathbf{56.45}_{\pm\mathbf{1.26}}$ | $\mathbf{123.36}_{\pm\mathbf{1.43}}$ |
| Energy | SpatDDIM | $0.135_{\pm0.021}$ | $\underline{1.814}_{\pm0.373}$ | $\underline{0.142}_{\pm0.013}$ | $\mathbf{0.253}_{\pm\mathbf{0.000}}$ | $1.55_{\pm0.90}$ | $6.75_{\pm1.00}$ | $-0.70_{\pm0.98}$ |
| | SpatBDIA | $0.142_{\pm0.027}$ | $2.104_{\pm0.254}$ | $0.149_{\pm0.025}$ | $\mathbf{0.253}_{\pm\mathbf{0.000}}$ | $52.86_{\pm0.90}$ | $63.56_{\pm1.26}$ | $81.46_{\pm1.11}$ |
| | TempDDIM | $\mathbf{0.110}_{\pm\mathbf{0.019}}$ | $\mathbf{1.724}_{\pm\mathbf{0.270}}$ | $0.142_{\pm0.023}$ | $0.254_{\pm0.000}$ | $1.72_{\pm0.92}$ | $3.64_{\pm0.90}$ | $2.24_{\pm0.93}$ |
| | TimeWak | $\underline{0.121}_{\pm0.016}$ | $1.977_{\pm0.750}$ | $\mathbf{0.142}_{\pm\mathbf{0.008}}$ | $\underline{0.254}_{\pm0.000}$ | $\mathbf{231.28}_{\pm\mathbf{1.45}}$ | $\mathbf{267.53}_{\pm\mathbf{2.60}}$ | $\mathbf{245.37}_{\pm\mathbf{2.88}}$ |
| fMRI | SpatDDIM | $0.198_{\pm0.023}$ | $2.014_{\pm0.046}$ | $0.139_{\pm0.030}$ | $0.101_{\pm0.001}$ | $90.69_{\pm0.94}$ | $61.12_{\pm0.87}$ | $81.31_{\pm0.70}$ |
| | SpatBDIA | $\mathbf{0.188}_{\pm\mathbf{0.004}}$ | $\mathbf{1.974}_{\pm\mathbf{0.074}}$ | $\underline{0.124}_{\pm0.035}$ | $0.101_{\pm0.000}$ | $76.48_{\pm0.89}$ | $93.74_{\pm0.82}$ | $75.93_{\pm0.66}$ |
| | TempDDIM | $\underline{0.193}_{\pm0.018}$ | $2.097_{\pm0.086}$ | $0.143_{\pm0.020}$ | $\underline{0.101}_{\pm0.000}$ | $\mathbf{381.15}_{\pm\mathbf{0.81}}$ | $\mathbf{617.91}_{\pm\mathbf{0.98}}$ | $\mathbf{828.89}_{\pm\mathbf{1.01}}$ |
| | TimeWak | $0.199_{\pm0.010}$ | $\underline{2.006}_{\pm0.053}$ | $\mathbf{0.122}_{\pm\mathbf{0.033}}$ | $\mathbf{0.100}_{\pm\mathbf{0.000}}$ | $\underline{379.51}_{\pm0.82}$ | $\underline{595.68}_{\pm1.03}$ | $\underline{526.81}_{\pm13.12}$ |

## E.7 Challenging time series datasets

We evaluate `TimeWak` on 2 additional and more challenging real-world datasets, (i) the ILI dataset, which records influenza-like illness cases in the United States, and (ii) the Weather dataset, which is sparse and noisy. Both datasets are standard benchmarks and are used in TimesNet [30]. Results run on these datasets are shown in Table 15, where `TimeWak` achieves robust watermark detectability while preserving the quality of the synthetic data.

Table 15: Results of synthetic time series quality and watermark detectability for 64-length sequences. No watermarking ('W/O') is included. Best results are in **bold**, and second-best are underlined.

| Dataset | Method | Context-FID ↓ | Correlational ↓ | Discriminative ↓ | Predictive ↓ | Z-score ↑ |
|---|---|---|---|---|---|---|
| Illness | W/O | $0.411_{\pm 0.040}$ | $0.073_{\pm 0.061}$ | $0.147_{\pm 0.102}$ | $0.028_{\pm 0.002}$ | - |
| | TR | $1.530_{\pm 0.177}$ | $0.149_{\pm 0.056}$ | $0.286_{\pm 0.096}$ | $0.035_{\pm 0.003}$ | $5.09_{\pm 0.06}$ |
| | GS | $0.734_{\pm 0.030}$ | $0.159_{\pm 0.035}$ | $0.397_{\pm 0.054}$ | $0.030_{\pm 0.001}$ | $\underline{78.18}_{\pm 0.84}$ |
| | HTW | $0.439_{\pm 0.040}$ | $\mathbf{0.069}_{\pm \mathbf{0.041}}$ | $0.217_{\pm 0.060}$ | $0.032_{\pm 0.002}$ | $7.37_{\pm 0.75}$ |
| | TabWak | $\mathbf{0.239}_{\pm \mathbf{0.030}}$ | $\underline{0.070}_{\pm 0.036}$ | $0.131_{\pm 0.132}$ | $\mathbf{0.027}_{\pm \mathbf{0.001}}$ | $-2.06_{\pm 0.88}$ |
| | TabWak$^\top$ | $0.295_{\pm 0.045}$ | $0.071_{\pm 0.035}$ | $\underline{0.114}_{\pm 0.041}$ | $0.028_{\pm 0.002}$ | $21.26_{\pm 1.13}$ |
| | TimeWak | $\underline{0.240}_{\pm 0.009}$ | $0.076_{\pm 0.050}$ | $\mathbf{0.111}_{\pm \mathbf{0.074}}$ | $\underline{0.028}_{\pm 0.001}$ | $\mathbf{151.03}_{\pm \mathbf{1.60}}$ |
| Weather | W/O | $0.647_{\pm 0.079}$ | $1.429_{\pm 0.089}$ | $0.175_{\pm 0.011}$ | $0.002_{\pm 0.000}$ | - |
| | TR | $3.381_{\pm 0.577}$ | $2.518_{\pm 0.039}$ | $0.388_{\pm 0.020}$ | $\mathbf{0.002}_{\pm \mathbf{0.000}}$ | $0.40_{\pm 0.02}$ |
| | GS | $4.495_{\pm 0.804}$ | $2.194_{\pm 0.140}$ | $0.446_{\pm 0.008}$ | $\mathbf{0.002}_{\pm \mathbf{0.000}}$ | $15.36_{\pm 0.92}$ |
| | HTW | $\underline{0.712}_{\pm 0.062}$ | $1.463_{\pm 0.098}$ | $0.190_{\pm 0.013}$ | $\mathbf{0.002}_{\pm \mathbf{0.000}}$ | $4.82_{\pm 0.83}$ |
| | TabWak | $0.751_{\pm 0.088}$ | $1.571_{\pm 0.168}$ | $0.200_{\pm 0.007}$ | $\mathbf{0.002}_{\pm \mathbf{0.000}}$ | $-1.23_{\pm 1.00}$ |
| | TabWak$^\top$ | $\mathbf{0.588}_{\pm \mathbf{0.081}}$ | $\underline{1.369}_{\pm 0.089}$ | $\mathbf{0.178}_{\pm \mathbf{0.012}}$ | $\mathbf{0.002}_{\pm \mathbf{0.000}}$ | $39.58_{\pm 0.89}$ |
| | TimeWak | $0.717_{\pm 0.071}$ | $\mathbf{0.951}_{\pm \mathbf{0.088}}$ | $\underline{0.184}_{\pm 0.007}$ | $\mathbf{0.002}_{\pm \mathbf{0.000}}$ | $\mathbf{205.53}_{\pm \mathbf{1.86}}$ |

## E.8 TPR@0.1%FPR performance

In Figure 6, we present the TPR@0.1%FPR metric against the number of samples across five datasets under 24, 64 and 128 window sizes. In most cases, `TimeWak` consistently outperforms other baselines, such as Gaussian Shading and TabWak$^\top$, by achieving significantly higher TPR values. Notably, `TimeWak` reaches a perfect 1.0 TPR@0.1%FPR in the majority of scenarios, with 7 cases requiring only a single sample and 4 cases needing just 2 samples, demonstrating its strong detectability with minimal data requirements.

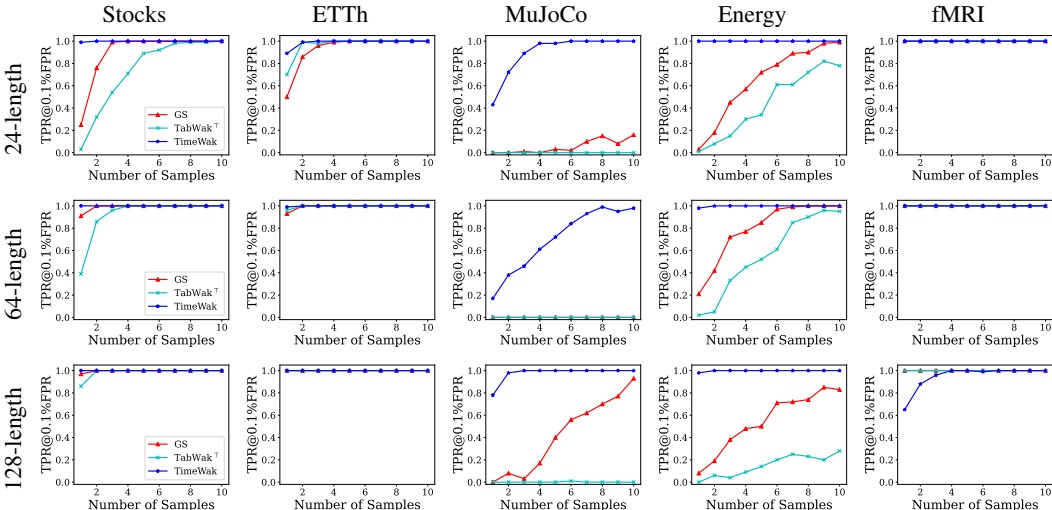

Figure 6: TPR@0.1%FPR against the number of samples across five datasets under different window sizes.

### E.9 TPR@0.1%FPR on mixed dataset

We construct a mixed dataset containing equal proportions (1/3 each) of real data, synthetic data without watermarks, and synthetic data with watermarks, totaling 100 trials with a window length of 64. We evaluate TPR@0.1%FPR with 1, 10, and 20 samples per record, as shown in Table 16. For comparison, we select the methods with the best detectability: GS and TabWak$^\top$. The results demonstrate that `TimeWak` achieves 99 to 100% true positive rates in this mixed data setting when the number of samples is 20. But GS completely fails to detect watermarks in the MuJoCo dataset with 0% TPR across all sample sizes, while TabWak$^\top$ fails on both MuJoCo with 0% TPR and Energy datasets with 0 to 1% TPR.

Table 16: Results of TPR@0.1%FPR on mixed dataset when the number of samples is 1, 10, and 20.

| Dataset | Method | 1 ↑ | 10 ↑ | 20 ↑ |
|---------|--------|-----|------|------|
| Stocks | GS | **0.33** | 0.87 | **0.99** |
| | TabWak$^\top$ | 0.13 | 0.47 | 0.68 |
| | TimeWak | 0.22 | **0.96** | **0.99** |
| ETTh | GS | 0.43 | **0.98** | **1.0** |
| | TabWak$^\top$ | **0.49** | 0.91 | 0.99 |
| | TimeWak | 0.38 | 0.92 | **1.0** |
| MuJoCo | GS | 0.0 | 0.0 | 0.0 |
| | TabWak$^\top$ | 0.0 | 0.0 | 0.0 |
| | TimeWak | **0.34** | **0.84** | **0.99** |
| Energy | GS | **0.42** | **1.0** | **1.0** |
| | TabWak$^\top$ | 0.01 | 0.0 | 0.0 |
| | TimeWak | 0.32 | 0.97 | **1.0** |
| fMRI | GS | **0.4** | 0.98 | **1.0** |
| | TabWak$^\top$ | 0.31 | **1.0** | **1.0** |
| | TimeWak | 0.26 | 0.97 | **1.0** |

### E.10 Preservation of key signal characteristics in watermarked time series data

To assess the preservation of key signal characteristics, we add 4 additional metrics from TSG-Bench [2]: Marginal Distribution Difference (MDD), AutoCorrelation Difference (ACD), Skewness Difference (SD), and Kurtosis Difference (KD). These measures are designed to capture inter-series correlations and temporal dependencies, thereby evaluating how well the generated time series preserves the original characteristics. For all these metrics, the lower the score, the better. As shown in Table 17, `TimeWak` consistently achieves scores very close to the non-watermarked (W/O) baseline across all datasets, indicating minimal distortion introduced by the watermark.

Table 17: Results of Marginal Distribution Difference (MDD), AutoCorrelation Difference (ACD), Skewness Difference (SD), and Kurtosis Difference (KD) for 64-length sequences.

| Dataset | Method | MDD ↓ | ACD ↓ | SD ↓ | KD ↓ |
|---|---|---|---|---|---|
| Stocks | W/O | 0.491 | 0.044 | 0.473 | 1.522 |
| | TR | 0.881 | 0.445 | 0.759 | 4.408 |
| | GS | 1.063 | 0.444 | 0.727 | 4.889 |
| | HTW | 0.491 | **0.044** | 0.473 | 1.522 |
| | TabWak | 0.470 | 0.078 | 0.180 | 0.544 |
| | TabWak$^\top$ | 0.478 | 0.098 | **0.060** | 0.852 |
| | TimeWak | **0.431** | 0.068 | 0.103 | **0.295** |
| ETTh | W/O | 0.176 | 0.421 | 0.255 | 1.359 |
| | TR | 0.546 | 1.128 | 0.633 | 3.474 |
| | GS | 0.712 | 1.581 | 0.723 | 2.725 |
| | HTW | 0.384 | **0.459** | 0.276 | 1.468 |
| | TabWak | 0.211 | 0.532 | 0.245 | 1.422 |
| | TabWak$^\top$ | **0.183** | 0.753 | **0.141** | **0.707** |
| | TimeWak | 0.184 | 0.511 | 0.197 | 1.109 |
| MuJoCo | W/O | 0.379 | 0.225 | 0.062 | 0.306 |
| | TR | 1.296 | 1.494 | 0.426 | 1.555 |
| | GS | 1.500 | 1.550 | 0.533 | 1.221 |
| | HTW | 0.941 | 0.434 | 0.076 | **0.287** |
| | TabWak | 0.420 | **0.262** | 0.087 | 0.311 |
| | TabWak$^\top$ | 0.521 | 0.532 | 0.084 | 0.351 |
| | TimeWak | **0.394** | 0.300 | **0.069** | 0.340 |
| Energy | W/O | 0.188 | 0.157 | 0.112 | 0.703 |
| | TR | 0.558 | 0.606 | 0.355 | 1.681 |
| | GS | 0.660 | 0.958 | 0.373 | 1.015 |
| | HTW | 0.363 | **0.148** | **0.108** | 0.701 |
| | TabWak | 0.235 | 0.270 | 0.126 | 0.618 |
| | TabWak$^\top$ | 0.263 | 0.336 | 0.112 | 0.682 |
| | TimeWak | **0.215** | 0.241 | 0.111 | **0.601** |
| fMRI | W/O | 0.099 | 0.153 | 0.041 | 0.128 |
| | TR | 0.447 | 1.356 | 0.066 | 0.428 |
| | GS | 0.818 | 1.435 | 0.096 | 0.174 |
| | HTW | 0.452 | 0.151 | 0.048 | 0.133 |
| | TabWak | 0.126 | 0.159 | 0.043 | 0.138 |
| | TabWak$^\top$ | 0.127 | 0.261 | **0.040** | **0.114** |
| | TimeWak | **0.120** | **0.148** | **0.040** | 0.132 |

### E.11 Hyperparameter evaluation

### E.11.1 Intervals

Interval, also referred to as $H$, is one of the key hyperparameters in our approach. Based on our experiments in Table 18 (24-length), Table 19 (64-length), and Table 20 (128-length), we found that setting $H = 2$ yields the best results across most datasets. For instance, consider the Stocks dataset, which consists of 6 features and 24 time steps. When $H = 8$, the number of bit templates that can be generated is $2^{(3\times 6)}$, whereas for $H = 2$, the number of bit templates increases significantly to $2^{(12\times 6)}$. A lower $H$ value allows for the generation of a greater number of bit combinations, leading to a more diverse seed distribution. However, as shown in Table 20, for datasets such as fMRI, tuning $H$ can enhance detectability while preserving the quality of the synthetic data. This could be attributed to the inherently noisy nature of the fMRI dataset, where adjusting $H$ helps balance detectability and data fidelity.

Table 18: Results of synthetic time series quality and watermark detectability with different intervals on `TimeWak`. Quality metrics and Z-score are for 24-length sequences.

| Dataset | Interval | Context-FID ↓ | Correlational ↓ | Discriminative ↓ | Predictive ↓ | Z-score ↑ |
|---|---|---|---|---|---|---|
| Stocks | 2 | $0.277_{\pm 0.019}$ | $0.020_{\pm 0.018}$ | $0.120_{\pm 0.039}$ | $0.038_{\pm 0.000}$ | $182.10_{\pm 0.73}$ |
|  | 4 | $0.419_{\pm 0.098}$ | $0.006_{\pm 0.002}$ | $0.162_{\pm 0.022}$ | $0.039_{\pm 0.000}$ | $202.45_{\pm 1.11}$ |
|  | 8 | $1.006_{\pm 0.085}$ | $0.023_{\pm 0.018}$ | $0.197_{\pm 0.015}$ | $0.041_{\pm 0.000}$ | $216.56_{\pm 1.43}$ |
| ETTh | 2 | $0.237_{\pm 0.017}$ | $0.212_{\pm 0.043}$ | $0.102_{\pm 0.014}$ | $0.122_{\pm 0.002}$ | $134.83_{\pm 0.95}$ |
|  | 4 | $0.463_{\pm 0.094}$ | $0.435_{\pm 0.047}$ | $0.113_{\pm 0.018}$ | $0.130_{\pm 0.001}$ | $166.50_{\pm 1.27}$ |
|  | 8 | $0.925_{\pm 0.140}$ | $0.576_{\pm 0.082}$ | $0.150_{\pm 0.014}$ | $0.135_{\pm 0.005}$ | $167.54_{\pm 1.36}$ |
| MuJoCo | 2 | $0.089_{\pm 0.017}$ | $0.532_{\pm 0.137}$ | $0.044_{\pm 0.021}$ | $0.008_{\pm 0.001}$ | $85.69_{\pm 1.08}$ |
|  | 4 | $0.148_{\pm 0.031}$ | $0.739_{\pm 0.086}$ | $0.087_{\pm 0.020}$ | $0.007_{\pm 0.000}$ | $71.95_{\pm 1.36}$ |
|  | 8 | $0.327_{\pm 0.059}$ | $1.179_{\pm 0.168}$ | $0.177_{\pm 0.019}$ | $0.009_{\pm 0.002}$ | $76.90_{\pm 1.29}$ |
| Energy | 2 | $0.121_{\pm 0.016}$ | $1.977_{\pm 0.750}$ | $0.142_{\pm 0.008}$ | $0.254_{\pm 0.000}$ | $231.28_{\pm 1.45}$ |
|  | 4 | $0.186_{\pm 0.017}$ | $3.315_{\pm 0.395}$ | $0.159_{\pm 0.011}$ | $0.254_{\pm 0.000}$ | $266.77_{\pm 1.99}$ |
|  | 8 | $0.363_{\pm 0.093}$ | $5.402_{\pm 0.499}$ | $0.184_{\pm 0.024}$ | $0.255_{\pm 0.000}$ | $279.40_{\pm 1.90}$ |
| fMRI | 2 | $0.199_{\pm 0.010}$ | $2.006_{\pm 0.053}$ | $0.122_{\pm 0.033}$ | $0.100_{\pm 0.000}$ | $379.51_{\pm 0.82}$ |
|  | 4 | $0.191_{\pm 0.013}$ | $2.117_{\pm 0.124}$ | $0.125_{\pm 0.034}$ | $0.101_{\pm 0.000}$ | $464.70_{\pm 0.92}$ |
|  | 8 | $0.204_{\pm 0.021}$ | $2.354_{\pm 0.110}$ | $0.171_{\pm 0.043}$ | $0.103_{\pm 0.000}$ | $506.95_{\pm 1.10}$ |

Table 19: Results of synthetic time series quality and watermark detectability with different intervals on `TimeWak`. Quality metrics and Z-score are for 64-length sequences.

| Dataset | Interval | Context-FID ↓ | Correlational ↓ | Discriminative ↓ | Predictive ↓ | Z-score ↑ |
|---|---|---|---|---|---|---|
| Stocks | 2 | $0.387_{\pm 0.054}$ | $0.017_{\pm 0.017}$ | $0.092_{\pm 0.041}$ | $0.037_{\pm 0.000}$ | $395.34_{\pm 1.24}$ |
|  | 4 | $0.466_{\pm 0.099}$ | $0.021_{\pm 0.006}$ | $0.077_{\pm 0.025}$ | $0.037_{\pm 0.000}$ | $397.68_{\pm 1.31}$ |
|  | 8 | $1.053_{\pm 0.099}$ | $0.017_{\pm 0.013}$ | $0.155_{\pm 0.037}$ | $0.037_{\pm 0.000}$ | $406.56_{\pm 1.77}$ |
| ETTh | 2 | $0.297_{\pm 0.038}$ | $0.133_{\pm 0.040}$ | $0.097_{\pm 0.015}$ | $0.115_{\pm 0.003}$ | $236.08_{\pm 1.63}$ |
|  | 4 | $0.514_{\pm 0.027}$ | $0.335_{\pm 0.040}$ | $0.098_{\pm 0.031}$ | $0.119_{\pm 0.008}$ | $272.70_{\pm 1.79}$ |
|  | 8 | $0.911_{\pm 0.048}$ | $0.540_{\pm 0.030}$ | $0.147_{\pm 0.038}$ | $0.123_{\pm 0.008}$ | $268.14_{\pm 2.00}$ |
| MuJoCo | 2 | $0.108_{\pm 0.014}$ | $0.413_{\pm 0.062}$ | $0.038_{\pm 0.021}$ | $0.007_{\pm 0.001}$ | $56.45_{\pm 1.26}$ |
|  | 4 | $0.205_{\pm 0.024}$ | $0.522_{\pm 0.024}$ | $0.088_{\pm 0.031}$ | $0.007_{\pm 0.002}$ | $58.75_{\pm 1.18}$ |
|  | 8 | $0.338_{\pm 0.068}$ | $0.768_{\pm 0.077}$ | $0.159_{\pm 0.020}$ | $0.007_{\pm 0.001}$ | $68.29_{\pm 1.19}$ |
| Energy | 2 | $0.143_{\pm 0.019}$ | $1.662_{\pm 0.298}$ | $0.145_{\pm 0.019}$ | $0.251_{\pm 0.000}$ | $267.53_{\pm 2.60}$ |
|  | 4 | $0.195_{\pm 0.017}$ | $2.760_{\pm 0.504}$ | $0.150_{\pm 0.011}$ | $0.252_{\pm 0.000}$ | $289.07_{\pm 2.77}$ |
|  | 8 | $0.407_{\pm 0.087}$ | $4.285_{\pm 0.229}$ | $0.170_{\pm 0.025}$ | $0.252_{\pm 0.000}$ | $345.97_{\pm 3.48}$ |
| fMRI | 2 | $0.441_{\pm 0.035}$ | $1.786_{\pm 0.043}$ | $0.314_{\pm 0.041}$ | $0.100_{\pm 0.000}$ | $595.68_{\pm 1.03}$ |
|  | 4 | $0.425_{\pm 0.027}$ | $1.834_{\pm 0.122}$ | $0.273_{\pm 0.076}$ | $0.100_{\pm 0.000}$ | $749.10_{\pm 1.08}$ |
|  | 8 | $0.469_{\pm 0.027}$ | $1.823_{\pm 0.065}$ | $0.294_{\pm 0.141}$ | $0.100_{\pm 0.000}$ | $817.23_{\pm 1.20}$ |

Table 20: Results of synthetic time series quality and watermark detectability with different intervals on `TimeWak`. Quality metrics and Z-score are for 128-length sequences.

| Dataset | Interval | Context-FID ↓ | Correlational ↓ | Discriminative ↓ | Predictive ↓ | Z-score ↑ |
|---|---|---|---|---|---|---|
| Stocks | 2 | $0.316_{\pm 0.044}$ | $0.021_{\pm 0.024}$ | $0.140_{\pm 0.029}$ | $0.037_{\pm 0.000}$ | $550.05_{\pm 1.18}$ |
| | 4 | $0.410_{\pm 0.104}$ | $0.019_{\pm 0.017}$ | $0.140_{\pm 0.067}$ | $0.037_{\pm 0.000}$ | $571.54_{\pm 1.22}$ |
| | 8 | $1.132_{\pm 0.551}$ | $0.035_{\pm 0.021}$ | $0.217_{\pm 0.019}$ | $0.038_{\pm 0.000}$ | $599.41_{\pm 1.39}$ |
| ETTh | 2 | $1.090_{\pm 0.100}$ | $0.135_{\pm 0.057}$ | $0.174_{\pm 0.007}$ | $0.110_{\pm 0.009}$ | $340.36_{\pm 2.06}$ |
| | 4 | $1.445_{\pm 0.119}$ | $0.333_{\pm 0.034}$ | $0.173_{\pm 0.026}$ | $0.114_{\pm 0.003}$ | $374.69_{\pm 2.39}$ |
| | 8 | $1.838_{\pm 0.099}$ | $0.497_{\pm 0.054}$ | $0.173_{\pm 0.021}$ | $0.116_{\pm 0.003}$ | $392.83_{\pm 2.39}$ |
| MuJoCo | 2 | $0.155_{\pm 0.016}$ | $0.316_{\pm 0.022}$ | $0.046_{\pm 0.030}$ | $0.005_{\pm 0.001}$ | $123.36_{\pm 1.43}$ |
| | 4 | $0.172_{\pm 0.038}$ | $0.410_{\pm 0.057}$ | $0.083_{\pm 0.011}$ | $0.006_{\pm 0.000}$ | $178.16_{\pm 1.60}$ |
| | 8 | $0.330_{\pm 0.077}$ | $0.685_{\pm 0.055}$ | $0.124_{\pm 0.026}$ | $0.006_{\pm 0.002}$ | $170.77_{\pm 1.93}$ |
| Energy | 2 | $0.148_{\pm 0.027}$ | $1.687_{\pm 0.328}$ | $0.140_{\pm 0.057}$ | $0.249_{\pm 0.000}$ | $245.37_{\pm 2.88}$ |
| | 4 | $0.261_{\pm 0.025}$ | $3.246_{\pm 0.345}$ | $0.114_{\pm 0.030}$ | $0.250_{\pm 0.001}$ | $354.38_{\pm 2.32}$ |
| | 8 | $0.506_{\pm 0.114}$ | $4.550_{\pm 0.745}$ | $0.147_{\pm 0.009}$ | $0.251_{\pm 0.000}$ | $395.20_{\pm 2.91}$ |
| fMRI | 2 | $0.855_{\pm 0.072}$ | $1.704_{\pm 0.060}$ | $0.298_{\pm 0.227}$ | $0.100_{\pm 0.000}$ | $526.81_{\pm 13.12}$ |
| | 4 | $0.884_{\pm 0.124}$ | $1.708_{\pm 0.050}$ | $0.348_{\pm 0.201}$ | $0.100_{\pm 0.000}$ | $1022.29_{\pm 1.75}$ |
| | 8 | $0.866_{\pm 0.075}$ | $1.698_{\pm 0.050}$ | $0.345_{\pm 0.195}$ | $0.100_{\pm 0.000}$ | $1097.78_{\pm 1.57}$ |

### E.11.2 Bits

We perform an empirical validation of the expected bit accuracy using simulations. Specifically, we set the synthetic time series data size as 24 time steps (window length) and 10 features, and evaluate the watermarking and detection pipeline using `TimeWak`. In this experiment, we intentionally omit both the forward diffusion and reverse (inversion) diffusion processes to focus solely on the effect of noise during reconstruction. Instead, we simulate the reconstruction noise directly by adding noise to the clean initial noise.

We generate initial samples using our watermarking method and simulate reconstruction error by adding noise with feature-specific means sampled from $\mathcal{N}(0, 5)$ and a shared variance $\sigma$. This results in a noise distribution of $\mathcal{N}(\mu_f, \sigma)$ per feature, where $\mu_f \sim \mathcal{N}(0, 5)$. We then apply watermark detection to the perturbed samples and compute the average bit accuracy.

We run the simulation using 100,000 samples, grouped into trials of 2,000 samples each. This process is repeated across 50 independent rounds to compute the average bit accuracy. Figure 7 shows the results we get. We observe that a larger $L$ leads to higher bit accuracy, indicating better detectability. In addition, we evaluate a "transposed" version of `TimeWak`, denoted as `TimeWak`$^\top$, where the chained hash is applied along the feature dimension instead of the time axis. We find that the bit accuracy of this variant remains close to 0.5 and is significantly lower than that of the original `TimeWak`. This simulation further validates the importance of applying the watermark along the time axis, rather than across features, to ensure reliable detection.

Additionally, we present the values of bit-length $L$ used across different experiments in Table 21–23. For bit-lengths greater than 2, we applied the valid bit mechanism from [41]. In general, a larger $L$ tends to improve watermark detectability. This is because, during bit accuracy calculation, a larger $L$ places more emphasis on the tail bits, which are less likely to be affected by reconstruction errors or noise. However, increasing $L$ also leads to lower sample quality, as it involves modifying more of the initial noise, making it deviate further from a standard Gaussian distribution. Our results show this trade-off holds across most scenarios, with the exception of the Stocks dataset.

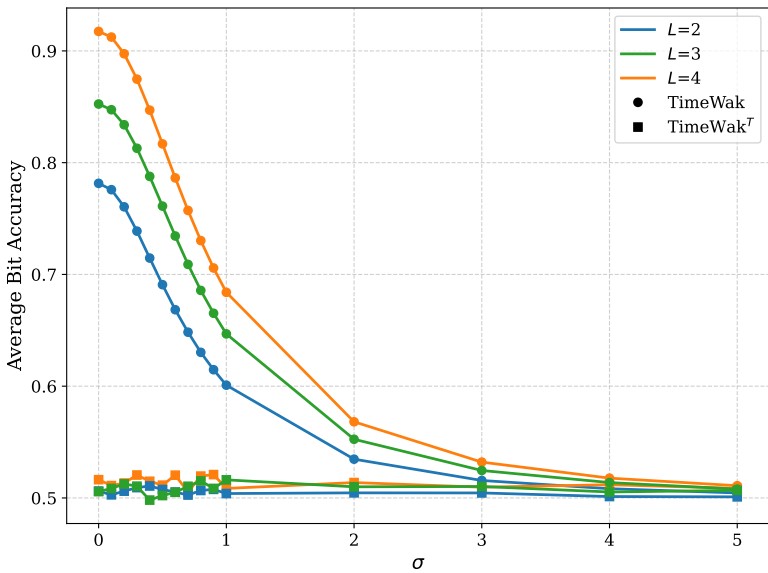

Figure 7: Average bit accuracy of different bit-length $L$.

Table 21: Results of synthetic time series quality and watermark detectability with different bits on `TimeWak`. Quality metrics and Z-score are for 24-length sequences.

| Dataset | Bit | Context-FID ↓ | Correlational ↓ | Discriminative ↓ | Predictive ↓ | Z-score ↑ |
|---------|-----|---------------|-----------------|------------------|--------------|-----------|
| Stocks | 2 | $0.277_{\pm0.019}$ | $0.020_{\pm0.018}$ | $0.120_{\pm0.039}$ | $0.038_{\pm0.000}$ | $182.10_{\pm0.73}$ |
| | 3 | $0.214_{\pm0.039}$ | $0.024_{\pm0.019}$ | $0.130_{\pm0.033}$ | $0.038_{\pm0.000}$ | $194.87_{\pm0.56}$ |
| | 4 | $0.328_{\pm0.110}$ | $0.023_{\pm0.026}$ | $0.155_{\pm0.027}$ | $0.038_{\pm0.000}$ | $182.58_{\pm0.33}$ |
| ETTh | 2 | $0.237_{\pm0.017}$ | $0.212_{\pm0.043}$ | $0.102_{\pm0.014}$ | $0.122_{\pm0.002}$ | $134.83_{\pm0.95}$ |
| | 3 | $0.211_{\pm0.020}$ | $0.206_{\pm0.031}$ | $0.093_{\pm0.003}$ | $0.124_{\pm0.003}$ | $162.25_{\pm0.89}$ |
| | 4 | $0.238_{\pm0.025}$ | $0.225_{\pm0.043}$ | $0.095_{\pm0.016}$ | $0.124_{\pm0.001}$ | $149.74_{\pm0.66}$ |
| MuJoCo | 2 | $0.089_{\pm0.017}$ | $0.532_{\pm0.137}$ | $0.044_{\pm0.021}$ | $0.008_{\pm0.001}$ | $85.69_{\pm1.08}$ |
| | 3 | $0.092_{\pm0.022}$ | $0.520_{\pm0.105}$ | $0.054_{\pm0.014}$ | $0.008_{\pm0.001}$ | $73.09_{\pm1.29}$ |
| | 4 | $0.099_{\pm0.019}$ | $0.524_{\pm0.079}$ | $0.056_{\pm0.013}$ | $0.007_{\pm0.000}$ | $67.67_{\pm1.23}$ |
| Energy | 2 | $0.121_{\pm0.016}$ | $1.977_{\pm0.750}$ | $0.142_{\pm0.008}$ | $0.254_{\pm0.000}$ | $231.28_{\pm1.45}$ |
| | 3 | $0.121_{\pm0.014}$ | $1.799_{\pm0.395}$ | $0.156_{\pm0.023}$ | $0.254_{\pm0.001}$ | $268.24_{\pm1.69}$ |
| | 4 | $0.143_{\pm0.015}$ | $1.721_{\pm0.347}$ | $0.155_{\pm0.010}$ | $0.254_{\pm0.000}$ | $269.83_{\pm1.60}$ |
| fMRI | 2 | $0.199_{\pm0.010}$ | $2.006_{\pm0.053}$ | $0.122_{\pm0.033}$ | $0.100_{\pm0.000}$ | $379.51_{\pm0.82}$ |
| | 3 | $0.195_{\pm0.008}$ | $1.987_{\pm0.076}$ | $0.113_{\pm0.031}$ | $0.101_{\pm0.000}$ | $456.02_{\pm0.67}$ |
| | 4 | $0.183_{\pm0.012}$ | $2.032_{\pm0.030}$ | $0.111_{\pm0.026}$ | $0.101_{\pm0.000}$ | $440.88_{\pm0.55}$ |

Table 22: Results of synthetic time series quality and watermark detectability with different bits on `TimeWak`. Quality metrics and Z-score are for 64-length sequences.

| Dataset | Bit | Context-FID ↓ | Correlational ↓ | Discriminative ↓ | Predictive ↓ | Z-score ↑ |
|---|---|---|---|---|---|---|
| Stocks | 2 | $0.387_{\pm0.054}$ | $0.017_{\pm0.017}$ | $0.092_{\pm0.041}$ | $0.037_{\pm0.000}$ | $395.34_{\pm1.24}$ |
| | 3 | $0.312_{\pm0.046}$ | $0.014_{\pm0.006}$ | $0.121_{\pm0.010}$ | $0.037_{\pm0.000}$ | $334.15_{\pm0.49}$ |
| | 4 | $0.251_{\pm0.053}$ | $0.014_{\pm0.018}$ | $0.095_{\pm0.022}$ | $0.037_{\pm0.000}$ | $309.59_{\pm0.33}$ |
| ETTh | 2 | $0.297_{\pm0.038}$ | $0.133_{\pm0.040}$ | $0.097_{\pm0.015}$ | $0.115_{\pm0.003}$ | $236.08_{\pm1.63}$ |
| | 3 | $0.369_{\pm0.043}$ | $0.182_{\pm0.036}$ | $0.102_{\pm0.013}$ | $0.117_{\pm0.003}$ | $249.67_{\pm1.41}$ |
| | 4 | $0.365_{\pm0.031}$ | $0.185_{\pm0.030}$ | $0.102_{\pm0.010}$ | $0.113_{\pm0.007}$ | $261.13_{\pm1.20}$ |
| MuJoCo | 2 | $0.108_{\pm0.014}$ | $0.413_{\pm0.062}$ | $0.038_{\pm0.021}$ | $0.007_{\pm0.001}$ | $56.45_{\pm1.26}$ |
| | 3 | $0.136_{\pm0.012}$ | $0.423_{\pm0.051}$ | $0.073_{\pm0.018}$ | $0.007_{\pm0.002}$ | $84.07_{\pm1.48}$ |
| | 4 | $0.126_{\pm0.017}$ | $0.381_{\pm0.063}$ | $0.036_{\pm0.030}$ | $0.007_{\pm0.001}$ | $96.73_{\pm1.45}$ |
| Energy | 2 | $0.143_{\pm0.019}$ | $1.662_{\pm0.298}$ | $0.145_{\pm0.019}$ | $0.251_{\pm0.000}$ | $267.53_{\pm2.60}$ |
| | 3 | $0.182_{\pm0.047}$ | $1.284_{\pm0.400}$ | $0.165_{\pm0.019}$ | $0.251_{\pm0.000}$ | $323.04_{\pm2.37}$ |
| | 4 | $0.143_{\pm0.010}$ | $1.460_{\pm0.354}$ | $0.152_{\pm0.023}$ | $0.251_{\pm0.000}$ | $322.38_{\pm2.25}$ |
| fMRI | 2 | $0.441_{\pm0.035}$ | $1.786_{\pm0.043}$ | $0.314_{\pm0.041}$ | $0.100_{\pm0.000}$ | $595.68_{\pm1.03}$ |
| | 3 | $0.423_{\pm0.024}$ | $1.782_{\pm0.082}$ | $0.216_{\pm0.175}$ | $0.100_{\pm0.000}$ | $724.69_{\pm0.85}$ |
| | 4 | $0.440_{\pm0.027}$ | $1.783_{\pm0.033}$ | $0.256_{\pm0.106}$ | $0.100_{\pm0.000}$ | $712.67_{\pm0.59}$ |

Table 23: Results of synthetic time series quality and watermark detectability with different bits on `TimeWak`. Quality metrics and Z-score are for 128-length sequences.

| Dataset | Bit | Context-FID ↓ | Correlational ↓ | Discriminative ↓ | Predictive ↓ | Z-score ↑ |
|---|---|---|---|---|---|---|
| Stocks | 2 | $0.316_{\pm0.044}$ | $0.021_{\pm0.024}$ | $0.140_{\pm0.029}$ | $0.037_{\pm0.000}$ | $550.05_{\pm1.18}$ |
| | 3 | $0.380_{\pm0.059}$ | $0.019_{\pm0.015}$ | $0.176_{\pm0.046}$ | $0.037_{\pm0.000}$ | $459.23_{\pm0.45}$ |
| | 4 | $0.391_{\pm0.087}$ | $0.017_{\pm0.020}$ | $0.134_{\pm0.060}$ | $0.037_{\pm0.000}$ | $427.53_{\pm0.23}$ |
| ETTh | 2 | $1.090_{\pm0.100}$ | $0.135_{\pm0.057}$ | $0.174_{\pm0.007}$ | $0.110_{\pm0.009}$ | $340.36_{\pm2.06}$ |
| | 3 | $1.111_{\pm0.137}$ | $0.151_{\pm0.040}$ | $0.153_{\pm0.010}$ | $0.118_{\pm0.005}$ | $352.26_{\pm1.63}$ |
| | 4 | $1.173_{\pm0.131}$ | $0.233_{\pm0.058}$ | $0.166_{\pm0.013}$ | $0.113_{\pm0.006}$ | $362.55_{\pm1.39}$ |
| MuJoCo | 2 | $0.155_{\pm0.016}$ | $0.316_{\pm0.022}$ | $0.046_{\pm0.030}$ | $0.005_{\pm0.001}$ | $123.36_{\pm1.43}$ |
| | 3 | $0.183_{\pm0.029}$ | $0.317_{\pm0.068}$ | $0.062_{\pm0.011}$ | $0.005_{\pm0.000}$ | $183.47_{\pm1.55}$ |
| | 4 | $0.150_{\pm0.013}$ | $0.349_{\pm0.028}$ | $0.051_{\pm0.031}$ | $0.006_{\pm0.001}$ | $174.45_{\pm1.55}$ |
| Energy | 2 | $0.148_{\pm0.027}$ | $1.687_{\pm0.328}$ | $0.140_{\pm0.057}$ | $0.249_{\pm0.000}$ | $245.37_{\pm2.88}$ |
| | 3 | $0.230_{\pm0.037}$ | $1.154_{\pm0.446}$ | $0.166_{\pm0.054}$ | $0.249_{\pm0.001}$ | $380.35_{\pm2.49}$ |
| | 4 | $0.167_{\pm0.014}$ | $1.700_{\pm0.524}$ | $0.168_{\pm0.069}$ | $0.249_{\pm0.001}$ | $389.26_{\pm2.01}$ |
| fMRI | 2 | $0.855_{\pm0.072}$ | $1.704_{\pm0.060}$ | $0.298_{\pm0.227}$ | $0.100_{\pm0.000}$ | $526.81_{\pm13.12}$ |
| | 3 | $0.819_{\pm0.010}$ | $1.688_{\pm0.049}$ | $0.374_{\pm0.193}$ | $0.100_{\pm0.000}$ | $986.17_{\pm1.08}$ |
| | 4 | $0.828_{\pm0.053}$ | $1.713_{\pm0.020}$ | $0.336_{\pm0.209}$ | $0.100_{\pm0.000}$ | $967.53_{\pm0.86}$ |

### E.12 Watermark detection overhead

To evaluate the practical feasibility of real-time deployment, we measure the watermark detection overhead for `TimeWak` using a single NVIDIA L40S GPU and Intel(R) Xeon(R) Platinum 8562Y+ CPU. Table 24 presents the computational overhead for watermark detection across different datasets and configurations. The results demonstrate that detection overhead remains consistently low, ranging from approximately 1.5 to 7.5 seconds depending on the dataset complexity and batch size. We consider this overhead acceptable for streaming scenarios, particularly given the security benefits provided by the watermarking system. Adapting the hashing mechanism to handle variable-length sequences without padding represents a promising direction for future research that could further enhance the method's applicability to diverse real-world scenarios.

Table 24: Watermark detection overhead in seconds for `TimeWak` when the batch size is 1 and 100.

| Dataset | Window Size | 1 | 100 |
|---------|-------------|------|------|
| Stocks  | 24          | 1.58 | 2.09 |
|         | 64          | 1.50 | 2.27 |
|         | 128         | 1.55 | 2.59 |
| ETTh    | 24          | 1.72 | 2.23 |
|         | 64          | 1.64 | 2.41 |
|         | 128         | 1.63 | 2.62 |
| MuJoCo  | 24          | 2.96 | 3.84 |
|         | 64          | 2.90 | 4.25 |
|         | 128         | 2.90 | 4.49 |
| Energy  | 24          | 3.89 | 5.22 |
|         | 64          | 3.88 | 5.73 |
|         | 128         | 3.89 | 6.58 |
| fMRI    | 24          | 4.69 | 6.31 |
|         | 64          | 4.75 | 7.00 |
|         | 128         | 4.81 | 7.47 |

# F Synthetic samples

Figures 8–12 show synthetic time series generated unconditionally by Diffusion-TS with/without watermark embedding. Each figure corresponds to one of the following datasets: Stocks, ETTh, MuJoCo, Energy, and fMRI. Within each figure, the columns represent the following algorithms: no watermark, `TimeWak`, TabWak, Gaussian Shading, and Tree-Ring watermarks. Up to 4 features are randomly selected from each dataset.

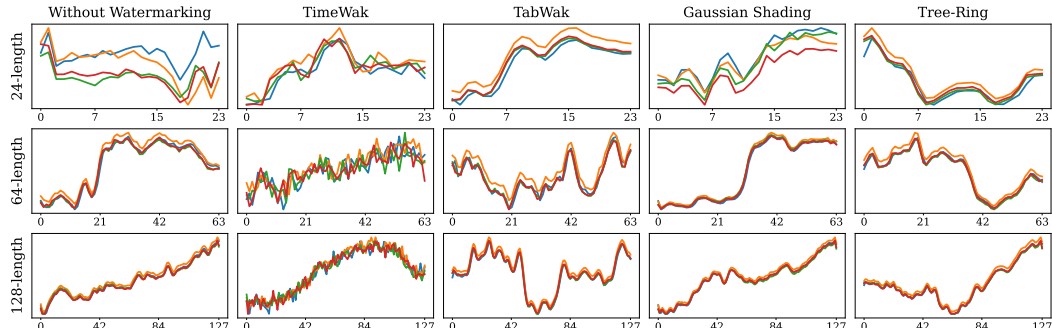

Figure 8: Non-watermarked (leftmost column) and watermarked (remaining columns) time series generated by `TimeWak`, TabWak, Gaussian Shading, and Tree-Ring watermarking for the Stocks dataset.

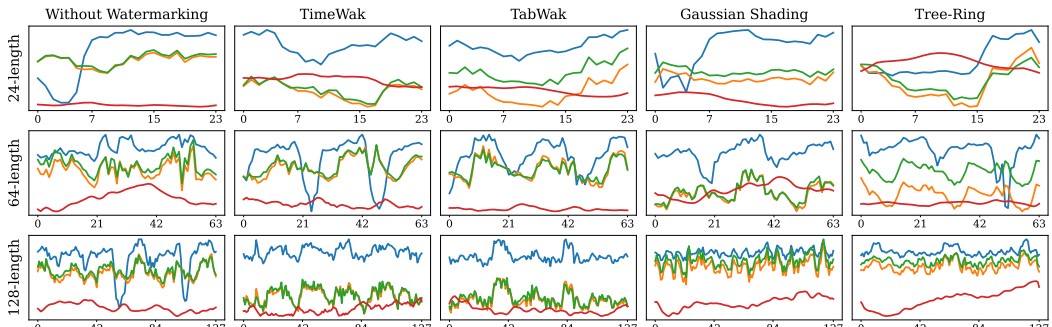

Figure 9: Non-watermarked (leftmost column) and watermarked (remaining columns) time series generated by `TimeWak`, TabWak, Gaussian Shading, and Tree-Ring watermarking for the ETTh dataset.

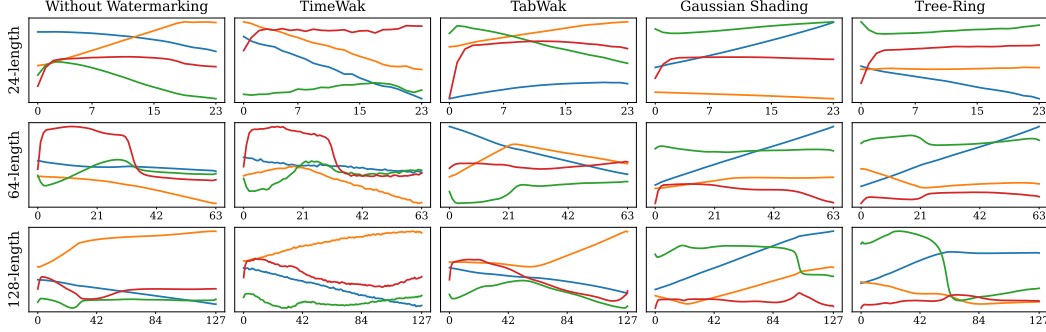

Figure 10: Non-watermarked (leftmost column) and watermarked (remaining columns) time series generated by `TimeWak`, TabWak, Gaussian Shading, and Tree-Ring watermarking for the MuJoCo dataset.

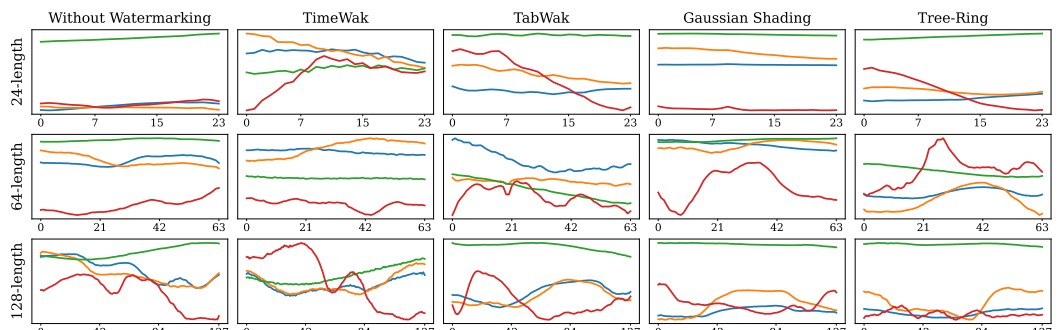

Figure 11: Non-watermarked (leftmost column) and watermarked (remaining columns) time series generated by `TimeWak`, TabWak, Gaussian Shading, and Tree-Ring watermarking for the Energy dataset.

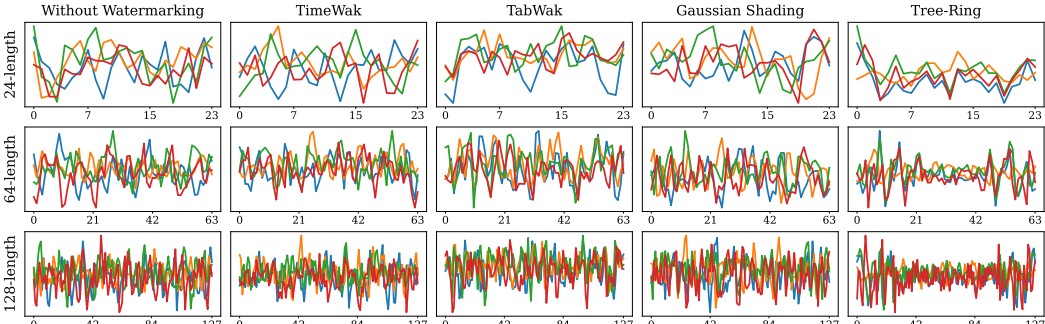

Figure 12: Non-watermarked (leftmost column) and watermarked (remaining columns) time series generated by `TimeWak`, TabWak, Gaussian Shading, and Tree-Ring watermarking for the fMRI dataset.

