# OpenReview forum: "TimeWak: Temporal Chained-Hashing Watermark for Time Series Data"
_NeurIPS.cc/2025/Conference — NeurIPS 2025 spotlight_

### Official Review · Reviewer_zoPw · 2025-06-11

**Clarity:** 3
**Significance:** 4
**Originality:** 4
**Rating:** 5
**Confidence:** 4

**Summary:**

The authors prpose TimeWAK. This is a new watermark for time-series data that is made via diffusion generative models. While existing works focus on watermarking in the latent space, TimeWAK performs the watermarking embedding and detection in the ambient space (referred to as 'real/smaple' space in the paper). The paper offers a combination of several novel ideas that improve upon existing mechanisms and aims to close the current gap in diffusion-based watermarking. Those include 'temporal chained hashing', feature-level shuffling of seeds and $\epsilon$-exact inversion. The authors propose an error bound on the inversion procedure. TimeWAK offers an improvement on the detection-quality tradeoff compared to existing SotA.

**Questions:**

Below are both questions and additional remarks on the paper-
- Figure 2 aims to demonstrate the heterogenity of the reconstruction errors, but I don't understand how is this implied from the figure. Can the authors elaborate on that?
- time series is in some parts written as 'timeseries' (e.g. lines 105, 108) which is inconsistent with the rest of the paper.
- I think the terms 'sampling time' and 'real-space' are not necessarilty the common practice and may be confused with other terms (e.g. from Fourier analysis/signal processing) - I would suggest clarifying them.
- Appendix B.3 - there are two notations $\epsilon_{\theta}(x_t,t)$ and $\epsilon_{\theta(x_t,t)}$ which I believe refer to the same term. Can the authors either explain the difference or fix the typo?
- line 47 ' we self-hash the seeds along the temporal axis...' - is 'self-hash' a common term? can the authors explain it?
-  In the current state of the paper, I would add an emphasis in the beginning of the overview that this is just an overview in a high level, followed by a deep explanation of each component. Reading through it did not feel clear to me.
- $S$ is said to be sampled from an $(0,L-1)$ uniform distribution but in this paper $S$ is only a single bit, so $L=2$? I am not sure this a 100% percent clear.
- What is the reasoning behind the specific choice within the inverse Gaussian CDF in eqn (5)?
- There is a slight point that is not clear to me. Most of the paper focuses on diffusion in terms of transition from $x_T$ to $x_0$  via $(x_t)$ for various values of $t$, and vice versa, while some parts talk about the pair $(x_0,x_1)$. Does $x_T=x_1$? otherwise, what is the relation between them?
- Is the bit accuracy (eqn 15) similarly calculated when $s$ is not binary?
- The statistical metrics that are reported are a little off - $Z$ values that are so big are not a convention ($Z$ around 100 corresponds to a $p$-value of 1e-2174, which is not quite odd) - this should be discussed.
- The fact that an attack increases the $z$-score of your method (increasing 'separability') is quite odd - can the authors further elaborate on this phenomena?

**Ethical Concerns:**

["NO or VERY MINOR ethics concerns only"]

**Final Justification:**

I find this to be a good work and highly relevant. I have engaged in discussion with the authors and feel that my concerns were clarified. Following our discussion and after reading the other reviews, I have decided to keep my score.

**Limitations:**

Yes.

**Paper Formatting Concerns:**

I did not notice any formatting issues.

**Quality:**

3

**Strengths And Weaknesses:**

I think this is a good contribution to the body of work on watermarking. Here is my assessment of the strengths and weaknesses of the paper:

**Strengths:**
1. The paper is well written. The concepts are clear and were made rather intuitive. The related works section is very well written, the steps of the method are clear.
2. The method is novel and aims at the current gap in the literature. novel and interesting tools are developed to close this gap.
3. Theoretical error bounds are provided for $\epsilon$-inversion.
4. The method is rigorously evaluated across several benchmarks and various settings. The authors compare their methods to various existing SotA methods.

**Weakness:**
1. The provided error bound lacks reasoning - it is not clear how tight it is, what are the roles of the parameters, and most importantly, how reasonable is the Lipschitz assumption compared what what kind of neural nets are actually employed.
2. A crisp summary of the results and conclusions from them is missing. The results section is dense with information that can be made much easier to navigate.

---

> ### Author Rebuttal · Authors · 2025-07-31
>
> Thank you for your valuable comments and positive feedback. It's helpful for us to futher improve this work.
>
> ### Weaknesses
> **[W1] Tightness of the bound.**
>
> [WA1] The parameters are summarized in Appendix A. The error bound in Thm. 3.1 shows how an initial error $\epsilon$ propagates, governed by diffusion parameters ($a_{t}$, $b_{t}$) and the noise estimator's Lipschitz constant ($\Delta_{t}$). The Lipschitz assumption is standard for the neural networks used. Crucially, we empirically validate this assumption in Appendix C (Fig. 5, Lines 559-561). We computed $\Delta_{t}$ on 10,000 samples for four datasets and found it behaves as expected from prior work: it starts large (28-80) but rapidly decreases to <1 as the timestep increases. And we also added an experiments by generating 10,000 samples (length 64) across five datasets, computing $x_0$ (final output) and $x_1$ (one step prior). Results in Table A show consistently small $L_1$ norms between $x_1$ and $x_0$ (avg: 5.1-7.0e-03, max: <0.23), validating our $\epsilon$-exact approximation. These results demonstrate the empirical value of $\epsilon$ in our proof.
>
> *Table A: $L_1$ norms between $x_1$ and $x_0$ for 64-length sequences over 10,000 samples.*
> Dataset|Avg. $L_1$|Max. $L_1$
> ---|---|---
> Stocks|7.031e-03|0.082
> ETTh|6.776e-03|0.104
> MuJoCo|5.146e-03|0.081
> Energy|5.687e-03|0.230
> fMRI|5.945e-03|0.070
>
> **[W2] Crisp summary of results.**
>
> [WA2] Thank you for the suggestion. We will add it in the next version.
>
> ### Questions
> **[Q1] Elaborate Figure 2.**
>
> [QA1] The detection of our watermark relies on accurate reconstruction through the inverse diffusion process; lower reconstruction error leads to better watermark detection. Figure 2 reveals a critical asymmetry: reconstruction errors vary dramatically across different features (plots a, c), while remaining much more uniform across time (plots b, d). This demonstrates spatial heterogeneity where reconstruction difficulty differs significantly between feature types. This spatial heterogeneity invalidates prior methods like TabWak that rely on feature-wise seed cloning. TabWak assumes that cloned watermark seeds across features will reconstruct identically, enabling detection through seeds comparison. However, the non-uniform reconstruction errors shown in Figure 2 cause cloned seeds to reconstruct differently across features, leading to low bit accuracy and failed detection. This fundamental limitation motivates our temporal chained-hashing approach: instead of comparing seeds across the problematic feature dimension, we chain and verify seeds along the more stable temporal dimension where reconstruction errors are uniform, ensuring reliable watermark detection despite spatial heterogeneity.
>
> **[Q2] "time series" written as "timeseries".**
>
> [QA2] We will fix it in the next version.
>
> **[Q3] Clarify the terms "sampling time" and "real-space".**
>
> [QA3] We use "sampling time" to refer to the generation phase of diffusion models, distinct from the signal processing definition of temporal intervals between measurements. To clarify, we will use "generation-time watermarking."
>
> Our "real-space" terminology indicates watermarking in the original time series data rather than in latent representations, analogous to spatial versus frequency domains in Fourier analysis. We will use "data-space watermarking" or "native-space watermarking" in next version.
>
> **[Q4] Notation issue in Appendix B.3.**
>
> [QA4] Yes, that is a typo, we will fix it in the next version.
>
> **[Q5] Clarify the term "self-hash".**
>
> [QA5] It should be "chained-hash", which is explained in Section 3.2.1, we will fix it in the next version.
>
> **[Q6] Add an emphasis in the beginning of the overview.**
>
> [QA6] We will add it in the next version.
>
> **[Q7] Clarify $L=2$.**
>
> [QA7] This is correct. In the experiments of the main article, we show results in $L=2$. However, other values of $L$ are feasible, such as $L=3$ or $4$. We include these possibilities in our results in Tables 16-18 in Appendix E.7.2. While these show promising results, $L=2$ yields the best results in most cases.
>
> **[Q8] Reason behind the specific choice within the inverse Gaussian CDF in eqn(5).**
>
> [QA8] The core intuition behind Equation (5) is to control Gaussian noise generation for watermarking while preserving the original $\mathcal{N}(0,1)$ distribution that diffusion models expect. Instead of generating completely random noise, we guide the sampling process: the watermark seed $s^{w,f}$ determines which "region" of the Gaussian distribution we sample from, while still maintaining overall Gaussian properties. The inverse CDF $\Phi^{-1}(u + s^{w,f}/L)$ partitions the distribution into $L$ equal-probability bins, ensuring each watermark bit steers noise generation toward specific quantile ranges without changing the fundamental distribution shape.
>
> This approach preserves generation quality because the noise remains genuinely Gaussian; we're simply controlling which part of the distribution each sample comes from based on our watermark. During detection, we reverse this process by checking which quantile bin the recovered noise falls into, revealing the original watermark seed. Alternative approaches like directly adding watermark signals to noise would visibly distort the Gaussian distribution and degrade sample quality, while our method maintains the statistical properties diffusion models rely on for high-quality generation.
>
> **[Q9] Clarify the notation $(x_0, x_1)$.**
>
> [QA9] We appreciate this clarification request about our notation. In standard diffusion models, $x_0$ and $x_1$ are distinct states in the forward diffusion process: $x_0$ represents the clean data (generated time series), while $x_1$ is the result after one forward diffusion step with small noise added and is different from $x_T$. $x_T$ represents the fully diffused state, approximating pure noise, after all $T$ forward steps are completed.
>
> The confusion arises in our $\epsilon$-exact inversion discussion (Section 3.2.2 and Theorem 3.1). Standard BDIA inversion requires knowing both $x_0$ and $x_1$ to compute exact reverse steps, but in practice we only observe the final generated sample $x_0$. Our key approximation is setting $x_1^{approx} = x_0$ instead of storing the true $x_1$, introducing error $\epsilon = x_1^{orig} - x_0$. This approximation enables practical watermark detection without requiring the intermediate diffusion state $x_1$, though it introduces bounded reconstruction error that our theorem quantifies. We will clarify this notation distinction between the true diffusion state $x_1$ and our practical approximation $x_1^{approx} = x_0$ in the revision.
>
> **[Q10] Clarify the calculation of bit accuracy eqn(15) when $s$ is not binary.**
>
> [QA10] Yes, Equation (15) can be applied identically for non-binary watermarks. The bit accuracy formula $\text{Acc} = \frac{1}{|W^{\*}|F} \sum_{w \in W^*} \sum_{f=1}^{F} \mathbb{I}[\hat{s}^{w,f} = s^{w,f}]$ compares recovered seeds $\hat{s}^{w,f}$ with ground truth seeds $s^{w,f}$ regardless of the size $L$. When $L > 2$ (non-binary), each seed takes values in $\\{0, 1, ..., L-1\\}$, and the indicator function $\mathbb{I}[\cdot]$ evaluates to 1 only when the recovered value exactly matches the original value. For this setting, the expected bit accuracy for time series without watermark is $1/L$.
>
> However, for our experiments in Tables 16-18 in Appendix E.7.2 with $L=3$ and $4$, we employed the valid bit mechanism from TabWak [1], which focuses on the tail regions of the distribution for superior noise resilience. This mechanism specifically targets extreme quantiles when calculating bit accuracy to detect whether values have flipped to the opposite side of the Gaussian distribution. For example, with $L=4$, we focus exclusively on bits 0 and 3, which correspond to the 0-25% and 75%-100% quantiles in the Gaussian distribution. We then verify whether the sign for these tail values has flipped, which enhances watermark detectability. We will provide a more detailed explanation of this mechanism in the next version.
>
> **[Q11] Z-score values too big.**
>
> [QA11] These extremely high Z-scores (ranging from ~50 to >500) are actually not uncommon in watermarking literature like TabWak [1] and represent a desirable outcome indicating strong watermark detectability. The large values arise from our evaluation setup using 1,000 samples (as same as TabWak), where the Z-score formula $Z = \frac{\mu_{Acc,W} - \mu_{Acc,NW}}{\sigma_{Acc,NW}/\sqrt{n}}$ amplifies differences by $\sqrt{1,000} \approx 31.6$. When watermarked samples achieve ~90% bit accuracy versus ~50% for non-watermarked samples, this creates substantial differences that, combined with moderate sample size, produce Z-scores in the hundreds.
>
> **[Q12] Explain the increase in Z-score after the attack on TimeWak.**
>
> [QA12] We acknowledge that this counter-intuitive phenomenon appears for some baselines in Table 2. However, we want to clarify that for our proposed method, TimeWak, the Z-score consistently decreases as the attack strength increases across almost all scenarios, which is the expected behavior. The increase in Z-score is primarily observed for less robust methods like Tree-Ring and Gaussian Shading. We hypothesize this is an artifact of how the Z-score is calculated and the nature of these non-native watermarking schemes. The Z-score measures the statistical deviation of an *attacked, watermarked* sample relative to *clean, non-watermarked* data. For methods not designed for time series, a strong attack can drastically alter the data's statistical properties in a way that, while destroying the original watermark, makes the data even *more* different from the clean distribution.
>
> ### References
> [1] Zhu, Chaoyi, et al. "TabWak: A Watermark for Tabular Diffusion Models." ICLR, 2025.

---

### Official Review · Reviewer_U6K7 · 2025-06-30

**Clarity:** 3
**Significance:** 2
**Originality:** 2
**Rating:** 4
**Confidence:** 3

**Summary:**

This work proposes TimeWak, the first sampling-time watermarking scheme for multivariate time series diffusion models. The method introduces a temporal chained-hashing scheme that embeds watermark seeds along the temporal dimension. In addition, it leverages ε-exact inversion to address inconsistencies in DDIM inversion. Experimental results demonstrate the effectiveness of TimeWak on multivariate time series data.

**Questions:**

1.	Please clarify difference from prior work, particularly in distinguishing its temporal chained-hashing and ε-exact inversion components from existing methods such as TabWak or BDIA-based inversion techniques.
2.	The ε-exact inversion assumption relies on the Lipschitz continuity of the noise estimator, which may not hold in practice—especially for time series data with abrupt transitions, feature scale heterogeneity, or distributional shifts between training and inference. For example, real-world datasets such as medical records may contain mixed feature types (e.g., binary gender and continuous income), or neurological signals like fMRI may include sudden pathological events (e.g., seizure episodes), where small input perturbations can lead to disproportionately large changes in the estimated noise. A discussion or empirical analysis of failure modes under such scenarios would be helpful.
3.	On the fMRI dataset, TabWak⊤ outperforms TimeWak in several Z-score metrics. Could you explain why TimeWak underperforms in this setting? Are there certain types of time series (e.g., low-signal or highly correlated) where your method is less effective?

**Ethical Concerns:**

["NO or VERY MINOR ethics concerns only"]

**Final Justification:**

After carefully reading the authors’ rebuttal and the other reviewers’ comments, I find that the authors have addressed my questions and largely alleviated my concerns. Based on this, I will raise my rating.

**Limitations:**

Yes

**Paper Formatting Concerns:**

1.	line 34 “offerring

**Quality:**

3

**Strengths And Weaknesses:**

Strengths:

1.	The proposed method effectively tackles the watermarking problem in multivariate time series data, a scenario where existing techniques for watermarking diffusion models or tabular data fall short.
2.	This work conducts an in-depth analysis of watermarking multivariate time series, revealing unique challenges such as spatial heterogeneity and temporal dependence, and design a tailored solution accordingly.
3.	Theoretical analysis combined with extensive experimental results demonstrates the effectiveness and robustness of the proposed method for the target task.

Weaknesses:

1.	Although TimeWak is the first sampling-time watermarking scheme for multivariate time series diffusion models, the main components: ε-exact inversion and chained-hashing, conceptually resemble existing techniques used in image and tabular watermarking methods such as TabWak, which raises concerns about the technical novelty and methodological complexity of the proposed approach.

2.	While the authors mention that the method "can be applied to streaming scenarios," it relies on access to the diffusion model’s inversion process, as well as full-sequence segmentation and predefined hash functions. These assumptions make deployment in practical, streaming, or black-box settings quite challenging.

3.	On the fMRI dataset, TimeWak performs worse than TabWak in terms of both robustness and detection accuracy, which raises the question of whether the proposed approach might underperform in real-world or specialized domains compared to existing techniques.

4.  The ε-exact inversion assumption relies on the Lipschitz continuity of the noise estimator, which may not hold in practice. The paper does not clearly explain the magnitude of this bound, nor whether it meaningfully impacts detection performance in practice.

---

> ### Author Rebuttal · Authors · 2025-07-31
>
> Thank you for your valuable comments.
>
> ### Weaknesses and Questions
> **[W1/Q1] Clarify differences from prior works (TabWak or BDIA).**
>
> [WA1/QA1] Thanks for pointing this out. The main differences are as follows:
>
> - With TabWak:
>
>     First, they operate in different spaces. TabWak is designed for the latent space, where reconstruction error is dominated by the autoencoder. In contrast, TimeWak operates directly in the real data space, where error stems from the diffusion inversion process itself. This critical difference motivates our use of a modified Bi-directional Integration Approximation (BDIA), which is specifically designed to minimize this diffusion inversion error and significantly improves detectability, as validated by our ablation study in Appendix E.5 (Table 12).
>
>     Second, the original TabWak employs a self-cloning mechanism which copies half of the watermark seed, a method that perturbs the initial noise distribution. In contrast, our temporal chained-hashing mechanism is designed to better preserve the statistical properties of the initial noise, resulting in higher-quality synthetic data.
>
>     Finally, TabWak verifies the watermark along the feature axis. Our analysis reveals that reconstruction error is unevenly distributed across features, which compromises detection reliability. By performing our chained-hashing along the more stable time axis, TimeWak directly mitigates the impact of this uneven error distribution, leading to demonstrably better watermark detectability.
>
> - With BDIA:
>
>     Standard BDIA assumes exact knowledge of $x_1$, which is unavailable in practical watermark detection. Our $\epsilon$-exact adaptation approximates $x_1 \approx x_0$, introducing controlled error bounds that standard BDIA cannot handle. Theorem 3.1 provides theoretical guarantees for this approximation. Our contribution quantifies how approximation errors propagate through the inversion process, enabling reliable watermark detection despite imperfect inversion.
>
>     We generated 10,000 samples (length 64) across five datasets, computing $x_0$ and $x_1$. Results in Table A show consistently small $L_1$ norms between $x_1$ and $x_0$ (avg: 5.1-7.0e-03, max: <0.23), validating our $\epsilon$-exact approximation. These empirically small errors confirm Theorem 3.1's theoretical bounds and demonstrate reliable watermark detection despite the approximation.
>
> *Table A: $L_1$ norms between $x_1$ and $x_0$ for 64-length sequences over 10,000 samples.*
> Dataset|Avg. $L_1$|Max. $L_1$
> ---|---|---
> Stocks|7.031e-03|0.082
> ETTh|6.776e-03|0.104
> MuJoCo|5.146e-03|0.081
> Energy|5.687e-03|0.230
> fMRI|5.945e-03|0.070
>
>
> **[W2] Practicality in streaming/black-box settings.**
>
> [WA2] We acknowledge this limitation in the paper (Lines 308-311). While the current implementation uses fixed-length windows for simplicity, the core temporal chained-hashing mechanism is adaptable. For streaming data applications, it can be readily deployed using a sliding-window approach, where a fixed-length model generates watermarks for each temporal window. To evaluate the practical feasibility of real-time deployment, we measured the watermark detection overhead using a single NVIDIA L40S GPU and Intel(R) Xeon(R) Platinum 8562Y+ CPU. Table B presents the computational overhead for watermark detection across different datasets and configurations. The results demonstrate that detection overhead remains consistently low, ranging from approximately 1.5 to 7.5 seconds depending on the dataset complexity and batch size. We consider this overhead acceptable for streaming scenarios, particularly given the security benefits provided by the watermarking system. Adapting the hashing mechanism to handle variable-length sequences without padding represents a promising direction for future research that could further enhance the method's applicability to diverse real-world scenarios.
>
> And TimeWak is designed for the standard "white-box" watermarking scenario where the entity generating the data also embeds the watermark for auditing and tracking purposes. In this setting, the model is fully accessible. The literature distinguishes this from "black-box" watermarking, where the watermark must be detected with only query access to the model. We acknowledge that a black-box detection scenario is a much more challenging problem that is out of the scope of this work.
>
> *Table B: Watermark detection overhead in seconds for TimeWak.*
> Dataset|Window Size|Batch Size = 1 (sec.)|Batch Size = 100 (sec.)
> ---|---|---|---
> Stocks|24|1.58|2.09
> ||64|1.50|2.27
> ||128|1.55|2.59
> ETTh|24|1.72|2.23
> ||64|1.64|2.41
> ||128|1.63|2.62
> MuJoCo|24|2.96|3.84
> ||64|2.90|4.25
> ||128|2.90|4.49
> Energy|24|3.89|5.22
> ||64|3.88|5.73
> ||128|3.89|6.58
> fMRI|24|4.69|6.31
> ||64|4.75|7.00
> ||128|4.81|7.47
>
>
> **[W3/Q3] Explain TimeWak underperforms TabWak$^\top$ on fMRI dataset.**
>
> [WA3/QA3] For the fMRI dataset, increasing the interval length $H$ in TimeWak leads to higher Z-score while preserving the quality of the synthetic data and improving robustness, as shown in Tables C (full results in Table 14 in Appendix E.7.1) and D. This highlights that TimeWak can, in fact, perform exceptionally well on the fMRI dataset, especially when compared to other baselines.
>
> *Table C: Results of synthetic time series quality and watermark detectability for 64-length sequences. TimeWak is applied with interval length $H=8$.*
> Method|Context-FID (↓)|Correlational (↓)|Discriminative (↓)|Predictive (↓)|Z-score (↑)
> ---|---|---|---|---|---
> TabWak$^\top$|0.554|1.955|0.331|**0.100**|743.33
> TimeWak|**0.469**|**1.823**|**0.294**|**0.100**|**817.23**
>
> *Table D: Results of robustness against post-editing attacks for 64-length sequences. TimeWak is applied with interval length $H=8$.*
> Method|Offset 5% (↑)|Offset 30% (↑)|Random Crop 5% (↑)|Random Crop 30% (↑)|Min-Max Insertion 5% (↑)|Min-Max Insertion 30% (↑)
> ---|---|---|---|---|---|---
> TabWak$^\top$|743.16|742.43|**636.28**|**317.24**|614.25|224.27
> TimeWak|**803.01**|**801.83**|610.02|151.69|**663.14**|**249.69**
>
> To further evaluate TimeWak, we added two additional and more challenging real-world datasets: (i) the ILI dataset, which records influenza-like illness cases in the United States, and (ii) the Weather dataset, which is sparse and noisy. Both datasets are standard benchmarks used in TimesNet [8]. As shown in Table E, TimeWak demonstrates strong performance on these challenging datasets.
>
> *Table E: Results of synthetic time series quality and watermark detectability for 64-length sequences.*
> Dataset|Method|Context-FID (↓)|Correlational (↓)|Discriminative (↓)|Predictive (↓)|Z-score (↑)
> ---|---|---|---|---|---|---
> ILI|W/O|0.411|0.073|0.147|0.028|-
> ||TR|1.530|0.149|0.286|0.035|5.09
> ||GS|0.734|0.159|0.397|0.030|78.18
> ||HTW|0.439|**0.069**|0.217|0.032|7.37
> ||TabWak|**0.239**|0.070|0.131|**0.027**|-2.06
> ||TabWak$^\top$|0.295|0.071|0.114|0.028|21.26
> ||TimeWak|0.240|0.076|**0.111**|0.028|**151.03**
> Weather|W/O|0.647|1.429|0.175|0.002|-
> ||TR|3.381|2.518|0.388|**0.002**|0.40
> ||GS|4.495|2.194|0.446|**0.002**|15.36
> ||HTW|0.712|1.463|0.190|**0.002**|4.82
> ||TabWak|0.751|1.571|0.200|**0.002**|-1.23
> ||TabWak$^\top$|**0.588**|1.369|**0.178**|**0.002**|39.58
> ||TimeWak|0.717|**0.951**|0.184|**0.002**|**205.53**
>
>
> **[W4/Q2] $\epsilon$-Exact inversion: Lipschitz continuity assumption.**
>
> [WA4/QA2] Thanks for point this point. This is a common assumption in the theoretical analysis of diffusion models/neural networks. Recent works have extensively studied the smoothness properties of diffusion models, providing theoretical guarantees for their stability and convergence under such assumptions [1-7]. While its validity in practice is an area of active research, we empirically validated this assumption in Appendix C (Figure 5, Lines 559-561), showing the Lipschitz constant is well-behaved for the models used.  In this figure, we empirically computed $\Delta_{t}$ (Lipschitz constant) across four different datasets, using 10,000 samples from each. Specifically,  $\Delta_{t}$ is calculated as the maximum ratio for different $t$ using the $L_1$ norm of the data samples.  At the first timestep, the constant is relatively large, ranging from 28-80 for the four different datasets, and then the constant decreases to a value smaller than 1 as the timestep increases, which is also according to the observation from existing work [7].
>
> ### Paper Formatting Concerns
> **[PFC1] Typo on line 34 "offerring".**
>
> [PFCA1] Thank you for pointing this out. We will fix it in the next version.
>
> ### References
> [1] Liang, Yingyu, et al. "Unraveling the smoothness properties of diffusion models: A gaussian mixture perspective." arXiv:2405.16418, 2024.
>
> [2] Mooney, Connor, et al. "Global well-posedness and convergence analysis of score-based generative models via sharp Lipschitz estimates." ICLR, 2025.
>
> [3] Conforti, Giovanni, Alain Durmus, and Marta Gentiloni Silveri. "KL convergence guarantees for score diffusion models under minimal data assumptions." SIAM Journal on Mathematics of Data Science 7.1 (2025): 86-109.
>
> [4] Lim, Jae Hyun, et al. "Score-based diffusion models in function space." arXiv:2302.07400, 2023.
>
> [5] Chen, Sitan, et al. "Sampling is as easy as learning the score: theory for diffusion models with minimal data assumptions." ICLR, 2023.
>
> [6] Han, Yinbin, Meisam Razaviyayn, and Renyuan Xu. "Neural network-based score estimation in diffusion models: Optimization and generalization." ICLR, 2024.
>
> [7] Yang, Zhantao, et al. "Lipschitz singularities in diffusion models." ICLR, 2024.
>
> [8] Wu, Haixu, et al. "TimesNet: Temporal 2D-Variation Modeling for General Time Series Analysis." ICLR, 2023.

---

> > ### Comment · Reviewer_U6K7 · 2025-08-08
> > **Response to the rebuttal**
> >
> > After carefully reading the authors’ rebuttal and the other reviewers’ comments, I find that the authors have addressed my questions and largely alleviated my concerns. Based on this, I will raise my rating.

---

### Official Review · Reviewer_gwg6 · 2025-07-07

**Clarity:** 3
**Significance:** 3
**Originality:** 3
**Rating:** 4
**Confidence:** 3

**Summary:**

In this work, the authors propose a novel watermarking method for multivariate time series diffusion models. This method embeds watermarks directly in the real space of time series data. It leverages a temporal chained-hashing scheme to recover watermarks robustly and accurately. The approach is evaluated against five strong baselines on five datasets. The results demonstrate the effectiveness.

**Questions:**

See the weakness

**Ethical Concerns:**

["NO or VERY MINOR ethics concerns only"]

**Limitations:**

Yes

**Quality:**

3

**Strengths And Weaknesses:**

The paper introduces a novel temporal chained-hashing scheme that accounts for both feature heterogeneity and temporal dependencies in time series data. This chaining mechanism, combined with feature permutation, provides strong resilience against inversion errors and post-processing attacks.

However, the approach relies on fixed-length intervals for seed chaining, which may limit its adaptability to time series with irregular event distributions or variable-length sequences—particularly in streaming applications.

Although TimeWak demonstrates superior performance over existing baselines, most of the comparisons involve methods originally designed for image or tabular data (e.g., TabWak, GS). Incorporating evaluations against more recent or time series-specific watermarking techniques would further strengthen the empirical validation.

---

> ### Author Rebuttal · Authors · 2025-07-31
>
> Thank you for your valuable comments and positive feedback.
>
> ### Weaknesses
>
> **[W1] Rely on fixed-length intervals during seed chaining.**
>
> [WA1] We acknowledge this limitation in the paper (Lines 308-311). While the current implementation uses fixed-length windows for simplicity, the core mechanism of temporal chained hashing is inherently adaptable. For streaming data applications, it can be readily deployed using a sliding-window approach, where a model generating fixed-length output sequences embeds watermarks for each temporal window. To evaluate the practical feasibility of real-time deployment, we measured the watermark detection overhead using a single NVIDIA L40S GPU and Intel(R) Xeon(R) Platinum 8562Y+ CPU. Table A presents the computational overhead for watermark detection across different datasets and configurations. The results demonstrate that detection overhead remains consistently low, ranging from approximately 1.5 to 7.5 seconds depending on the dataset complexity and batch size. We consider this overhead acceptable for streaming scenarios, particularly given the security benefits provided by the watermarking system. Adapting the hashing mechanism to handle variable-length sequences without padding represents a promising direction for future research that could further enhance the method's applicability to diverse real-world scenarios.
>
> *Table A: Watermark detection overhead in seconds for TimeWak.*
> Dataset|Window Size| Batch Size = 1 (sec.)|Batch Size = 100 (sec.)
> ---|---|---|---
> Stocks|24|1.58|2.09
> ||64|1.50|2.27
> ||128|1.55|2.59
> ETTh|24|1.72|2.23
> ||64|1.64|2.41
> ||128|1.63|2.62
> MuJoCo|24|2.96|3.84
> ||64|2.90|4.25
> ||128|2.90|4.49
> Energy|24|3.89|5.22
> ||64|3.88|5.73
> ||128|3.89|6.58
> fMRI|24|4.69|6.31
> ||64|4.75|7.00
> ||128|4.81|7.47
>
> **[W2] Compare with more recent or time series-specific watermarking methods.**
>
> [WA2] This is a valid point. We felt it was a necessary choice, as TimeWak's novelty is being the first time series generation watermarking scheme for diffusion models operating in data space, meaning there are no direct baselines for time series diffusion models. To make the evaluation more fair, we devised a two-part comparison strategy:
>
> 1. Against State-of-the-Art Watermarking Methods: We adapted the most prominent and recent watermarking methods from other domains. Tree-Ring [1] and Gaussian Shading [2] are state-of-the-art for image diffusion models, while TabWak [3] is the leading method for tabular data. By creating a stronger, time series-specific version of TabWak (TabWak$^\top$), we ensured the most rigorous comparison possible against the closest existing paradigms.
>
> 2. Against Time Series-Specific Methods: We also compared TimeWak against Heads-Tails Watermark (HTW) [4], a post-processing watermarking method designed specifically for time series. Our results consistently show that TimeWak provides significantly higher watermark detectability and robustness against attacks (Tables 1 & 2 in the paper) compared to HTW.
>
> This comprehensive comparison demonstrates that TimeWak performs better than SotA watermarking methods adapted from other modalities and post-processing time series-specific watermarking methods.
>
> ### References
> [1] Wen, Yuxin, et al. "Tree-Rings Watermarks: Invisible Fingerprints for Diffusion Images." NeurIPS, 2023.
>
> [2] Yang, Zijin, et al. "Gaussian Shading: Provable Performance-Lossless Image Watermarking for Diffusion Models." CVPR, 2024.
>
> [3] Zhu, Chaoyi, et al. "TabWak: A Watermark for Tabular Diffusion Models." ICLR, 2025.
>
> [4] N.J.I. van Schaik. "Robust Watermarking in Large Language Models for Time Series Generation." Thesis, Delft University of Technology, 2024.

---

### Official Review · Reviewer_j3Fa · 2025-07-08

**Clarity:** 3
**Significance:** 3
**Originality:** 3
**Rating:** 5
**Confidence:** 3

**Summary:**

This paper introduces TimeWak, a novel watermarking algorithm designed specifically for multivariate time series generated by diffusion models. Unlike existing approaches that operate in latent space, TimeWak embeds watermarks directly in the real-valued temporal-feature space, addressing the unique challenges of temporal dependencies and feature heterogeneity. The method employs a temporal chained-hashing mechanism combined with feature-wise permutation to maintain temporal consistency and watermark robustness. To improve detectability, the authors introduce an ϵ-exact inversion approach based on a modified bidirectional DDIM process, providing theoretical error bounds for reconstruction. Extensive evaluations across five datasets demonstrate TimeWak's effectiveness in preserving data utility, enhancing watermark detectability, and maintaining robustness under various post-editing attacks.

**Questions:**

Please see the weaknesses

**Ethical Concerns:**

["NO or VERY MINOR ethics concerns only"]

**Final Justification:**

The rebuttal addressed my main concerns through substantial new experiments: (i) robustness against a knowledgeable adversary via a DiffPure-based reconstruction attack, (ii) strong performance in realistic mixed-data scenarios, (iii) negligible impact on downstream forecasting and imputation tasks, (iv) resilience on challenging noisy/sparse datasets, and (v) preservation of key temporal characteristics. The only remaining gap is the absence of a formal threat model and human-in-the-loop evaluation, which I consider suitable for future work. Given the strengthened empirical evidence and practical relevance, I recommend acceptance.

**Limitations:**

yes

**Paper Formatting Concerns:**

The supplementary material is included within the main paper file rather than provided as a separate document.

**Quality:**

3

**Strengths And Weaknesses:**

$\textbf{Strengths:}$

1 -- The paper addresses a clear and underexplored gap in the field, watermarking of synthetic multivariate time series generated in real space by diffusion models, where latent-space methods fail. This positions the work as a timely and relevant contribution.

2 -- The introduction of temporal chained-hashing combined with feature-wise seed permutation presents a novel watermarking strategy that explicitly accounts for the temporal and heterogeneous nature of time series data. The integration of ϵ-exact inversion using BDIA adds theoretical depth to the detection mechanism.

3 -- The authors conduct extensive empirical analysis across five datasets and multiple post-editing attacks, comparing against both native and adapted state-of-the-art baselines. The results convincingly demonstrate the method’s superior trade-off between watermark detectability and data quality.

$\textbf{Weaknesses:}$

1 -- Although the paper evaluates robustness against various post-processing operations, it does not provide a formal security analysis or define an adversarial threat model. Critical aspects such as watermark forgery, key leakage, or deliberate evasion by a knowledgeable attacker are not addressed. Given that this is one of the first works on time-series watermarking in diffusion models, evaluating such adversarial robustness is essential to fully establish the method’s practical reliability and trustworthiness.

2 -- While the paper presents a thorough set of experiments, all evaluations are conducted on synthetic datasets using controlled, predefined attacks. The method is not tested in more realistic scenarios, such as when synthetic and real time series are mixed together (as often happens in training or deployment), or when the data is used in downstream tasks involving human interpretation or decision-making. Without such evaluation, it is unclear how well the method would perform in practical, real-world settings.

3 -- As far as I understand, all the datasets used in the experiments are structured, regularly sampled, and relatively clean. There is no evaluation on noisy, sparse, or irregularly sampled time series, which are common in real-world domains like healthcare, IoT, or sensor networks. This raises concerns about how well TimeWak would generalize to such challenging and realistic data conditions.

4 -- The paper does not evaluate how watermarking affects the preservation of key signal characteristics in time series data, such as trends, seasonality, or anomalies. In domains like healthcare or finance, where subtle temporal patterns are crucial for downstream tasks, even minor distortions could impact the reliability of model outputs. Without such analysis, it is unclear whether the method maintains the integrity of information needed for sensitive or high-stakes applications.

---

> ### Author Rebuttal · Authors · 2025-07-31
>
> Thank you for your valuable comments and positive feedback.
>
> ### Weaknesses
> **[W1] Lack of formal security analysis.**
>
> [WA1] Thank you for raising this important point. We agree that conducting a formal security analysis against knowledgeable adversaries is a crucial next step for this research. In this paper, we also implemented and evaluated a reconstruction attack in Appendix E.2.2 based on the DiffPure [1]. In this attack scenario, we assume the adversary has direct access to the diffusion model and attempts to use it to perform reconstruction-based watermark removal. As demonstrated in Table 8, our watermarking scheme maintains robust performance even under this adversarial setting.
>
> **[W2] Test on more realistic scenarios, or when the data is used in downstream tasks.**
>
> [WA2] To address your concerns, we conducted the following additional experiments:
> 1) True positive rate evaluation in a mixed data setting
> 2) Evaluation on more downstream tasks
>
> We constructed a mixed dataset containing equal proportions (1/3 each) of real data, synthetic data without watermarks, and synthetic data with watermarks, totaling 100 trials with a window length of 64. We evaluated TPR\@0.1% FPR with 1, 10, and 20 samples per record, as shown in Table A. For comparison, we selected the methods with the best detectability: GS and TabWak$^\top$. The results demonstrate that TimeWak achieves 99-100% true positive rates in this mixed data setting when the number of samples is 20. But GS completely fails to detect watermarks in the MuJoCo dataset (0% TPR across all sample sizes), while TabWak$^\top$ fails on both MuJoCo (0% TPR) and Energy datasets (0-1% TPR).
>
> *Table A: Results of TPR\@0.1% FPR on mixed dataset when number of samples are 1, 10, and 20.*
> Dataset|Method|1 (↑)|10 (↑)|20 (↑)
> ---|---|---|---|---
> Stocks|GS|**0.33**|0.87|**0.99**
> ||TabWak$^\top$|0.13|0.47|0.68
> ||TimeWak|0.22|**0.96**|**0.99**
> ETTh|GS|0.43|**0.98**|**1.0**
> ||TabWak$^\top$|**0.49**|0.91|0.99
> ||TimeWak|0.38|0.92|**1.0**
> MuJoCo|GS|0.0|0.0|0.0
> ||TabWak$^\top$|0.0|0.0|0.0
> ||TimeWak|**0.34**|**0.84**|**0.99**
> Energy|GS|**0.42**|**1.0**|**1.0**
> ||TabWak$^\top$|0.01|0.0|0.0
> ||TimeWak|0.32|0.97|**1.0**
> fMRI|GS|**0.4**|0.98|**1.0**
> ||TabWak$^\top$|0.31|**1.0**|**1.0**
> ||TimeWak|0.26|0.97|**1.0**
>
> Due to time limitations, we are unable to evaluate downstream tasks that involve human interpretation or decision-making. For instance, in a time series classification task, models are typically trained on labeled datasets. While our approach can generate large amounts of watermarked synthetic time series data, the corresponding labels for these synthetic sequences are not available, making it infeasible to use them directly for training in such tasks. Hence, for the downstream tasks evaluation, we assessed the utility of watermarked synthetic data on real-world forecasting and imputation tasks. Models were trained on different data types (real, synthetic without watermark, and synthetic with TimeWak watermark) and tested on real data.
>
> Tables B and C show that synthetic data with our watermark achieves comparable MSE performance to non-watermarked synthetic data in both forecasting and imputation tasks, demonstrating that our watermarking method preserves data utility.
>
> *Table B: Results of time series forecasting that train on real and synthetic data and test on real data with a 24 timesteps forecast horizon. MSE values ($×10^{-3}$).*
> Dataset|Training Data|MSE $×10^{-3}$ (↓)
> ---|---|---
> Stocks|Real|2.119
> ||Synth$_{\mathrm{W/O}}$|2.022
> ||Synth$_{\mathrm{TimeWak}}$|2.014
> ETTh|Real|6.678
> ||Synth$_{\mathrm{W/O}}$|8.541
> ||Synth$_{\mathrm{TimeWak}}$|8.655
> MuJoCo|Real|1.312
> ||Synth$_{\mathrm{W/O}}$|1.615
> ||Synth$_{\mathrm{TimeWak}}$|1.741
> Energy|Real|12.715
> ||Synth$_{\mathrm{W/O}}$|13.480
> ||Synth$_{\mathrm{TimeWak}}$|13.717
> fMRI|Real|36.423
> ||Synth$_{\mathrm{W/O}}$|67.796
> ||Synth$_{\mathrm{TimeWak}}$|67.944
>
> *Table C: Results of time series imputation that train on real and synthetic data and test on real data with 70% missing ratio. MSE values ($×10^{-3}$).*
> Dataset|Training Data|MSE $×10^{-3}$ (↓)
> ---|---|---
> Stocks|Real|1.020
> ||Synth$_{\mathrm{W/O}}$|0.855
> ||Synth$_{\mathrm{TimeWak}}$|0.858
> ETTh|Real|1.526
> ||Synth$_{\mathrm{W/O}}$|1.842
> ||Synth$_{\mathrm{TimeWak}}$|1.963
> MuJoCo|Real|0.101
> ||Synth$_{\mathrm{W/O}}$|0.364
> ||Synth$_{\mathrm{TimeWak}}$|0.387
> Energy|Real|7.926
> ||Synth$_{\mathrm{W/O}}$|8.258
> ||Synth$_{\mathrm{TimeWak}}$|8.279
> fMRI|Real|27.439
> ||Synth$_{\mathrm{W/O}}$|45.412
> ||Synth$_{\mathrm{TimeWak}}$|47.349
>
> **[W3] Evaluate on challenging time series datasets.**
>
> [WA3] We initially considered the fMRI dataset as a realistic dataset, as it is high-dimensional, noisy, and has complex temporal dependencies. However, as you raise a valid point on realistic conditions, we evaluate TimeWak on 2 additional and more challenging real-world datasets, (i) the ILI dataset, which records influenza-like illness cases in the United States, and (ii) the Weather dataset, which is sparse and noisy. Both datasets are standard benchmarks, and are used in TimesNet [2]. Results run on these datasets are shown in Table D, where TimeWak achieves robust watermark detectability while preserving the quality of the synthetic data.
>
> *Table D: Results of synthetic time series quality and watermark detectability for 64-length sequences.*
> Dataset|Method|Context-FID (↓)|Correlational (↓)|Discriminative (↓)|Predictive (↓)|Z-score (↑)
> ---|---|---|---|---|---|---
> ILI|W/O|0.411|0.073|0.147|0.028|-
> ||TR|1.530|0.149|0.286|0.035|5.09
> ||GS|0.734|0.159|0.397|0.030|78.18
> ||HTW|0.439|**0.069**|0.217|0.032|7.37
> ||TabWak|**0.239**|0.070|0.131|**0.027**|-2.06
> ||TabWak$^\top$|0.295|0.071|0.114|0.028|21.26
> ||TimeWak|0.240|0.076|**0.111**|0.028|**151.03**
> Weather|W/O|0.647|1.429|0.175|0.002|-
> ||TR|3.381|2.518|0.388|**0.002**|0.40
> ||GS|4.495|2.194|0.446|**0.002**|15.36
> ||HTW|0.712|1.463|0.190|**0.002**|4.82
> ||TabWak|0.751|1.571|0.200|**0.002**|-1.23
> ||TabWak$^\top$|**0.588**|1.369|**0.178**|**0.002**|39.58
> ||TimeWak|0.717|**0.951**|0.184|**0.002**|**205.53**
>
> **[W4] Evaluate how watermarking affects the preservation of key signal characteristics in time series data.**
>
> [WA4] Thanks for pointing this out. To assess the preservation of key signal characteristics, we added 4 additional metrics from TSGBench [3]: Marginal Distribution Difference (MDD), AutoCorrelation Difference (ACD), Skewness Difference (SD), and Kurtosis Difference (KD). These measures are designed to capture inter-series correlations and temporal dependencies, thereby evaluating how well the generated time series preserves the original characteristics. For all these metrics, the lower the score, the better. As shown in Table E, TimeWak consistently achieves scores very close to the non-watermarked (W/O) baseline across all datasets, indicating minimal distortion introduced by the watermark.
>
> *Table E: Results of 4 additional metrics for 64-length sequences.*
> Dataset|Method|MDD (↓)|ACD (↓)|SD (↓)|KD (↓)
> ---|---|---|---|---|---
> Stocks|W/O|0.491|0.044|0.473|1.522
> ||TR|0.881|0.445|0.759|4.408
> ||GS|1.063|0.444|0.727|4.889
> ||HTW|0.491|**0.044**|0.473|1.522
> ||TabWak|0.470|0.078|0.180|0.544
> ||TabWak$^\top$|0.478|0.098|**0.060**|0.852
> ||TimeWak|**0.431**|0.068|0.103|**0.295**
> ETTh|W/O|0.176|0.421|0.255|1.359
> ||TR|0.546|1.128|0.633|3.474
> ||GS|0.712|1.581|0.723|2.725
> ||HTW|0.384|**0.459**|0.276|1.468
> ||TabWak|0.211|0.532|0.245|1.422
> ||TabWak$^\top$|**0.183**|0.753|**0.141**|**0.707**
> ||TimeWak|0.184|0.511|0.197|1.109
> MuJoCo|W/O|0.379|0.225|0.062|0.306
> ||TR|1.296|1.494|0.426|1.555
> ||GS|1.500|1.550|0.533|1.221
> ||HTW|0.941|0.434|0.076|**0.287**
> ||TabWak|0.420|**0.262**|0.087|0.311
> ||TabWak$^\top$|0.521|0.532|0.084|0.351
> ||TimeWak|**0.394**|0.300|**0.069**|0.340
> Energy|W/O|0.188|0.157|0.112|0.703
> ||TR|0.558|0.606|0.355|1.681
> ||GS|0.660|0.958|0.373|1.015
> ||HTW|0.363|**0.148**|**0.108**|0.701
> ||TabWak|0.235|0.270|0.126|0.618
> ||TabWak$^\top$|0.263|0.336|0.112|0.682
> ||TimeWak|**0.215**|0.241|0.111|**0.601**
> fMRI|W/O|0.099|0.153|0.041|0.128
> ||TR|0.447|1.356|0.066|0.428
> ||GS|0.818|1.435|0.096|0.174
> ||HTW|0.452|0.151|0.048|0.133
> ||TabWak|0.126|0.159|0.043|0.138
> ||TabWak$^\top$|0.127|0.261|**0.040**|**0.114**
> ||TimeWak|**0.120**|**0.148**|**0.040**|0.132
>
>
> ### References
> [1] Nie, Weili, et al. "Diffusion models for adversarial purification." ICML, 2022.
>
> [2] Wu, Haixu, et al. "TimesNet: Temporal 2D-Variation Modeling for General Time Series Analysis." ICLR, 2023.
>
> [3] Ang, Yihao, et al. "TSGBench: Time Series Generation Benchmark." Proc. VLDB Endow. 17, 3 (2023), 305–318.

---

> > ### Comment · Reviewer_j3Fa · 2025-08-08
> >
> > I thank the authors for their thorough and well-structured rebuttal, which directly addresses my primary concerns.
> >
> > The inclusion of a DiffPure-based reconstruction attack provides evidence of robustness even against an informed adversary, partially addressing the earlier call for a security perspective. The new downstream evaluations on forecasting and imputation tasks show that TimeWak watermarks have negligible impact on predictive utility, alleviating concerns about distortion in practical applications. The added results on challenging datasets (ILI, Weather) confirm the method’s resilience in noisy, sparse, and irregular domains.
> >
> > For the final version, I encourage the authors to formalize an explicit adversarial threat model, explore human-in-the-loop downstream tasks, and investigate scalability to very long or irregular sequences. Clearer dataset descriptions and pre-processing details would also aid reproducibility. Overall, the rebuttal meaningfully strengthens the contribution, and I am raising my score from borderline accept to accept.

---

> > > ### Author Response · Authors · 2025-08-09
> > >
> > > Dear Reviewer j3Fa,
> > >
> > > We sincerely thank for your thoughtful feedback and for recognizing our rebuttal efforts. We are especially grateful for the decision to raise the rating, and we will incorporate these valuable recommendations to further strengthen the final version.

---

### Comment · Area_Chair_PZYD · 2025-08-08
**Please engage in the author-reviewer discussion ASAP - Deadline Aug 8, 11.59pm AoE**

Dear Reviewer j3Fa, gwg6, U6K7,

Thanks for your contribution in the reviewing process. The authors have provided their rebuttal. Please engage in the author-reviewer discussion as soon as possible as the discussion period ends today (Aug 8, 11.59pm AoE).

Please note “Mandatory Acknowledgement” button is to be submitted only when reviewers fulfill all conditions below (conditions in the acknowledgment form):
- read the author rebuttal
- engage in discussions (reviewers must talk to authors, and optionally to other reviewers and AC - ask questions, listen to answers, and respond to authors)
- fill in "Final Justification" text box and update “Rating” accordingly (this can be done upon convergence - reviewer must communicate with authors first)

Best regards,
AC

---

### Note · Authors · 2025-08-12

**Dear Reviewers, ACs, and PCs,**

Thank you for your dedication, support, and insightful feedback. We greatly appreciate your suggestions, which have significantly strengthened our work. Below is a summary of the key updates and improvements we have made:

- Added evaluation on mixed dataset and two downstream tasks: forecasting and imputation. (Reviewer j3Fa)
- Added evaluation on two additional challenging datasets used in TimesNet [1]: ILI and Weather. (Reviewers j3Fa, U6K7)
- Added four additional metrics from TSGBench [2] to assess how watermarking affects the preservation of key signal characteristics in time series data. (Reviewer j3Fa)
- Added evaluation on the computational overhead for TimeWak watermark detection. (Reviewers gwg6, U6K7)
- Clarified the baselines used in the experiments. (Reviewer gwg6)
- Clarified the novelty of TimeWak and its performance on fMRI dataset. (Reviewer U6K7)
- Clarified the $\epsilon$-exact inversion assumption. Computed $L_1$ norms between $x_1$ and $x_0$. (Reviewers U6K7, zoPw)
- Clarified the calculation of bit accuracy for non-binary watermarks. (Reviewer zoPw)
- Clarified the reason behind the specific choice within the inverse Gaussian CDF. (Reviewer zoPw)
- Clarified the abnormal Z-score values. (Reviewer zoPw)
- Clarified terms, notations and typos. (Reviewers U6K7, zoPw)


**Best regards,**

The Authors

### References
[1] Wu, Haixu, et al. "TimesNet: Temporal 2D-Variation Modeling for General Time Series Analysis." ICLR, 2023.

[2] Ang, Yihao, et al. "TSGBench: Time Series Generation Benchmark." Proc. VLDB Endow. 17, 3 (2023), 305–318.

---

### Decision · Program_Chairs · 2025-09-17

**Decision:**

Accept (spotlight)

**Comment:**

This paper proposes the first watermarking method for multivariate time series data generated by diffusion models in real space. This is a relevant contribution in watermarking methods that fills a gap in the scenarios where latent diffusion models fail. It introduces several key components including temporal chained-hashing and feature permutation to handle temporal dependence and spatial heterogeneity, and also provides theoretical error bounds for reconstruction. Extensive experiments on both synthetic and real-data mixed datasets demonstrate its efficacy and superiority again baselines in their respective settings.

Almost all concerns from reviewers were addressed properly with additional experiments and clarification during rebuttal. The authors also acknowledge a limitation of the absence of a formal threat model and human-in-the-loop evaluation, which reviewers agree it could be left for future work. As a result, all reviewers agree this work is worth acceptance at NeurIPS.